# Strategic Navigation or Stochastic Search?
# How Agents and Humans Reason Over Document Collections

**Łukasz Borchmann** ❄ **Jordy Van Landeghem** 🟥 **Michał Turski** ❄ **Shreyansh Padarha** 🏛
**Ryan Othniel Kearns** 🏛 **Adam Mahdi** 🏛 **Niels Rogge** 🤗 **Clémentine Fourrier** 🤗
**Siwei Han** 🏛 **Huaxiu Yao** 🏛 **Artemis Llabrés** ᶜᵛᶜ **Yiming Xu** ᶜᵛᶜ **Dimosthenis Karatzas** ᶜᵛᶜ
**Hao Zhang** ❄ **Anupam Datta** ❄

🏅 Snowflake/MADQA-Leaderboard   </> OxRML/MADQA   🗄 OxRML/MADQA

## Abstract

Multimodal agents offer a promising path to automating complex document-intensive workflows. Yet, a critical question remains: do these agents demonstrate genuine strategic reasoning, or merely stochastic trial-and-error search? To address this, we introduce MADQA, a benchmark of 2,250 human-authored questions grounded in 800 heterogeneous PDF documents. Guided by *Classical Test Theory*, we design it to maximize discriminative power across varying levels of agentic abilities. To evaluate agentic behavior, we introduce a novel protocol that measures the accuracy-effort trade-off. Using this framework, we show that while the best agents can match human searchers in raw accuracy, they succeed on largely different questions and rely on brute-force search to compensate for weak strategic planning. They fail to close the nearly 20% gap to oracle performance, persisting in unproductive loops. We release the dataset and evaluation harness to help facilitate the transition from brute-force retrieval to calibrated, efficient reasoning.

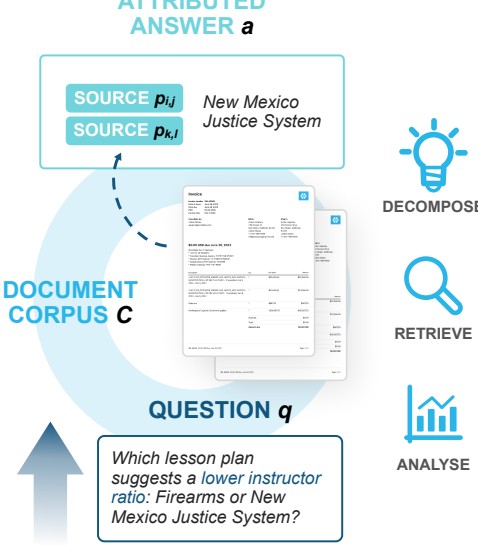

ATTRIBUTED ANSWER *a*

SOURCE $p_{i,j}$ — *New Mexico Justice System*
SOURCE $p_{k,l}$

DECOMPOSE

DOCUMENT CORPUS *C*

RETRIEVE

QUESTION *q*

*Which lesson plan suggests a lower instructor ratio: Firearms or New Mexico Justice System?*

ANALYSE

*Figure 1.* Given a query $q$ over corpus $\mathcal{C}$, the system iteratively retrieves pages, reasons over visual and textual content, and aggregates evidence from multiple pages $\mathcal{E} = \{p_{i,j}, \ldots\}$ to produce a grounded answer $a$ with attribution. The process typically requires decomposing $q$, iterative retrieval, and synthesizing across $\mathcal{E}$.

## 1. Introduction

We introduce *Multimodal Agentic Document QA* benchmark (MADQA). It focuses on the capabilities of multimodal large language model (MLLM) based *agentic* systems to handle complex, multi-stage information retrieval and reasoning tasks that one would encounter in enterprise settings.

### 1.1. Motivation and Related Works

Current benchmarks in this area operate in a fragmented landscape, as detailed in Table 1 (Appendix J.1 provides per-benchmark justifications).

*(i)* **Format.** Agent-centric benchmarks such as Researchy Questions (Rosset et al., 2025) and BRIGHT (Su et al., 2025) capture the complexity of "agentic research." How-

❄ Snowflake AI Research  🟥 Instabase  🏛 University of Oxford  🤗 Huggingface  🏛 UNC-Chapel Hill  ᶜᵛᶜ Computer Vision Center

Correspondence to: <lukasz.borchmann@snowflake.com>

*Proceedings of the 43rd International Conference on Machine Learning*, Seoul, South Korea. PMLR 306, 2026. Copyright 2026 by the author(s).

*Table 1.* **Comparing MADQA with existing benchmarks.** Our work is distinguished by its focus on a collection of complex PDFs with *fresh documents* (not recycled from existing benchmarks) and *fully human-authored questions*, designed to test agentic reasoning. We denote diversity levels using ● high, ● medium, and ● low as coarse, relative categories (see Appendix J.1 for detailed assessment).

| Name and Reference | Input File(s) | Diversity (Domains / Layouts) | Human -annotated | Problem Framing |
|---|---|---|---|---|
| DocVQA (Mathew et al., 2021) | Single document image | medium / medium | ✓ | **Document VQA** |
| InfographicVQA (Mathew et al., 2022) | Single image | medium / high | ✓ | |
| TAT-DQA (Zhu et al., 2022) | Mostly single-page PDF | low / low | ✓ | Question grounded |
| DUDE (Landeghem et al., 2023) | Multi-page PDF file | high / high | ✓ | on a single |
| MP-DocVQA (Tito et al., 2023) | Multi-page PDF | medium / medium | ✓ / ✗ | rich document |
| SlideVQA (Tanaka et al., 2023) | Series of slides | medium / low | ✓ | |
| M-LongDoc (Chia et al., 2025) | Multi-page PDF | medium / medium | ✓ / ✗ | |
| MMLongBench-Doc (Ma et al., 2024b) | Multi-page PDF | high / high | ✓ | |
| MuRAR (Zhu et al., 2025) | Collection of web pages | low / medium | ✓ / ✗ | **Multimodal RAG** |
| M$^2$RAG (Liu et al., 2025) | Collection of web pages | high / high | ✓ / ✗ | |
| MR$^2$-Bench (Zhou et al., 2025) | Interleaved image-text | high / high | ✓ / ✗ | |
| ViDoRE v3 (Loison et al., 2026) | Collection of PDFs | high / high | ✓ / ✗ | **Document RAG** |
| DocBench (Zou et al., 2024) | Collection of PDFs | high / high | ✓ / ✗ | |
| M3DocRAG (Cho et al., 2024) | PDFs made from web pages | medium / low | ✓ / ✗ | |
| MMDocIR (Dong et al., 2025) | Collection of PDFs | high / high | ✓ / ✗ | Single-step |
| FinRAGBench-V (Zhao et al., 2025) | Collection of PDFs | low / high | ✓ | retrieval |
| VIMDoc (Kim et al., 2025) | Collection of PDFs | high / high | ✓ / ✗ | and answer |
| JINA-VDR (Günther et al., 2025) | Collection of PDFs | high / high | ✓ / ✗ | |
| UniDoc-Bench (Peng et al., 2026) | Collection of PDFs | high / high | ✓ / ✗ | |
| BRIGHT (Su et al., 2025) | Collection of web pages | high / medium | ✓ | **Agentic Research** |
| Researchy Questions (Rosset et al., 2025) | Collection of texts | high / high | ✓ / ✗ | |
| ViDoSeek (Wang et al., 2025) | Collection of slide decks | low / high | ✓ / ✗ | |
| DOUBLE-BENCH (Shen et al., 2025) | Collection of PDFs | medium / medium | ✓ / ✗ | Multi-step |
| MRMR (Zhang et al., 2025b) | Interleaved image-text | high / low | ✓ / ✗ | answering |
| **Our benchmark** | Collection of PDFs | high / high | ✓ | with tools |

ever, they rely on HTML or plain text, disregarding visual comprehension required for real-world documents.

*(ii)* **Scope.** Domain-specific PDF benchmarks such as FinRAGBench-V (Zhao et al., 2025) and ViDoSeek (Wang et al., 2025) confront visually-rich PDFs but restrict evaluation to narrow verticals like finance or rely on single-step metrics that fail to capture iterative planning and refinement.

*(iii)* **Data Provenance.** General-purpose document benchmarks attempt to bridge these gaps but suffer from methodological concerns. ViDoRE (Faysse et al., 2025b) relies on MLLM-generated questions, risking bias toward similar models. ViDoRe v3 (Loison et al., 2026) mitigates this with contextually-blind generation and extensive human verification, but the generating model still shapes the distribution of question types—a concern absent from fully human-authored benchmarks. Similarly, VIMDoc (Kim et al., 2025) and DOUBLE-BENCH (Shen et al., 2025) recycle documents from older datasets (e.g., DocVQA, Wikipedia), increasing the risk of data contamination.

We addresses all three limitations: combine the high layout diversity of a large-scale, heterogeneous PDF collection *(format)* with broad domain coverage and multi-step reasoning requirements *(scope)*, grounded in rigorous, fully human-authored questions over fresh documents *(data provenance)*.

## 1.2. Guiding Principles

MADQA exemplifies *Agentic Document Collection Visual Question Answering*. Given a corpus $\mathcal{C}$ of multi-page documents and a natural language query $q$, the task is to produce an answer $a$ and a minimal evidence set $\mathcal{E} \subseteq \mathcal{C}$ (see Appendix A.2 for full formulation). Six properties distinguish this task from standard document QA:

| # | Property | Definition |
|---|---|---|
| 1 | Extractive | Answer tokens must appear physically in the evidence set $\mathcal{E}$. |
| 2 | Multi-Hop | $\mathcal{E}$ may span disjoint pages (cross-page) or documents (cross-doc). |
| 3 | Closed-World | Answer derived solely from $\mathcal{C}$; no external parametric knowledge. |
| 4 | Grounded | $\mathcal{E}$ must entail $a$ and be minimal (no superfluous pages). |
| 5 | Agentic | No single retrieval query $q'$ may exist such that $\text{RETRIEVE}(q') \supseteq \mathcal{E}$. |
| 6 | Visual | Answering may require non-textual information (layout, tables, figures) in $\mathcal{E}$. |

Properties 1, 3, and 4 are *enforced by construction*: answers must be *extractive* (tokens appear in evidence), derived under a *closed-world* assumption, and *attributed* to minimal evidence such that evaluation rewards genuine corpus use.

Properties 2, 5, 6 are *targeted by design*: our annotation protocol encourages—but does not strictly enforce—questions requiring *multi-hop* reasoning across disjoint pages or documents, conditions where single-query retrieval is unlikely to surface all evidence (*agentic*), or where *visual* comprehension of layout structure is beneficial.

The *agentic* property, when satisfied, necessitates planning (decomposing $q$ into sub-queries), navigation (traversing a collection), and aggregation (synthesizing partial answers).

### 1.3. Contributions

**Task Formalization.** We formally define *Agentic Document Collection VQA* with six core properties that distinguish it from prior document QA formulations (§1.2).

**Validated Benchmark.** We release a fully human-authored dataset of 2,250 questions over 800 heterogeneous, fresh PDF documents (§2). Unlike prior work, we operationalize a Construct Validity framework (Bean et al., 2025) to certify the benchmark's integrity. (§2.3-2.4).

**Evaluation Protocol.** We provide principled methods of measuring answer correctness, evidence attribution, and a novel metric for effort calibration, all directly motivated by our formal properties (§3).

**Principled Split Creation.** We eploy *Classical Test Theory* to derive a test set with strong rank correlation to the full benchmark while reserving hard items for long-term relevance, enabling low-cost experimentation (§2.4).

**Human vs. Agent Comparison.** We provide the first comparative study of human vs. agentic research behaviors in this domain. Results reveal a critical *efficiency gap* and fundamentally different competencies, with failures spanning retrieval, navigation, comprehension, and refusal (§5.2).

**Efficiency Analysis.** We compare static RAG, unconstrained Recursive Language Models (RLM), and tool-augmented Agents, and demonstrate that constrained agency significantly outperforms static RAG while avoiding the catastrophic effort overhead of RLMs (§5).

## 2. Dataset Construction and Validity

To ensure high-quality, solvable, and unambiguous benchmarks, we orchestrated a rigorous human annotation pipeline involving over 1,200 hours of professional work.

### 2.1. Sourcing and Filtering Documents

We manually curated PDFs from DocumentCloud,[1] intentionally seeking clusters of up to 30 related documents (e.g., sequential reports or menus from different restaurants). It

was crucial for enabling realistic cross-document multi-hop questions (e.g., comparing values between documents).

The created corpus covers 800 PDFs divided into 63 fine-grained categories in 13 high-level domains (as detailed in Table 2), ranging from single-page summaries to extensive 800+ page filings (see Appendix B.1 for detailed statistics; Appendix A.1 provides a full Dataset Card).

**Layout and Domains Diversity.** We extract layout elements and compute per-element z-score normalization to highlight domain-specific patterns (see Appendix B.2). Figure 2 visualizes the structural heterogeneity of our corpus by showing layout element density across domains: financial and government documents exhibit high table density, while technical documents are figure-heavy. Legal documents show elevated text density with minimal visuals, and Reference materials (catalogs, guides) contain diverse lists.

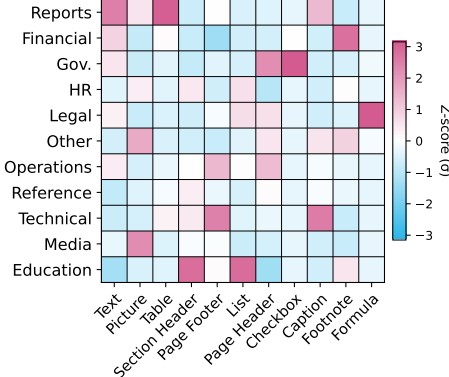

*Figure 2.* **Layout element density across document domains in MADQA.** The heatmap shows the standardized (z-scored) concentration of individual layout elements within each domain. **Pink** indicates above-average density, while **cyan** indicates below-average density. A detailed discussion is provided in Appendix B.2.

### 2.2. Question Annotation and Quality Assurance

We contracted a professional data vendor with full-time employees who were experienced in labeling QA datasets over PDF documents.

**Annotation Protocol.** We formulated strict guidelines to ensure solvability and lack of ambiguity (detailed in Appendix C.1): (1) Questions must be answerable entirely from the provided documents, strictly excluding external world knowledge. (2) Questions must be specific enough to pinpoint a unique answer but must not reveal the source location too easily. (3) For every question, collect the minimal set of evidence pages required to answer (tagging all pages contributing to the final answer for multi-hop queries).

**Annotator Selection.** We provided 20 annotators with the same guidelines to annotate the same group of documents. Their annotations were subjected to a two-step verification:

---

[1] https://www.documentcloud.org/documents/

(1) We provided GPT-5 with the human-annotated oracle evidence pages. If the model failed to answer correctly using the perfect context, the instance was flagged for manual check. (2) Only annotators with zero errors in manual review participated in MADQA annotation.

**Supervision, Checks, and Corrections.** Domain expert from the authors' institution maintained constant synchronization with contractors to resolve ambiguities in real-time and enforce strict adherence to the guidelines. After the initial version of the dataset was composed, we reviewed the questions that were not answered correctly by any of the tested baseline models or any of real humans employed to establish human baseline (§4). This led to replacing ($< 1\%$, mostly typos) or extending gold standard ($< 5\%$, mostly adding optional, desirable context).

**Annotation Statistics.** Table 2 presents the distribution of 2,250 QA pairs. We targeted approximately 20% multi-hop questions in our annotation guidelines, though without strict enforcement. The resulting distribution spread of 17.3% multi-hop, 82.7% single-hop instances closely matches this target. 8.3% require synthesizing information from multiple pages within the same document (*X-Page*), while 9.0% demand cross-document reasoning (*X-Doc*). The single-hop majority lets us detect over-searching and reflects the difficulty mix of real workflows.

*Table 2.* **Constituents of MADQA.** The corpus spans documents (median 5 pages, mean 23.3, max 859) totaling 12.2M tokens.

| Domain | Docs | Pages | Pg/Doc | Qs | Q/Doc |
|---|---|---|---|---|---|
| Financial | 131 | 6,149 | 46.9 | 460 | 3.5 |
| Reports | 127 | 2,665 | 21.0 | 360 | 2.8 |
| Gov/Regulatory | 105 | 702 | 6.7 | 304 | 2.9 |
| Legal | 69 | 1,154 | 16.7 | 182 | 2.6 |
| HR/Employment | 68 | 813 | 12.0 | 159 | 2.3 |
| Reference | 62 | 1,292 | 20.8 | 218 | 3.5 |
| Miscellaneous | 56 | 155 | 2.8 | 92 | 1.6 |
| Events | 43 | 117 | 2.7 | 88 | 2.0 |
| Financial/Tax | 39 | 2,925 | 75.0 | 82 | 2.1 |
| Media/Publishing | 31 | 1,492 | 48.1 | 113 | 3.6 |
| Technical | 29 | 842 | 29.0 | 68 | 2.3 |
| Education | 26 | 255 | 9.8 | 68 | 2.6 |
| Cases/Logs | 14 | 58 | 4.1 | 55 | 3.9 |
| **Total** | **800** | **18,619** | **23.3** | **2,250** | **2.8** |

### 2.3. Construct Validity Analysis

We measure to what extend guidelines from Section 1.2 are satisfied, explicitly dissociating the construct from confounders (Bean et al., 2025).

**Lexical Overlap vs. Reasoning.** To verify that MADQA requires *planning retrieval trajectories*, we check if gold evidence can be retrieved based on question n-grams. Unigram matching yields a median of ∼4k pages per query with only 0.03% precision—questions are drowned in false positives. Bigram matching reduces hits to ∼24 pages but

precision remains at 2.6%. Trigram matching achieves ∼1 page hit, yet recall drops to 51%. This confirms that solving our benchmark requires semantic understanding, not just lexical overlap (Appendix E.1). These findings hold beyond keyword matching: even strong dense retrieval leaves 27% of questions unsolvable in a single pass (Appendix E.2).

**Parametric Knowledge vs. Grounding.** To confirm that the benchmark measures the ability to *synthesize a faithful answer* solely from $\mathcal{C}$, we prompt six frontier models to guess answers based solely on question text. Then, we categorize correct guesses into three types: yes/no questions, other binary-choice questions, and memorization (facts recalled from the training data). Across models, measured guessability ranges from 9.1% (Claude Haiku) to 15.2% (GPT-5), with an average of 11.2%. Based on our question classification, 3% stems from random chance on yes/no and binary questions, while the remaining 8% reflects training data contamination—models correctly recalling facts from public documents.

When models achieve 80%+ accuracy with document evidence, the additional 70+ percentage points represent genuine comprehension. We intentionally retain these guessable items (Appendix E.3 gives the full argument).

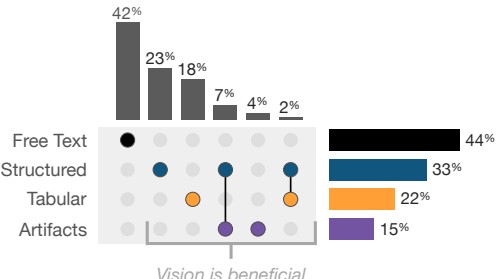

*Figure 3.* **Visual necessity in MADQA.** 58% of the questions benefit from understanding *Structured* layouts, *Tabular* data, or Visual *Artifacts* (e.g., charts, stamps). The matrix highlights that multi-category dependencies (e.g., *Structured + Artifacts*) are a significant driver of benchmark difficulty.

**Visual Perception.** To quantify visual understanding (Property 6), we categorize questions by whether answers are extractable from running text, benefit from layout comprehension (forms, tables), or require visual artifacts (checkboxes, figures). Figure 3 shows that only 42% of questions can be answered from free text alone. Structure is not the only visual challenge: 15% involve visual artifacts, with 7% requiring both—e.g., interpreting a checkbox within a form field. While our documents are text-heavy, this text is rarely unstructured, and relationships between elements must be taken into account. While tabular relationships can sometimes be inferred from linearized text, our fine-grained annotations enable measuring where visual encoders, layout-aware parsing, or pure text extraction each succeed. See

Appendix E.4 for the full taxonomy.

## 2.4. Principled Splits Creation

To reduce evaluation costs while maintaining statistical power, we apply *Classical Test Theory* (Crocker & Algina, 1986). We evaluate each questions's Difficulty and Discrimination, prioritizing questions that best distinguish between strong and weak models (Figure 4, Appendix D).

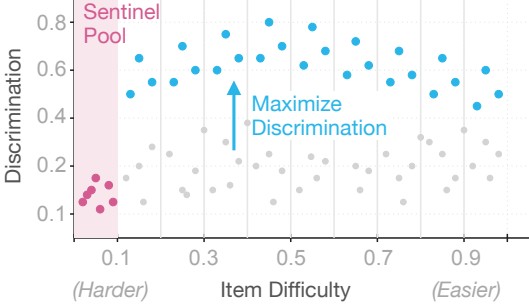

*Figure 4.* **Principled dev/test set selection.** We evaluate every question based on Difficulty (mean accuracy) and Discrimination (point-biserial correlation). The Sentinel Pool (●) captures the hardest items to preserve headroom, regardless of discrimination scores. For the remaining budget, we stratify questions into difficulty bins and greedily select those with the highest discrimination signal (●), discarding questions with lower predictive power (●).

This approach yields non-overlapping Test ($n = 500$) and Development ($n = 200$) sets. The Test set achieves a strong rank correlation with the complete benchmark (Spearman's $\rho > 0.85$) while retaining 100 items that are too complex for current models to ensure long-term relevance. The remaining $1,550$ items are provided as a Train set.

## 3. Evaluation Protocol

### 3.1. Answer Correctness (Property: *Extractive*)

We use LLM-based **Accuracy** to balance two needs: answers must be concrete values suitable for downstream automation, yet the metric should accept semantically correct responses even when they differ in surface form from ground truth.[2] By focusing on the *Extractive* property, we ensure the benchmark rewards answers grounded in the corpus, not stylistic choices like formatting or verbosity.

Answers are represented as lists of strings to accommodate multi-part responses (e.g., "list all board members"). The evaluation prompt was iteratively refined through human calibration rounds, addressing edge cases in list formatting, verbosity tolerance, and unit qualifier handling. Excluding ∼50% exact match cases where agreement is perfect by

---

[2]We initially considered ANLS* (Peer et al., 2025), but found it too strict—even after adding alternative answers, 35% of predictions where ANLS* assigned zero score were false negatives.

construction, the final setup achieves a quadratic-weighted Cohen's $\kappa = 0.88$ with human judgments, indicating almost perfect agreement (Landis & Koch, 1977).

Finally, to ensure statistical validity, we measured the calibrated judge's sensitivity and specificity on human-annotated samples, and apply bias correction to aggregate scores following Lee et al. (2026) (see Appendix F.1).

### 3.2. Retrieval and Attribution (Property: *Grounded*)

As our questions are by design unanswerable using the general knowledge, we chose the **Page F1** metric (Appendix F.4) to serve as a proxy for the *Context Relevance* component of the RAG Triad (Madzou, 2024).

It measures the overlap between the unique set of pages cited by the agent and the human-annotated minimal evidence set $\mathcal{E}$. A high score certifies that the agent successfully navigated to the precise location of the answer, penalizing both "lazy citations" (failure to cite necessary pages) and "spurious citations" (citing irrelevant pages).

We also report **Doc F1**, which relaxes the constraint to the document level. Comparing Doc F1 against Page F1 allows us to diagnose "last-mile" navigation failures, where an agent correctly identifies the relevant document but fails to pinpoint the specific page containing the evidence. We annotate at page level rather than bounding-box level; Appendix A.3 discusses this design choice.

### 3.3. Efficiency and Calibration (Property: *Agentic*)

The defining feature of the Agentic property is the ability to invest variable compute into a problem. To measure whether this investment is rational, we design a metric based on the Cumulative Difference method (Kloumann et al., 2024), treating the number of discrete steps (e.g., search iterations) as a proxy for implicit uncertainty.

Specifically, given evaluation tuples $\{(s_i, y_i)\}_{i=1}^{N}$, corresponding to different test set items, with effort $s_i \in \mathbb{N}$ and correctness $y_i \in \{0, 1\}$, we sort by nondecreasing effort via permutation $\pi$ and define mean accuracy $\bar{y} = \frac{1}{N} \sum_{i=1}^{N} y_i$. The cumulative deviation curve:

$$D_0 = 0, \quad D_k = \sum_{j=1}^{k} \left( y_{\pi(j)} - \bar{y} \right)$$

tracks how accuracy conditioned on increasing effort departs from the global mean: upward (downward) segments indicate effort ranges with above-mean (below-mean) accuracy.

We quantify the dependency between effort and accuracy using the **Kuiper** range statistic (Appendix F.5; illustrated empirically in Figure 8, and on toy example in Figure 5):

$$K = \max_{0 \leq k \leq N} D_k - \min_{0 \leq k \leq N} D_k$$

A low Kuiper score indicates stable, "effort-invariant" performance. A high score reveals poor calibration, specifically identifying regimes where the agent is expending significant budget on complex queries it ultimately fails to solve. In practice, we interpret Kuiper jointly with accuracy and grounding: low values indicate weak dependence on effort, which can reflect either well-calibrated termination or uniformly effort-invariant behavior. These regimes can be distinguished with the diagnostics proposed in Appendix F.6.

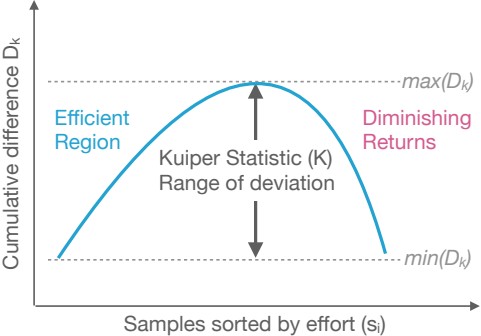

*Figure 5.* **Illustration of the cumulative difference curve.** Upward segments indicate above-average accuracy at low effort, while downward indicate below-average accuracy at higher effort. The Kuiper statistic measures the total range of this deviation.

The choice of effort measure has limited impact on calibration scores: step counts and token-based measures correlate strongly (Spearman $\rho > 0.85$), with Kuiper values varying by less than 20% across definitions (Appendix F.9). We report step counts (tool calls) by default.

## 4. Baseline Approaches

**BM25 MLLM Agent** (Textual Retrieval, Visual Reasoning). We propose an iterative system that couples text-based retrieval provided as a search tool with a MLLM, allowing the agent to formulate search queries and analyze rendered page images to answer questions (Appendix G.1).

**Claude Agent with Semtools** (CLI-Based Semantic Search). We propose a solution based on Claude Agents SDK, which operates through Unix-style tools such as *parse* (converts PDFs to MD) and *search* (performs semantic search), combined with regular bash commands, enabling flexible query reformulation and document exploration (Appendix G.4).

**Recursive Language Models** (Programmatic Context Decomposition). A task-agnostic approach which enables LLMs to programmatically examine and recursively process the input (Zhang et al., 2025a). The document collection is exposed as a variable, allowing the LLM to write code processing it with sub-LLM calls (Appendix G.5).

**MDocAgent** (Collaborative Agents). Integrates textual

and visual cues through a fixed five-stage pipeline of specialized agents. By employing parallel text and image retrieval, it coordinates General, Critical, Text, and Image agents to extract and analyze key information, which is then consolidated by a Summarizing agent (Han et al., 2025).

**Managed RAG Services** (Blackbox Retrieval). An industry-standard reference point, relying on "RAG-as-a-Service" solutions. These include *Gemini File Search* and *OpenAI Assistants File Search* described in Appendix G.2.

**M3DocRAG** (Visual Retrieval). Proposed by Cho et al. (2024) system with vision-aware retriever that encodes document pages as images, allowing it to capture visual cues (e.g., charts, layout). Results are then fed into a MLLM.

**HEAVEN** (Hybrid Visual Retrieval). Multi-vector retrieval proposed by Kim et al. (2025). Uses DSE (Ma et al., 2024a) to efficiently retrieve candidate pages, and then re-ranks candidates using ColQwen2.5 (Faysse et al., 2025a). Retrieved pages are passed to a MLLM (see Appendix G.3).

**ColBERTv2 + LLaMA** (Text-Only Late Interaction). A simple open-source baseline with late-interaction retrieval (Santhanam et al., 2022). Relevant pages are retrieved and fed to Llama 3 8B model, following Han et al. (2025) setup.

**Human Performance.** To contextualize model performance, we collected human baseline annotations on all test questions (full interface details and annotator instructions are provided in Appendix C.2). We report inter-annotator agreement on this baseline in Appendix C.3.

In the first setup, annotators used the same search engine as our *BM25 MLLM Agent* baseline. The interface logged complete interaction trajectories: every search query, page view, and timestamp. This enables direct comparison not only of accuracy but also of search efficiency between humans and LLM agents.

Additionally, we tested humans with oracle retrieval, eliminating the effect of an imperfect search tool on results, and reducing task cognitive load.

## 5. Results and Analysis

**Simple Agentic Systems Can Outperform Strong, Static RAG.** The best-performing system, *Gemini 3 Pro BM25 MLLM Agent*, achieves an accuracy of 82.2%, representing a substantial improvement over its optimized non-agentic counterpart, *Gemini 3 Pro File Search* (78.6%). This trend holds also for all models from GPT family, confirming that the iterative planning capability is beneficial in MADQA. The only exception is Gemini 2.5 Flash, which performs exceptionally well with Google's managed RAG.

**Specialized Solutions Punch Above Their Weight.** In the class of 8B parameter backbones, M3DocRAG and MDocA-

*Table 3.* **Main evaluation results on MADQA.** Agentic systems consistently outperform their static RAG counterparts, yet an 18% oracle gap reveals that retrieval—not reasoning—remains the primary bottleneck. We report aggregate Accuracy (± confidence intervals) alongside specific multi-hop reasoning subsets (X-Page and X-Doc). Attribution is measured via Page F1 and Doc F1 to assess grounding fidelity. The Kuiper statistic (↓ is better) quantifies effort calibration; it is excluded for Non-Agentic systems as they operate with fixed computational budgets.

Pink *indicates higher Kuiper values (worse; ↓ better).* Blue *indicates higher performance (better).* **Bold** *denotes the best performing model in Agentic and Non-Agentic systems for each subset.*

| Model / Framework | Accuracy | X-Page | X-Doc | Page F1 | Doc F1 | Kuiper ↓ |
|---|---|---|---|---|---|---|
| ***Non-Agentic Systems*** | | | | | | |
| Gemini 3 Pro ₍File Search₎ | **78.6** ± 2.2 | **74.1** ± 3.6 | **75.0** ± 3.6 | **70.1** ± 2.0 | **94.2** ± 1.0 | – |
| Gemini 2.5 Flash ₍File Search₎ | 71.8 ± 2.4 | 61.3 ± 4.1 | 73.0 ± 3.7 | 52.2 ± 2.2 | 80.9 ± 1.8 | – |
| M3DocRAG | 61.6 ± 2.6 | 31.0 ± 3.9 | 35.0 ± 4.0 | 68.2 ± 2.1 | 82.6 ± 1.7 | – |
| GPT-5.2 (2024-08) ₍HEAVEN₎ | 52.9 ± 2.7 | 38.9 ± 4.1 | 53.0 ± 4.2 | 48.4 ± 2.2 | 62.3 ± 2.2 | – |
| GPT-5.2 (2025-12) ₍File Search₎ | 50.0 ± 2.7 | 39.5 ± 4.1 | 23.0 ± 3.5 | 28.5 ± 2.0 | 68.5 ± 2.1 | – |
| GPT-5 (2025-08) ₍File Search₎ | 49.6 ± 2.7 | 36.4 ± 4.0 | 25.0 ± 3.6 | 29.3 ± 2.0 | 66.6 ± 2.1 | – |
| GPT-4o (2024-08) ₍HEAVEN₎ | 48.6 ± 2.7 | 32.2 ± 3.9 | 37.0 ± 4.0 | 43.2 ± 2.2 | 59.2 ± 2.2 | – |
| GPT-5 Mini (2025-08) ₍File Search₎ | 48.5 ± 2.7 | 32.8 ± 3.9 | 26.0 ± 3.7 | 29.0 ± 2.0 | 67.3 ± 2.1 | – |
| ColBERTv2 + Llama-3.1-8B | 40.2 ± 2.6 | 23.7 ± 3.5 | 26.0 ± 3.7 | 43.4 ± 2.2 | 52.0 ± 2.2 | – |
| ***Agentic Systems*** | | | | | | |
| Gemini 3 Pro ₍BM25 Agent₎ | **82.2** ± 2.0 | **66.8** ± 3.9 | 73.0 ± 3.7 | 78.5 ± 1.8 | 90.2 ± 1.3 | 25.8 |
| Claude Sonnet 4.5 (2025-09) ₍BM25 Agent₎ | 80.6 ± 2.1 | **66.8** ± 3.9 | **82.0** ± 3.2 | **79.1** ± 1.8 | **93.0** ± 1.1 | 35.1 |
| GPT-5 (2025-08) ₍BM25 Agent₎ | 77.7 ± 2.2 | 60.1 ± 4.1 | 74.0 ± 3.7 | 74.2 ± 2.0 | 86.5 ± 1.5 | 52.6 |
| Gemini 3 Pro ₍RLM₎ | 73.8 ± 2.3 | **66.8** ± 3.9 | 66.0 ± 3.9 | 69.1 ± 2.1 | 89.8 ± 1.4 | **22.9** |
| Claude Agent ₍Semtools₎ | 72.6 ± 2.4 | 62.0 ± 4.0 | 60.0 ± 4.1 | 51.1 ± 2.2 | 89.5 ± 1.4 | 37.9 |
| Claude 4.5 Sonnet (2025-09) ₍RLM₎ | 70.5 ± 2.4 | 65.0 ± 4.0 | 69.0 ± 3.9 | 66.5 ± 2.1 | 88.9 ± 1.5 | 42.3 |
| Claude Haiku 4.5 (2025-10) ₍BM25 Agent₎ | 68.2 ± 2.5 | 48.0 ± 4.2 | 65.0 ± 4.0 | 72.0 ± 2.0 | 88.2 ± 1.4 | 50.7 |
| GPT-5.2 (2025-12) ₍BM25 Agent₎ | 67.8 ± 2.5 | 51.6 ± 4.2 | 55.0 ± 4.1 | 67.6 ± 2.1 | 83.7 ± 1.7 | 64.8 |
| GPT-5 Mini (2025-08) ₍BM25 Agent₎ | 66.9 ± 2.5 | 48.0 ± 4.2 | 48.0 ± 4.2 | 67.6 ± 2.1 | 82.4 ± 1.7 | 73.2 |
| GLM-4.6V ₍BM25 Agent₎ | 66.1 ± 2.5 | 37.1 ± 4.0 | 70.0 ± 3.8 | 65.9 ± 2.1 | 86.6 ± 1.5 | 51.4 |
| GPT-5.2 (2025-12) ₍RLM₎ | 64.2 ± 2.6 | 55.3 ± 4.1 | 56.0 ± 4.1 | 67.6 ± 2.1 | 83.7 ± 1.7 | 30.0 |
| MDocAgent | 63.8 ± 2.6 | 37.1 ± 4.0 | 41.0 ± 4.1 | 67.1 ± 2.1 | 82.1 ± 1.7 | – |
| Qwen3-VL (235B-A22B-Thinking) ₍BM25 Agent₎ | 60.3 ± 2.6 | 36.4 ± 4.0 | 55.0 ± 4.1 | 58.6 ± 2.2 | 80.5 ± 1.8 | 36.6 |
| GPT-4.1 (2025-04) ₍BM25 Agent₎ | 60.0 ± 2.6 | 42.5 ± 4.1 | 44.0 ± 4.1 | 64.1 ± 2.1 | 82.8 ± 1.7 | 43.2 |
| Gemini 2.5 Flash ₍BM25 Agent₎ | 58.5 ± 2.6 | 30.4 ± 3.8 | 56.0 ± 4.1 | 61.0 ± 2.2 | 78.8 ± 1.8 | 46.5 |
| GPT-5 Nano (2025-08) ₍BM25 Agent₎ | 58.2 ± 2.6 | 32.8 ± 3.9 | 35.0 ± 4.0 | 60.9 ± 2.2 | 82.2 ± 1.7 | 49.8 |
| Qwen3-VL (8B-Thinking) ₍BM25 Agent₎ | 47.3 ± 2.7 | 17.0 ± 3.1 | 44.0 ± 4.1 | 47.6 ± 2.2 | 69.4 ± 2.1 | 50.2 |
| GLM-4.6V Flash ₍BM25 Agent₎ | 46.0 ± 2.7 | 18.2 ± 3.2 | 47.0 ± 4.2 | 28.9 ± 2.0 | 51.5 ± 2.2 | 27.5 |
| GPT-4.1 Nano (2025-04) ₍BM25 Agent₎ | 19.5 ± 2.1 | 6.1 ± 2.0 | 9.0 ± 2.4 | 27.6 ± 2.0 | 40.2 ± 2.2 | 28.6 |
| ***Human Performance*** | | | | | | |
| Human ₍Oracle Retriever₎ | **99.4** ± 0.4 | **100.0** | **98.0** ± 1.2 | – | – | – |
| Human ₍BM25 Agent₎ | 82.2 ± 2.0 | 79.6 ± 3.4 | 72.0 ± 3.7 | 79.3 ± 1.8 | 93.4 ± 1.1 | **14.6** |

gent achieve accuracy above 60%, rivaling larger commercial models, and significantly outperforming *ColBERTv2 + Llama 3* (40.2%) and *Qwen3-VL 8B BM25 MLLM Agent* (47.3%). This performance gap highlights the potential of domain-specific innovations.

**Retrieval Constraints are Essential for Cost-Effective Reasoning.** Incorporating inference cost in analysis (Figure 6) reveals that the constraints imposed by search tools and RAG pipelines are beneficial. While RLMs offer theoretical flexibility, their lack of constraints leads to inefficient information processing without performance gains. For example, the *Claude Sonnet 4.5 RLM* processed over 270 million input tokens—incurring a cost of $850—yet failed to match the accuracy of its *BM25 MLLM* counterpart.

**Agents Achieve Superior Page-Level Attribution.** While managed RAGs often achieve high Doc F1, they struggle with precise localization, as evidenced by lower Page F1 scores. Agentic systems offer better page-level attribution. High Doc F1 can mask a "last-mile" failure at the page level, justifying the diagnostic value of both.

**Calibration is a Distinct Axis from Accuracy.** Kuiper varies widely and is not monotonic in answer quality: for example, *GPT-5 BM25 Agent* reaches 77.7% accuracy but exhibits substantially worse calibration (Kuiper 52.6) than

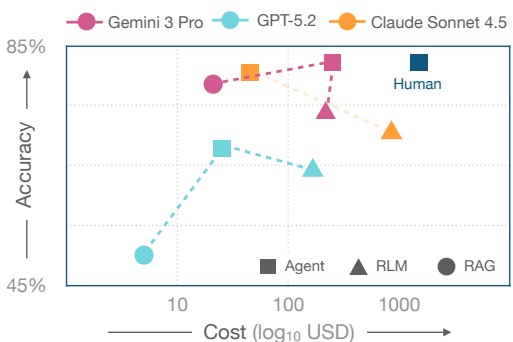

*Figure 6.* **Performance and cost of running test set inference.** Leading models with Managed RAG, compared to employing them in BM25 MLLM Agent or RLM setup.

*Gemini 3 Pro BM25 Agent* (82.2%, Kuiper 25.8).

## 5.1. Search Dynamics and Error Taxonomy

We analyze errors of the BM25 MLLM Agent family, which has the most models evaluated (Appendix H.3).

**Failure Modes Differ Qualitatively Across Models.** Among 3,273 agent errors, retrieval failures (wrong document) account for 35.7%, followed by comprehension failures (right page, wrong answer, 28.8%), navigation failures (right document, wrong page, 23.0%), and refusals (12.6%). However, these proportions vary drastically by model (Figure 13): Gemini 2.5 Pro fails predominantly at retrieval (21.4% of all predictions), while Claude Sonnet 4.5 retrieves well (4.0% retrieval failure) but struggles more with comprehension (8.6%). GPT-4.1 Nano is dominated by refusals (48.2%), indicating premature search abandonment. Detailed per-system profiles, including a retrieval-vs.-comprehension scatter (Figure 15), are in Appendix H.3.

**Query Reformulation Magnitude Predicts Success.** When an initial search fails, effective agents try a substantially *different* query rather than a minor rephrasing. We quantify this by embedding all search queries and measuring cosine drift between consecutive queries (Appendix H.6). Top-performing systems reformulate more aggressively: Claude Sonnet 4.5 has the highest mean drift per step (0.38), while GPT-4.1 Nano barely changes its queries (drift 0.10). Accuracy correlates strongly with reformulation magnitude.

**Semantic Discontinuity, Not Physical Distance, Drives Failure.** Concerning multi-hop queries, the physical "page gap" between pieces of evidence is irrelevant to performance. Difficulty is determined by *semantic distance*: accuracy drops by 38 percentage points when evidence spans conceptually dissimilar contexts (Appendix H.1).

**Errors Reflect Genuine Misunderstanding.** Only 5% of correct predictions are over-verbose concerning our specification (Section 3.1), indicating that when systems find the

right evidence, they almost always provide the requested *Extractive* answer we require. Among errors, 87.4% involve a concrete retrieval, navigation, or comprehension failure—not hallucination or refusal—confirming that the benchmark primarily tests information-seeking ability.

## 5.2. Human-Agent Comparative Analysis

**Human-Agent Performance Gap.** Despite the impressive capabilities of frontier models, humans with access to oracle retrieval perform at least 18% better. The "Oracle Gap" reveals that nearly 18% of the benchmark remains unsolved due to retrieval bottlenecks. The plateau at 80% represents a limitation of current search capability, not a saturation of the benchmark's reasoning difficulty. This is consistent with the fact that, given the same BM25 search tool, humans and Gemini 3 Pro achieve comparable performance.

**Same Accuracy, Different Competencies.** Although humans and Gemini 3 Pro both reach $\sim$82% accuracy, pairwise item agreement is remarkably low (Cohen's $\kappa = 0.24$; Figure 19 in Appendix H.4): they succeed on *different* questions. Of 107 disagreement items, 54 are solved only by humans and 53 only by the model. Human-specific failures are dominated by comprehension errors (64%), reflecting attention fatigue on complex extractions, while model-specific failures split evenly between retrieval (43%) and comprehension (43%). In contrast, model–model pairs at similar accuracy levels agree substantially more ($\kappa \approx 0.43$), sharing systematic weaknesses. This complementarity suggests that hybrid human–agent pipelines could exceed the accuracy ceiling of either alone.

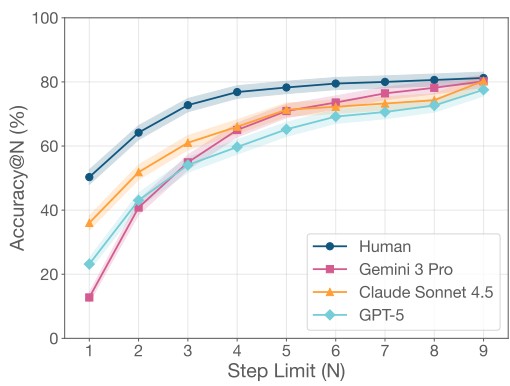

*Figure 7.* **Accuracy as a function of step limit ($N$) for BM25 MLLM agents.** While frontier agents eventually match human accuracy given a large budget ($N = 9$), they suffer from a severe "Cold Start" efficiency gap. Humans achieve 50% accuracy on their first query, whereas Gemini 3 Pro starts at only 12%, requiring an aggressive, high-cost recovery strategy to reach parity.

**Illusion of Infinite Budget.** Figure 7 plots the cumulative accuracy against the step limit $N$. While frontier agents eventually converge near the retrieval tool's theoretical ceil-

ing (∼80%), they suffer from a severe "Cold Start" disparity. Human annotators demonstrate strong zero-shot strategic calibration, achieving ∼50% accuracy on their very first query. In contrast, Gemini 3 Pro starts at only ∼12%, relying on a steep, compute-intensive recovery. The "Oracle Gap" of 17.2% combined with this cold-start inefficiency proves that MADQA remains unsolved for *efficient* agents.

**Perfect Retrieval Eliminates Human Reasoning Errors.** Humans occasionally succumbed to "negation blindness," temporal confusion, or role conflation (e.g., mistaking a signer for an approver). Detailed examples of these genuine human mistakes, distinct from annotation noise, are provided in Appendix H.2. Interestingly, these problems almost disappear entirely with perfect retrieval—when the annotator is focusing on the right place in the document.

**Humans Calibrate Effort Better.** Humans achieve a Kuiper statistic of 14.6—below every agent system, whose scores range from 22.9 (Gemini 3 Pro RLM) to 73.2 (GPT-5 Mini BM25 Agent). When initial queries fail, humans quickly change strategy. Agents, in contrast, often persist through minor query reformulations and spend compute on problems they ultimately fail to solve. The two phenomena are linked: systems with high Kuiper are precisely those with the lowest mean reformulation drift and the lowest recovery rate after an initial miss (Appendix H.6, H.5).

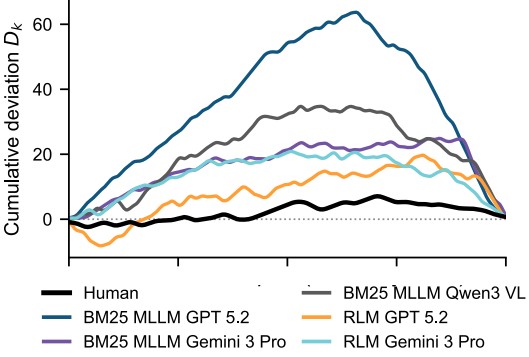

*Figure 8.* **Kuiper calibration curves** measuring effort-accuracy alignment. Human annotators remain well-calibrated across effort levels. RLM agents exhibit calibration closer to humans.

**Response Time Inversely Correlates with Accuracy.** Humans answered questions in a median time of 2 minutes (mean 3.3 minutes), with 50% of responses falling between 1–4 minutes. Response time inversely relates to accuracy: questions answered in under one minute achieve 86%, while those requiring over ten minutes—68%.

## 6. Conclusion

Even when frontier MLLM agents can answer challenging document-grounded questions, they expend substantial effort without reliably recognizing *if* and *what* additional

exploration is beneficial. Our error decomposition makes clear that this calibration gap has *multiple* sources—retrieval misses, navigation errors, comprehension mistakes, and premature refusals—rather than a single bottleneck.

To support the shift toward calibrated, efficient reasoning, we release MADQA alongside open-source implementations of our strong agentic baselines. Furthermore, Appendix I characterizes a broad design space of architectural "toggles," encouraging the benchmark's use for evaluating diverse strategies we did not explicitly consider. Our primary findings suggest two concrete directions. First, *episodic memory* could help agents learn corpus-specific terminology and document structure across queries. Second, *reinforcement learning* with search tool feedback could significantly improve exploration policies.

As the community progresses, we will adapt MADQA to target new bottlenecks, ensuring the benchmark continues to serve as a measure of frontier capabilities.

## 7. Limitations

**Coverage and Representativeness.** MADQA is predominantly U.S.-centric, so performance may not transfer to other languages or regions. Documents are sourced from public sources, which may underrepresent certain problems.

**Public Documents and Training Data Exposure.** Some content may have been observed during model pretraining; we analyze and bound this issue (Appendix E.3).

**Attribution Granularity.** We assume citations at the *page* level. This granularity supports reliable annotation at scale but limits diagnosis of intra-page grounding failures (we detail this design choice in Appendix A.3).

**Evaluation Methodology and Residual Noise.** We score correctness, attribution, and effort calibration separately rather than using a combined score. For accuracy, we rely on an LLM-as-a-judge calibrated to human judgments. It is imperfect and, in practice, errs almost exclusively by rejecting correct answers (Appendix F.2).

**Effort and its Operationalization.** The operationalization of effort as *step count* is system-dependent. Kuiper scores are most interpretable under a consistent agent/tooling regime and should be reported alongside complementary cost measures. We measure robustness to the choice of effort measure and provide supplementary diagnostics (Appendix F.9, F.6).

**Sensitivity of Ranking to Budget.** Both accuracy and Kuiper grow with the step budget $N$, and at small budgets ($N \le 3$) the effort distribution is too flat for Kuiper to be meaningful. Rankings are, by contrast, largely insensitive to the retrieval width $k$ (Appendix F.7, F.8).

## Impact Statement

This paper presents a benchmark for evaluating multimodal agents that automate document-intensive workflows. Beyond standard considerations, two impacts of our methodology deserve explicit attention.

**Calibrated effort as a quality axis.** Our headline finding is an *efficiency gap*: agents routinely enter loops of diminishing returns, expending substantial compute on queries they ultimately fail to solve. By promoting the Kuiper statistic as a first-class evaluation axis, we frame compute-efficiency as a quality property, not a deployment afterthought. Knowing *when to stop* is as much competence as knowing the answer, and we hope this benchmark steers the field toward agents that are penalized for wasteful computational flailing.

**Faithfulness and grounded attribution.** In high-stakes domains such as healthcare, law, or finance, a model that fabricates a citation can cause material harm. By scoring agents with Page F1 and Doc F1—which penalize answers not explicitly supported by the retrieved evidence—MADQA rewards systems that prioritize grounded attribution over plausible-sounding generation.

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

# Appendix

# A. Benchmark Specification

## A.1. Dataset Card

**Dataset Summary.** The Multimodal Agentic Document QA Benchmark is a comprehensive dataset designed to evaluate agentic systems, RAG pipelines, and multimodal models on their ability to perform multi-step reasoning over collections of visually rich PDF documents. This dataset enforces a closed-world assumption where systems must navigate, retrieve, and reason across multiple pages or documents (e.g., cross-referencing annual reports, legal contracts, and government forms).

*Table 4.* Summary of dataset statistics and splits.

| Metric | Value |
|---|---|
| **Total Questions** | 2,250 |
| **Total Documents** | 800 (sourced from DocumentCloud) |
| **Total Pages** | 18,619 (median 5, mean 23.3, max 859 per doc) |
| **Total Tokens** | 12.2M (assuming Qwen3 tokenizer) |
| **Multi-hop Questions** | 17.3% (8.3% cross-page, 9.0% cross-document) |
| **Visual Necessity** | 58% require layout/table/artifact understanding |
| **Domains** | 13 high-level domains (63 categories) |
| **Language** | English |

**Data Splits.** The dataset is partitioned using *Classical Test Theory* to ensure the test set retains high discrimination power while preserving hard questions, solved by none of the current models. Test: 500 samples (inputs released; labels hidden for leaderboard evaluation). Development: 200 samples (released with ground truth). Train: annotations released to facilitate RL-based optimization ($\sim$1,550 samples).

**Curation Rationale.** Existing benchmarks often lack the complexity to evaluate agentic capabilities such as planning and multi-step retrieval. This benchmark was created to bridge the gap between "chat-with-PDF" tasks and realistic document automation workflows involving heterogeneous layouts and long contexts.

**Source Data.** Documents were manually curated from DocumentCloud, specifically looking for clusters of related documents (e.g., sequential reports, invoices, legal filings) to enable cross-document reasoning. The corpus covers multiple domains, including Financial, Legal, Government, Commercial, and Personal domains.

**Annotation Process.** The benchmark required over 1,200 hours of professional annotation work. 100% of questions and answers are human-generated by trained annotators who were restricted from using external knowledge. All answers are strictly grounded in the provided PDFs. A multi-stage pipeline was employed: (1) a pilot phase where 20 candidate annotators were evaluated on identical documents, (2) automated verification using GPT-5 with oracle evidence to flag potential errors, (3) manual review by domain experts from the authors' institution, and (4) final corrections based on baseline model and human performance analysis.

**Biases and Limitations.** The majority of documents originate from the United States. The dataset is in English only. Guessability analysis shows that 9–15% of answers (depending on the model) can be correctly guessed without document access, with approximately 3% attributable to random chance on binary questions and the remainder to training data contamination from public documents.

**Personal and Sensitive Information.** The corpus references public records, which may contain personally identifiable information. The benchmark itself releases only questions, answers, and document identifiers—not the documents themselves. Users should exercise appropriate care when handling retrieved documents and follow applicable institutional guidelines for working with sensitive content.

**Licensing and Data Access.** All human-generated question-answer pairs and evidence mappings are released under the CC BY-NC 4.0 license. Additionally, the benchmark provides a curated index of documents hosted on DocumentCloud by third-party organizations (e.g., news outlets, government agencies, non-profits). The authors do not own, host, or control DocumentCloud or the referenced documents.

## A.2. Desired Properties

We formally define *Agentic Document Collection Visual Question Answering*, a task requiring systems to navigate, retrieve, reason over, and aggregate information from heterogeneous document collections.

**Definition A.1** (Document Collection). Let $\mathcal{C} = \{D_1, D_2, \ldots, D_N\}$ be a corpus of $N$ multi-page PDF documents. Each document $D_i = (p_{i,1}, p_{i,2}, \ldots, p_{i,|D_i|})$ is an ordered sequence of pages, where each page $p_{i,j}$ comprises visual content (layout, tables, figures) and textual content $\mathcal{T}(p_{i,j})$ representing its token sequence.

**Definition A.2** (Agentic Document Collection VQA). Given a corpus $\mathcal{C}$ and a natural language query $q$, the task is to produce:

1. An answer $a = (t_1, t_2, \ldots, t_k)$ as a sequence of tokens, and

2. A minimal evidence set $\mathcal{E} = \{p_{i_1,j_1}, \ldots, p_{i_m,j_m}\} \subseteq \mathcal{P}(\mathcal{C})$, where $\mathcal{P}(\mathcal{C}) = \bigcup_{i=1}^{N} \{p_{i,j}\}_{j=1}^{|D_i|}$ denotes all pages in the corpus, and the evidence set contains $m$ pages from potentially distinct documents.

The task is characterized by six formal properties that distinguish it from standard document QA:

**Desired Property 1: Extractive.** Answer tokens are drawn from the evidence pages rather than generated abstractly:

$$\forall t_\ell \in a : \exists p \in \mathcal{E} \text{ such that } t_\ell \in \mathcal{T}(p) \tag{1}$$

This permits answers spanning multiple pages or documents, but requires lexical grounding.

**Desired Property 2: Multi-Hop.** The evidence set may comprise multiple disjoint pages requiring aggregation:

$$|\mathcal{E}| \geq 1, \quad \text{with } |\mathcal{E}| > 1 \text{ for multi-hop questions} \tag{2}$$

Multi-hop questions further decompose into *cross-page* (evidence within one document: $\exists D_i : \mathcal{E} \subseteq D_i$) and *cross-document* (evidence spanning documents: $\nexists D_i : \mathcal{E} \subseteq D_i$).

**Desired Property 3: Closed-World.** The answer must be derivable solely from $\mathcal{C}$, independent of parametric world knowledge $\theta$ encoded in the model:

$$a = f(\mathcal{C}, q) \quad \text{where } f \text{ uses no knowledge from } \theta \setminus \mathcal{C} \tag{3}$$

Any answer leveraging external facts constitutes a hallucination.

**Desired Property 4: Grounded Attribution.** The answer must be faithfully attributed to the evidence set:

$$\text{ENTAILS}(\mathcal{E}, a) = \text{True} \quad \wedge \quad \mathcal{E} \text{ is minimal} \tag{4}$$

where minimality requires that no proper subset $\mathcal{E}' \subset \mathcal{E}$ suffices to derive $a$.

**Desired Property 5: Agentic.** The task requires iterative retrieval and reasoning that cannot be solved in a single forward pass:

$$\nexists \text{ query } q' : \text{RETRIEVE}(q', \mathcal{C}) \supseteq \mathcal{E} \text{ in one step} \tag{5}$$

This necessitates *planning* (decomposing $q$ into sub-queries), *navigation* (iterating retrieval based on intermediate findings), and *aggregation* (synthesizing partial answers across $\mathcal{E}$).

Unlike standard RAG benchmarks where relevant contexts are often provided or easily retrievable via lexical overlap, our formulation imposes that (1) information is often encoded in non-textual modalities or layout structures (see Figure 1), (2) single retrieval steps are insufficient, and (3) the system must function under the closed-world assumption.

**Desired Property 6: Visual.** The evidence contains information encoded in non-textual modalities—such as spatial layout, table structure, figures, or graphical elements—that is necessary for deriving $a$:

$$\nexists a' : \text{ENTAILS}(\mathcal{T}(\mathcal{E}), a') \wedge \text{EQUIV}(a', a) \qquad (6)$$

where $\mathcal{T}(\mathcal{E})$ denotes the linearized textual content (e.g., OCR output) stripped of visual structure. This property distinguishes tasks requiring genuine multimodal comprehension from those solvable via text extraction alone.

### A.3. Benchmark Design Decisions

We structure the development of this benchmark around design decisions that distinguish agentic reasoning from standard QA. Table 5 summarizes this design space, highlighting our specific choices.

*Table 5.* **Design space for agentic document benchmark.** We map the broader landscape of document evaluation choices (center) against the specific instantiation in our benchmark (right), emphasizing strict PDF-grounding, multi-hop reasoning, and efficiency-aware metrics.

| Axis | Choices in the Design Space | Our Decision in this Benchmark |
|---|---|---|
| Corpus modality & source | Single web pages; HTML dumps; synthetic PDFs; scanned documents only; born-digital only; mixture of images and text; private vs. public corpora. | Real-world, publicly hostable multi-page PDFs sourced from DocumentCloud; mixture of scanned and digital documents; benchmark definition is agnostic to whether systems use images, OCR, or layout-aware models. |
| Domain & layout diversity | Single domain (e.g., finance); few related domains; broad open-domain; homogeneous layouts (forms only); mixed narrative + tables; highly heterogeneous visual structure. | Broad coverage of real-world domains (financial, legal, government, commercial, personal) with heterogeneous layouts, as detailed in §B.1. |
| Document granularity | Single isolated document; small set of thematically related documents; large corpus/collection per query; streaming documents. | Each query is posed over a *collection* of PDFs, encouraging cross-document and cross-page reasoning rather than single-document lookup. |
| Grounding assumptions | World-knowledge allowed; mixed (doc + world); purely corpus-grounded; open-ended generation without evidence requirement. | Strictly PDF-grounded: by construction, every question is answerable using the provided documents alone; use of external world knowledge is unnecessary and treated as a source of hallucination. |
| Question source & semantics | Fully human-authored; LLM-generated; human-edited LLM generations; templated questions; questions that can be answered from a single span vs. questions requiring multi-hop reasoning. | Fully human-authored questions created directly over the PDFs, with explicit control to ensure $\sim$20% multi-hop cases (cross-page or cross-document) and strict PDF-grounding. |
| Answer format | Free-form natural language; single spans; multiple spans; sets/lists; numeric or boolean values; open-ended rationales. | Short, concrete answers: spans, booleans, lists, and numeric values. Answers are represented as small sets of strings (possibly multi-element for list questions). No rationales are required in the official metric. |
| Evidence granularity | No evidence labels; document-level only; page-level; finer-grained regions (bounding boxes, table cells, tokens); full rationale chains. | Minimal *page-level* evidence sets, including cross-document and cross-page hops. For multi-hop questions, the annotation captures the full reasoning path by tagging all pages contributing to the final answer. We deliberately chose page-level over bounding-box annotations (see §A.3). |

*Continued on next page*

*Table 5 (continued).* Design space for agentic document benchmarks and our instantiation.

| Axis | Choices in the Design Space | Our Decision in this Benchmark |
|---|---|---|
| Multi-hop definition | No explicit notion of "hop"; only single-context questions; soft multi-hop (multiple relevant snippets but not required); explicit hops with annotated paths. | Multi-hop questions explicitly require combining information across pages or documents. We annotate the minimal set of pages required to solve the question and use this to define our principled splits and multi-hop subset. |
| Difficulty control | No explicit notion of difficulty; manual tagging; heuristics (e.g., answer length); psychometric methods for subset selection; no hard items preserved. | Difficulty estimated from model responses using Classical Test Theory. We select dev/test subsets to preserve discrimination and include a dedicated pool of hard items to avoid premature saturation. |
| Answer correctness metrics | Exact-match accuracy; token-level F1; ROUGE/bleu; normalized edit distance (ANLS); LLM-as-a-judge semantic scoring; numeric tolerance-aware metrics; per-type custom metrics. | Task-optimized LLM-as-a-Judge. We found standard string-matching metrics (like ANLS*) too strict. Our judge is calibrated to maximize agreement with humans and focuses on semantic equivalence rather than surface formatting. |
| Attribution & retrieval metrics | No grounding metrics; recall@k on documents; MRR/NDCG based on retrieval logs; F1 over documents; F1 over pages; span-level overlap; learned entailment / attribution scores. | Two F1-style metrics on the final citations: *Doc F1* (did the system identify the correct documents?) and *Page F1* (did it cite the minimal set of gold pages?). |
| Faithfulness metrics | Separate textual entailment models; LLM judges checking that answers are supported; combined answer+evidence scores; human audits on subsets. | We rely on the benchmark design (PDF-only solvability) and Page F1 as a proxy for groundedness: a correct answer with low Page F1 is flagged as unfaithful. |
| Efficiency metrics | Wall-clock latency; FLOPs or energy; number of retrieved documents; number of tool calls; number of model invocations; input/output token counts. | Step Counts (number of tool calls), in our analyses supplemented by Inference Cost (USD). We penalize architectures that incur catastrophic overhead for marginal accuracy gains. |
| Calibration / process metrics | Standard ECE/Brier score using explicit confidences; risk–coverage curves; abstain rates; trajectory-level metrics (e.g., number of retries vs. success). | The *Kuiper* statistic, derived from cumulative difference curves, to quantify *effort calibration*. It measures whether an agent's decision to invest more compute (steps) correlates with success, or if it suffers from diminishing returns in long trajectories. |
| Data splits & training signal | Evaluation-only benchmark; public train/dev, hidden test; additional unlabeled corpora; RL-ready logs; challenge sets only. | Public train and dev splits with full annotations, plus a CTT-selected held-out test set served via a test server. We explicitly design the train split to support RL and prompt-search on realistic agent traces, while keeping the test set hidden. |
| Contamination & governance | No contamination checks; ad-hoc manual checks; explicit URL/document-level overlap analysis; rotating hidden tests; no leaderboard policy vs. strict disclosure. | We commit to semantic versioning of the dataset and metrics, and a hidden test server with periodic refreshes. Submissions must disclose models, training data statements, and compute budgets. |

**Rationale for Page-Level Evidence.** Unlike benchmarks with bounding-box or region-level annotations (e.g., ViDoRe v3, FinRAGBench-V), we annotate at page level. Our pilot study showed high inter-annotator agreement at this granularity, which also aligns with how humans navigate documents and how retrieval systems operate. For agentic evaluation, page-level evidence suffices to verify correct navigation without introducing annotation noise; finer localization can be delegated to downstream grounding models.

# B. Document Corpus

## B.1. Domains and Categories

The benchmark relies on real-world PDFs sourced from DocumentCloud. Unlike datasets focused solely on academic papers or financial reports, our corpus includes high variability in layout, OCR quality, and visual density. Table 6 lists the primary document categories present in the test set.

*Table 6.* **Detailed statistics per document category.** Categories are sorted by document count. The corpus exhibits high variance in document length, from single-page posters to 859-page expense document.

| Category | Docs | Pages | Mean | Med. | Max |
|---|---|---|---|---|---|
| Annual Report | 30 | 1,754 | 58.5 | 33 | 222 |
| 990 Form | 20 | 1,890 | 94.5 | 36 | 697 |
| Verdict form | 20 | 143 | 7.2 | 3 | 46 |
| News, Journal | 20 | 446 | 22.3 | 10 | 80 |
| Expenses | 20 | 914 | 45.7 | 2 | 859 |
| Lesson plan | 20 | 247 | 12.3 | 10 | 46 |
| Toxicology Report | 19 | 77 | 4.1 | 4 | 7 |
| Inspection report | 19 | 227 | 11.9 | 4 | 102 |
| Performance review | 19 | 87 | 4.6 | 4 | 9 |
| Termination letter | 18 | 50 | 2.8 | 2 | 10 |
| Financial disclosure report | 17 | 111 | 6.5 | 7 | 9 |
| Financial Statement | 16 | 794 | 49.6 | 46 | 163 |
| Form 8-K | 16 | 165 | 10.3 | 3 | 85 |
| Incident Report | 15 | 131 | 8.7 | 3 | 40 |
| Other Tax Filing | 15 | 208 | 13.9 | 6 | 70 |
| Guide | 15 | 422 | 28.1 | 20 | 124 |
| Market Report | 15 | 435 | 29.0 | 13 | 139 |
| Arrest report | 15 | 65 | 4.3 | 4 | 10 |
| Sustainability report | 15 | 595 | 39.7 | 41 | 54 |
| Search warrant | 14 | 148 | 10.6 | 8 | 27 |
| Climate action plan | 14 | 550 | 39.3 | 37 | 80 |
| Missing person poster | 14 | 15 | 1.1 | 1 | 2 |
| Meeting agenda | 14 | 44 | 3.1 | 2 | 11 |
| Form DOI | 14 | 15 | 1.1 | 1 | 2 |
| Catalog | 13 | 573 | 44.1 | 34 | 248 |
| Public records request | 13 | 23 | 1.8 | 2 | 4 |
| Contact list | 13 | 224 | 17.2 | 2 | 177 |
| Campaign Finance Report | 13 | 399 | 30.7 | 33 | 84 |
| Resume | 13 | 31 | 2.4 | 2 | 3 |
| Warrant | 12 | 125 | 10.4 | 5 | 41 |
| Notification | 12 | 25 | 2.1 | 2 | 6 |
| Fine | 11 | 88 | 8.0 | 2 | 57 |
| Restaurant | 11 | 48 | 4.4 | 3 | 12 |
| Yearbook | 11 | 1,046 | 95.1 | 69 | 248 |
| Data Sheet | 11 | 77 | 7.0 | 4 | 21 |
| Contract, Service Agreement | 11 | 76 | 6.9 | 4 | 18 |
| Audit report | 10 | 624 | 62.4 | 48 | 164 |
| Annual Firearms Discharge Report | 10 | 720 | 72.0 | 78 | 102 |
| Manual | 10 | 463 | 46.3 | 15 | 318 |
| Patent | 10 | 132 | 13.2 | 14 | 31 |
| Employee handbook | 10 | 624 | 62.4 | 61 | 103 |
| Letter | 10 | 75 | 7.5 | 6 | 23 |
| Settlement | 10 | 670 | 67.0 | 8 | 533 |
| Budget | 10 | 958 | 95.8 | 31 | 330 |
| Building permit | 10 | 115 | 11.5 | 2 | 44 |
| Invitation | 10 | 16 | 1.6 | 1 | 4 |
| Leaderboard, Ranking | 10 | 25 | 2.5 | 1 | 8 |
| Poster | 10 | 10 | 1.0 | 1 | 1 |
| Damage Report | 10 | 242 | 24.2 | 3 | 205 |
| Contest | 10 | 25 | 2.5 | 2 | 5 |
| Crop report | 10 | 218 | 21.8 | 19 | 40 |

Table 6 (continued)

| Category | Docs | Pages | Mean | Med. | Max |
|---|---|---|---|---|---|
| Biography | 10 | 30 | 3.0 | 2 | 11 |
| Guaranty | 9 | 79 | 8.8 | 7 | 22 |
| Case Log | 9 | 43 | 4.8 | 4 | 10 |
| Conference Agenda | 9 | 32 | 3.6 | 3 | 9 |
| Marriage, Birth | 9 | 10 | 1.1 | 1 | 4 |
| NDA | 9 | 54 | 6.0 | 4 | 23 |
| Job Offer | 8 | 21 | 2.6 | 2 | 7 |
| Specification | 8 | 302 | 37.8 | 12 | 139 |
| Speed camera | 6 | 13 | 2.2 | 1 | 5 |
| Diploma, Award, Certificate | 6 | 8 | 1.3 | 1 | 3 |
| Flight plan | 5 | 15 | 3.0 | 1 | 8 |
| SEC filing | 4 | 827 | 206.8 | 189 | 274 |

## B.2. Layout Element Density Heatmap

To visualize the distribution of layout elements across document categories, we construct a heatmap with per-element z-score normalization.

For each document, we use the Granite-Docling MLLM (Livathinos et al., 2025) to extract layout elements from PDF pages. The model identifies various element types including tables, figures, lists, headers, text blocks, checkboxes, and other structural components.

For each document category $c$ and element type $e$, we compute the element density as the average number of elements per page:

$$d_{c,e} = \frac{\sum_{p \in P_c} n_{p,e}}{|P_c|} \tag{7}$$

where $P_c$ is the set of all pages in category $c$, and $n_{p,e}$ is the count of element type $e$ on page $p$.

To enable meaningful comparison across element types with different baseline frequencies, we apply column-wise (per-element) z-score normalization. For each element type $e$, we compute:

$$z_{c,e} = \frac{d_{c,e} - \mu_e}{\sigma_e} \tag{8}$$

where $\mu_e = \frac{1}{|C|} \sum_{c \in C} d_{c,e}$ is the mean density across all categories, and $\sigma_e$ is the corresponding standard deviation. This normalization centers each element's distribution at zero, allowing us to identify categories with unusually high or low presence of specific elements regardless of the element's overall frequency.

The resulting z-scores are displayed using a diverging colormap where cyan indicates below-average density ($z < 0$), white indicates average density ($z \approx 0$), and pink indicates above-average density ($z > 0$). Cell annotations show the original density values $d_{c,e}$ (elements per page), while the color intensity reflects the magnitude of deviation from the element-specific mean. Element labels include the corpus-wide mean $\mu_e$ for reference.

## Category × Layout Element Density

| | Text (μ=56.9) | Picture (μ=4.5) | Section Header (μ=1.5) | Table (μ=0.76) | Page Footer (μ=0.70) | List (μ=0.40) | Page Header (μ=0.24) | Checkbox (μ=0.02) | Caption (μ=0.03) | Footnote (μ=0.01) |
|---|---|---|---|---|---|---|---|---|---|---|
| Financial disclosure report | 4.0 | 0.31 | 1.8 | 2.5 | 0 | 0.15 | 0.14 | 0 | 0 | 0.17 |
| Toxicology Report | 46 | 2.3 | 1.6 | 11 | 0.89 | 0.63 | 0.06 | 0 | 0 | 0 |
| Performance review | 43 | 6.9 | 3.0 | 0.63 | 0.08 | 1.2 | 0.01 | 0 | 0 | 0 |
| Verdict form | 69 | 0.68 | 1.3 | 0.03 | 0.53 | 1.6 | 0.51 | 0 | 0 | 0 |
| Arrest report | 223 | 13 | 0.54 | 0.15 | 0.25 | 0 | 0.06 | 0 | 0 | 0 |
| News, Journal | 44 | 3.3 | 1.7 | 0 | 0.50 | 0.04 | 0.13 | 0 | 0 | 0 |
| Letter | 47 | 0.42 | 0.33 | 0.04 | 0.25 | 0.10 | 0.63 | 0 | 0 | 0.10 |
| Expenses | 71 | 1.3 | 0.71 | 0.82 | 0.12 | 0.22 | 0.06 | 0 | 0 | 0 |
| Inspection report | 90 | 15 | 1.2 | 0.33 | 0.43 | 0.10 | 0.08 | 0 | 0.37 | 0 |
| Search warrant | 57 | 0.76 | 1.0 | 0 | 0.29 | 0.24 | 0.53 | 0.11 | 0 | 0 |
| Other Tax Filing | 181 | 1.8 | 0.23 | 0.74 | 0 | 0 | 0.05 | 0 | 0 | 0 |
| Form 8-K | 21 | 0.14 | 2.5 | 0.38 | 0.76 | 0.14 | 0.86 | 0.79 | 0 | 0 |
| Termination letter | 22 | 2.0 | 0.45 | 0 | 0.72 | 0.40 | 0.10 | 0 | 0 | 0.03 |
| Case Log | 48 | 2.5 | 1.8 | 0.85 | 0.76 | 0 | 0.30 | 0 | 0.03 | 0 |
| Incident Report | 103 | 0.61 | 0.67 | 0.79 | 0.88 | 0.03 | 0.27 | 0 | 0 | 0 |
| Meeting agenda | 50 | 0.85 | 2.7 | 0.12 | 1.2 | 1.5 | 0.79 | 0 | 0 | 0 |
| Resume | 33 | 0.39 | 3.7 | 0 | 0.16 | 1.5 | 0.06 | 0 | 0 | 0 |
| NDA | 29 | 1.6 | 1.3 | 0.06 | 0.55 | 0.77 | 0.19 | 0 | 0 | 0 |
| Fine | 19 | 1.6 | 0.69 | 0.14 | 9.0 | 0.24 | 0.52 | 0 | 0 | 0.03 |
| Lesson plan | 6.0 | 1.9 | 5.4 | 0.14 | 0.68 | 3.4 | 0 | 0 | 0 | 0 |
| Manual | 12 | 28 | 1.2 | 0.86 | 1.4 | 0.32 | 0.29 | 0 | 0.18 | 0 |
| Data Sheet | 61 | 1.6 | 2.6 | 3.1 | 1.2 | 0.30 | 0.15 | 0 | 0.07 | 0 |
| Restaurant | 56 | 1.8 | 4.0 | 0.48 | 0.44 | 0 | 0.32 | 0 | 0 | 0 |
| Contest | 7.0 | 28 | 2.2 | 0 | 0.28 | 1.2 | 0 | 0 | 0 | 0 |
| Leaderboard, Ranking | 13 | 0.72 | 0.52 | 1.8 | 0.12 | 0 | 0.16 | 0 | 0.08 | 0 |
| Contract, Service Agreement | 83 | 1.4 | 1.3 | 0.17 | 0.79 | 0.96 | 0 | 0 | 0 | 0 |
| Notification | 45 | 1.2 | 1.0 | 0 | 0.38 | 0.12 | 0.08 | 0 | 0 | 0 |

Layout Element Type

*Figure 9.* **Per-category layout element density.** Cell values indicate average elements per page; colors show z-score normalized deviation from the element-type mean. *(Continued on next page...)*

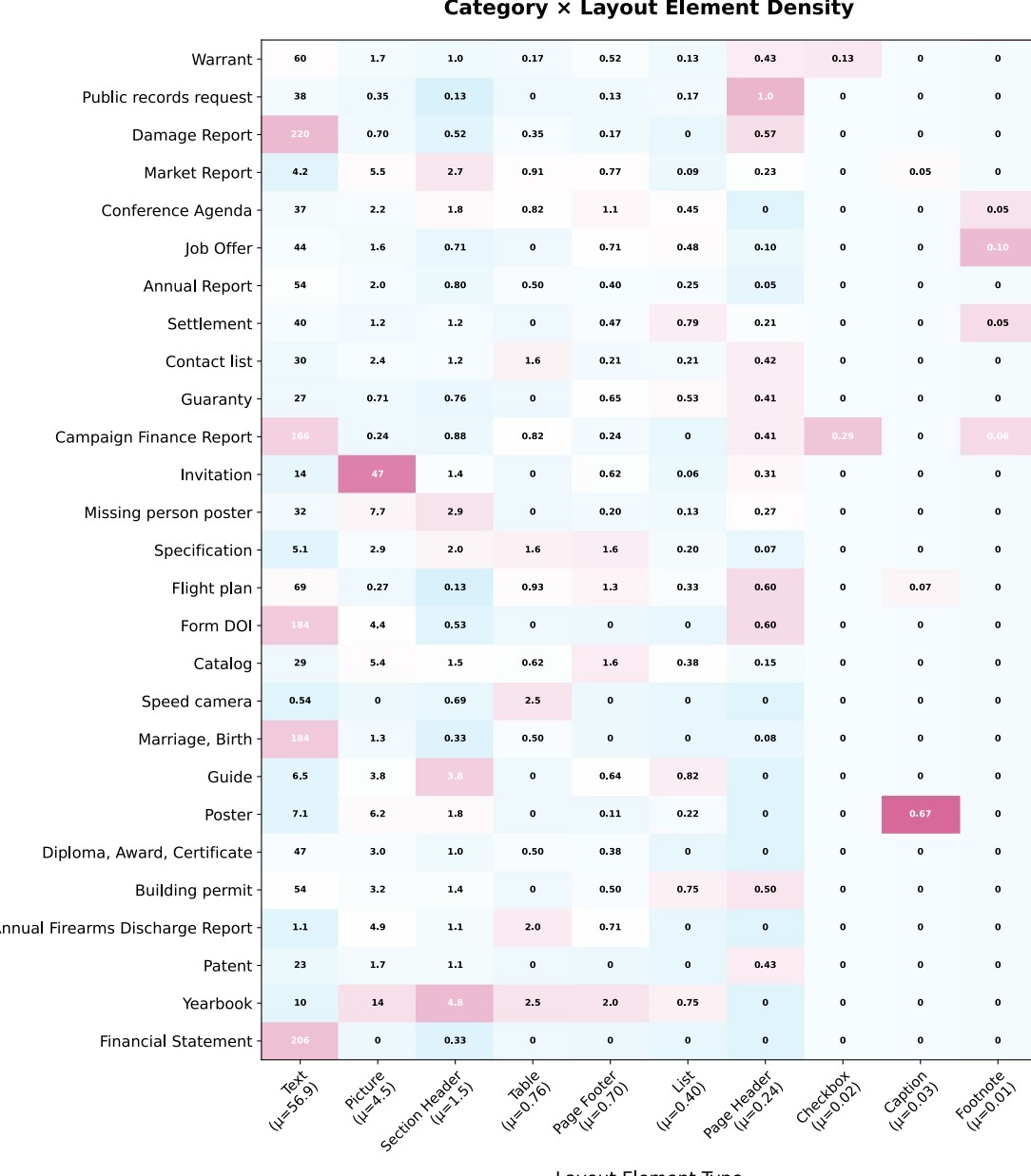

*Figure 9.* **Per-category layout element density** *(Continued).* Categories such as *Yearbook* and *News, Journal* exhibit high picture density, while *990 Form* and *Budget* show elevated table presence. Text-heavy categories like *Settlement* and *Patent* display high text block counts but minimal visual elements. This heterogeneity ensures the benchmark tests model robustness across diverse layout structures.

# C. Annotations and Human Baseline

## C.1. Annotation Guidelines

Annotators were tasked with creating Question-Answer (QA) pairs grounded strictly in the provided PDF collections. The core constraints were:

- Questions must be unanswerable without the documents (e.g., forbidding generic facts like "What is the capital of Poland?").

- Questions must be unambiguous without self-referential location cues.

    - ✗ Bad: "What is the title of *this* document?" (Ambiguous in a collection).
    - ✗ Bad: "What is the email on *page 4*?" (Trivializes retrieval).
    - ✓ Good: "What is the email address of Site Director William Wood?" (Assumes uniqueness in the corpus).

- The question must be answerable using the text or visual elements present in the PDFs.

For every question, annotators provided the *Minimal Evidence Set*, i.e., the specific file names and page numbers required to answer the question. If a question requires comparing two menus to find the "most expensive spaghetti," pages from *both* menus must be listed as evidence, even if the answer comes from only one. If evidence appears multiple times (duplication), only the most plausible or first occurrence is noted.

Annotators were instructed to ensure ∼20% of questions required *Multiple Pieces of Evidence*. These questions fall into specific reasoning categories:

1. *Bridging.* The answer to part A is required to find part B.

    Example: "What is the national bird of the nation that has a negative carbon footprint?" (Requires finding the nation with the footprint in Doc A, then finding its bird in Doc B).

2. *Comparison.* Aggregating values across documents.

    Example: "Which country has won more soccer World Cups: Argentina or Brazil?"

3. *Common Properties.* Intersection of attributes.

    Example: "Who is the only person to win an Olympic medal and a Nobel prize?"

## C.2. Human Baseline Collection

To establish a meaningful upper bound on benchmark performance and enable behavioral comparison with LLM-based agents, we collected human baseline annotations on the full test set (500 questions). Unlike typical human evaluation setups where annotators have unrestricted access to documents, our human participants used *the same retrieval interface* as our agentic baselines—a keyword-based search engine. This design choice allows us to compare not only answer accuracy but also retrieval strategies and search efficiency between humans and AI agents.

We developed a custom web application using Streamlit that provides annotators with:

- A Whoosh-based[3] BM25 search engine operating at the page level, supporting Boolean operators (AND, OR, NOT), phrase matching with quotes, and wildcard searches (Figure 10).

- An interactive PDF viewer with page navigation (first, previous, next, last, jump-to-page) allowing annotators to explore documents beyond the initial search results.

- Ability to mark specific pages as evidence supporting the answer.

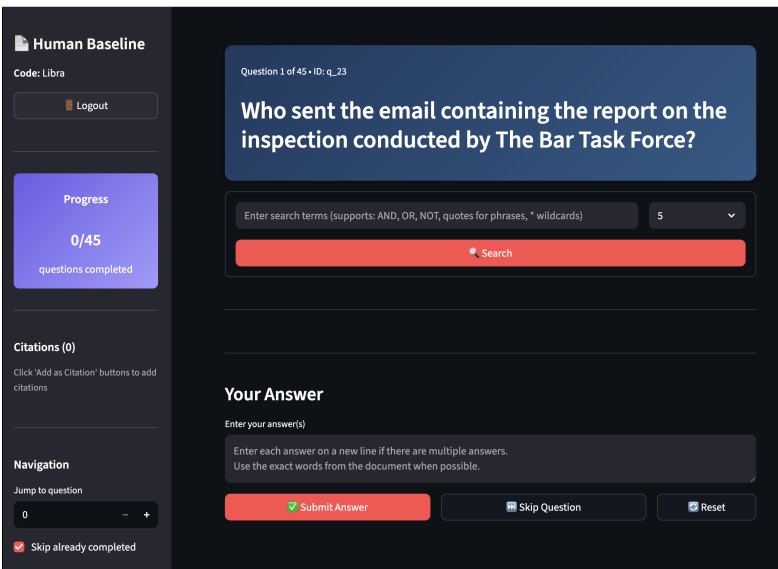

*Figure 10.* **Main annotation interface.** The question appears at the top (blue box), followed by the search bar. Search results display as interactive PDF pages with navigation controls. The sidebar shows progress, selected citations, and search history.

A key feature of our interface is comprehensive trajectory logging. For each question, we record: (1) Start and end timestamps, enabling per-question time measurement. (2) Every search query with timestamp, number of results returned, and the specific pages retrieved. (3) All document navigation actions (viewing pages, scrolling through documents) with timestamps. (4) Final evidence pages selected by the annotator.

This detailed logging enables analysis beyond accuracy, including average number of searches per question, time spent per question, retrieval precision (whether humans find evidence pages faster than agents), search query patterns and reformulation strategies.

**Annotation Protocol.** The 500 test questions were divided into 20 batches of 25 questions each. Each batch was assigned a constellation code name (Aquarius, Orion, Phoenix, etc.) for anonymized tracking. Annotators received the following instructions:

1. Read the question carefully.

2. Use the search interface to find relevant document pages. The search operates on OCR-extracted text and supports:
   - Multiple keywords: *mountain car* finds pages containing both words
   - Boolean operators: *mountain AND car*, `mountain OR car`
   - Exact phrases: *"annual report"* matches the exact phrase
   - Wildcards: *car\** matches "car", "cars", "carriage", etc.

3. Navigate within documents to find supporting evidence. Search results show individual pages, but annotators can browse the full document.

4. Mark relevant pages as citations using the "Add as Citation" button.

5. Enter the answer in the text box. For questions with multiple answers, enter each on a separate line.

The human baseline interface was intentionally designed to match the capabilities available to our agentic systems. This controlled setup enables fair comparison of retrieval effectiveness and answer accuracy between humans and LLM-based agents operating under identical constraints.

---

[3]https://github.com/whoosh-community/whoosh

### C.3. Question Distribution and Inter-Annotator Agreement

We distinguish two annotation processes, each with a different notion of inter-annotator agreement (IAA):

- **Data annotation** (§2). Annotators received documents and *created* the (Q, A, evidence) triples. Each annotator produced different questions, so traditional IAA (multiple annotators on the same item) is not computable. Quality was instead assured by (i) a pilot study with 20 candidate annotators on a shared document set; (ii) document assignment by category, so each annotator specialized in a coherent set of categories; (iii) constant supervision by a domain expert from the authors' institution; and (iv) post-hoc review by baseline models and the human evaluators ($< 1\%$ replaced, $< 5\%$ extended; §2).

- **Human baseline** (§4, Appendix C.2). Annotators received (document collection, question) pairs and produced an answer — the same task as the agentic systems. The 500 test questions were distributed equally across 5 annotators (100 each, no overlap), drawn from a different pool than the data-annotation team. IAA is meaningful here because two humans attempting the same task can be compared item-by-item.

**IAA on the human baseline.** To measure agreement on the baseline task itself, a co-author re-annotated 50 stratified questions through the identical BM25 interface, without seeing the original answers. We compare the two human responses under three independent agreement measures:

*Table 7.* **Inter-annotator agreement on the human baseline** (50 stratified questions, two independent annotators).

| Metric | Agreement |
|---|---|
| ANLS$^*$ $\geq 0.5$ | 43/50 (86.0%) |
| LLM judge (Section 3.1) | 42/50 (84.0%) |
| Manual review by a third reader | 44/50 (88.0%) |

Of the 6 disagreements identified by manual review, only 2 are genuine factual disagreements; the remaining 4 are differences in answer granularity (the same core information at different levels of detail).

These numbers measure agreement *between two humans attempting the same search-and-answer task*, not annotation quality. Humans achieve $\sim 82\%$ accuracy on the benchmark (Table 3), so the 84–88% pairwise agreement is consistent with the expected joint accuracy rate—two humans tend to make different errors in a small but non-trivial set of places.

## C.4. Sample Questions

We present a few examples to illustrate the reasoning complexity required by MADQA. Success on our benchmark requires identifying relevant documents among hundreds of distractors and switching fluidly between modalities—reading handwriting, interpreting color-coded tables, and aggregating data across disjoint files.

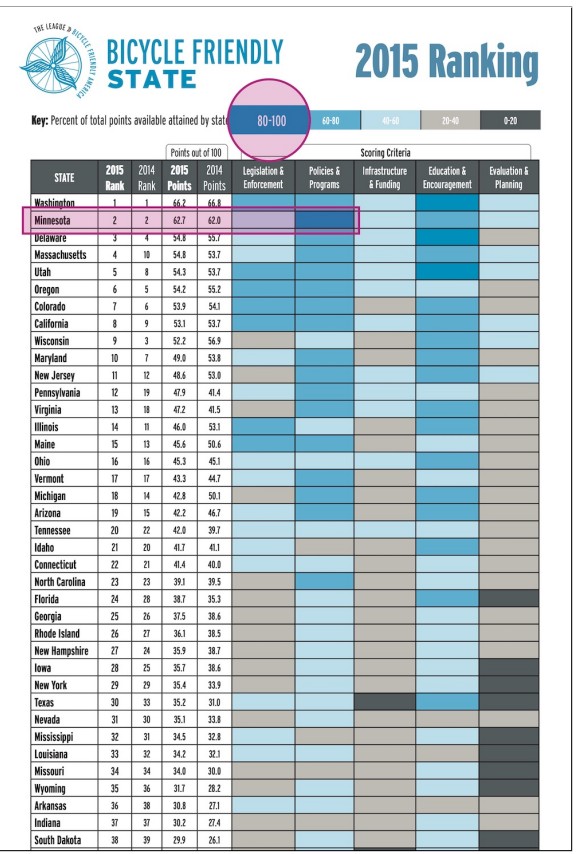

**Which state had the highest score in terms of Policies & Programs in the Bicycle Friendly State 2015 Ranking?** The agent must identify the correct "2015 Ranking" document among similar reports or yearbooks. The answer "Minnesota" cannot be found via text search because the value is encoded purely through color intensity (dark blue indicating "80-100").

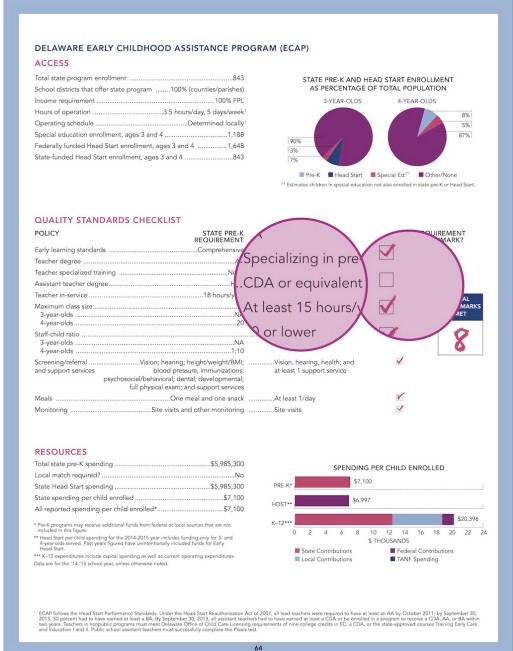

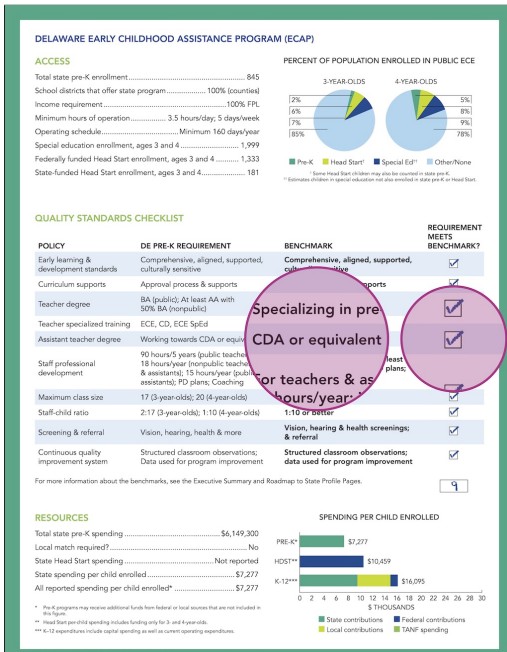

**Which quality standards benchmark(s) for preschool programs did Delaware meet in the 2019-2020 school year that it did not meet in 2014-2015?** The agent must retrieve two State Preschool Yearbook editions from different years, locate the relevant benchmarks tables in each, interpret the associated checkboxes to determine which standards were met, and compare them to identify newly met standards.

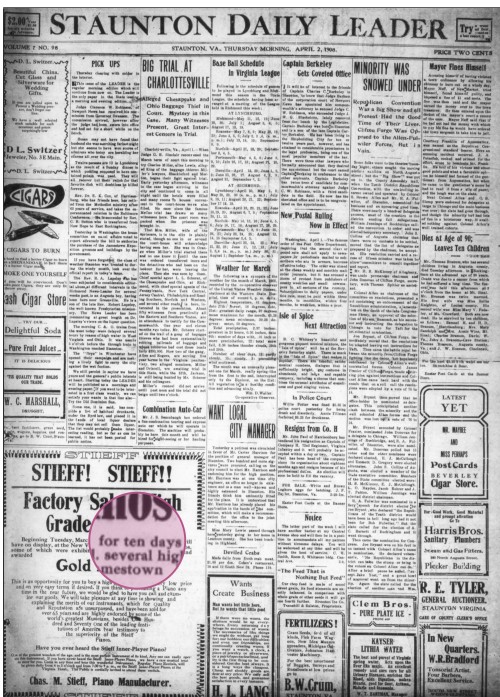

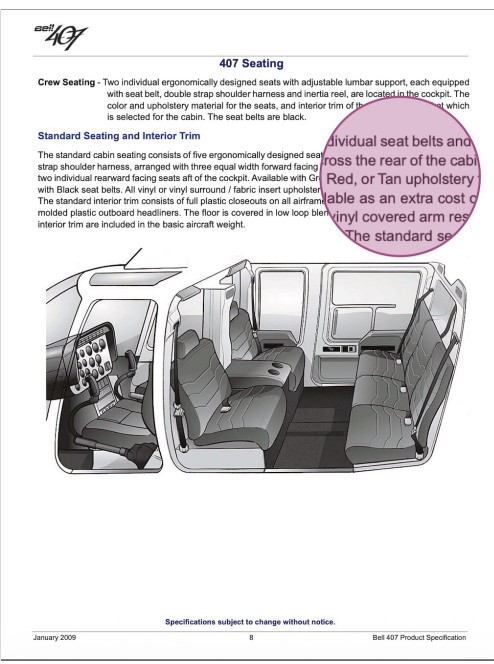

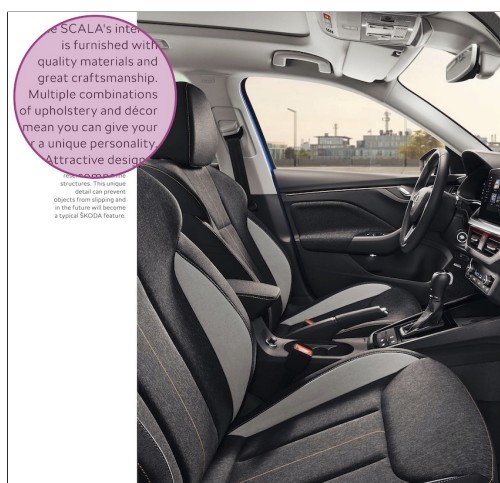

**How many days was the sale of pianos at the New Virginia Hotel planned to last?** The target information is buried within a dense, multi-column newspaper layout, likely surrounded by hundreds of unrelated articles and advertisements in the corpus. Standard chunking algorithms may fail to correctly segment the advertisement column from adjacent news stories.

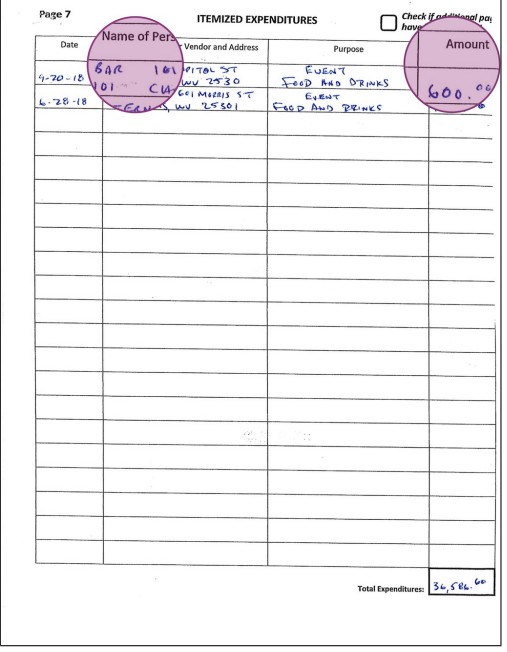

**Which vehicles offer multiple upholstery designs?** The agent must retrieve separate product brochures from a large collection and compare interior specification tables across documents to identify which vehicles list more than one upholstery option.

**How much was spent on Food and drinks at Bar 101 during the event?** The model must first determine that this information is in one of the financial statements within the collection. Once retrieved, the task involves reading handwritten entries in a table to extract the correct monetary value.

# D. Principled Splits Creation

This appendix details the Classical Test Theory (CTT) methodology used to split the complete evaluation dataset ($N = 2250$) into the Test ($N = 500$) and Development ($N = 200$) subsets (the Train set consists of questions rejected from the Development and Test sets). To ensure the subset is not biased toward a single architecture, we computed item statistics using predictions from 16 distinct models (Table 8). These models cover a diverse range of providers, sizes, and capabilities.

*Table 8.* Models used for Item Response Analysis (Sorted by Performance)

| Rank | Provider | Model Name | Model ID | ANLS* |
|------|----------|------------|----------|-------|
| 1 | Google | Gemini 3 Pro Preview | `gemini-3-pro-preview` | 0.688 |
| 2 | Anthropic | Claude 4.5 Sonnet | `claude-sonnet-4-5-20250929` | 0.681 |
| 3 | OpenAI | GPT-5 | `gpt-5-2025-08-07` | 0.647 |
| 4 | Zhipu AI | GLM-4.6V | `zai-org/GLM-4.6V` | 0.616 |
| 5 | Anthropic | Claude 4.5 Haiku | `claude-haiku-4-5-20251001` | 0.601 |
| 6 | Alibaba | Qwen3-VL-235B Thinking | `Qwen/Qwen3-VL-235B-A22B-Thinking` | 0.598 |
| 7 | Alibaba | Qwen3-VL-32B Thinking | `Qwen/Qwen3-VL-32B-Thinking` | 0.594 |
| 8 | Google | Gemini 2.5 Pro | `gemini-2.5-pro` | 0.590 |
| 9 | Google | Gemini 2.5 Flash | `gemini-2.5-flash` | 0.576 |
| 10 | OpenAI | GPT-5 Mini | `gpt-5-mini-2025-08-07` | 0.550 |
| 11 | OpenAI | GPT-5 Nano | `gpt-5-nano-2025-08-07` | 0.548 |
| 12 | Alibaba | Qwen3-VL-8B Thinking | `Qwen/Qwen3-VL-8B-Thinking` | 0.530 |
| 13 | Alibaba | Qwen3-VL-235B Instruct | `Qwen/Qwen3-VL-235B-A22B-Instruct` | 0.516 |
| 14 | Zhipu AI | GLM-4.6V Flash | `zai-org/GLM-4.6V-Flash` | 0.444 |
| 15 | OpenAI | GPT-4.1 Nano | `gpt-4.1-nano-2025-04-14` | 0.302 |
| 16 | Alibaba | Qwen3-VL-32B Instruct | `Qwen/Qwen3-VL-32B-Instruct` | 0.300 |

For each item $j$ in the dataset, we calculate two key metrics:

- **Item Difficulty** ($p_j$) Defined as the mean accuracy of all models on item $j$. A lower $p$-value indicates a harder question: $p_j = \frac{1}{M} \sum_{i=1}^{M} X_{ij}$. (Note that "$p$-value" is used here to denote is distinct from the statistical $p$-value used in hypothesis testing to measure significance. We are consistent with how Crocker & Algina (1986) uses the term.)

- **Item Discrimination** ($r_{pbis}$) We use the point-biserial correlation to measure how well item $j$ distinguishes between strong and weak models. To prevent autocorrelation, we use the *corrected* total score, subtracting the item's own score from the total sum:$r_{pbis,j} = \mathrm{Corr}(X_{\cdot,j}, S_{-j})$where $S_{-j} = \sum_{k \neq j} X_{\cdot,k}$.

Our selection strategy balances the need for high signal (high discrimination) with the need for future headroom (hard items). We select items with $p \leq 0.1$ (solved by $\leq 10\%$ of models) to form the Sentinel Pool. These items typically have low variance (and thus low discrimination metrics) but are crucial for measuring frontier capability. The remaining budget is filled by items with $p > 0.1$. We stratify these items into 9 uniform difficulty bins (e.g., $0.1 < p \leq 0.2$, $0.2 < p \leq 0.3$, etc.). Within each bin, we greedily select items with the highest discrimination ($r_{pbis}$) to maximize the subset's predictive power.

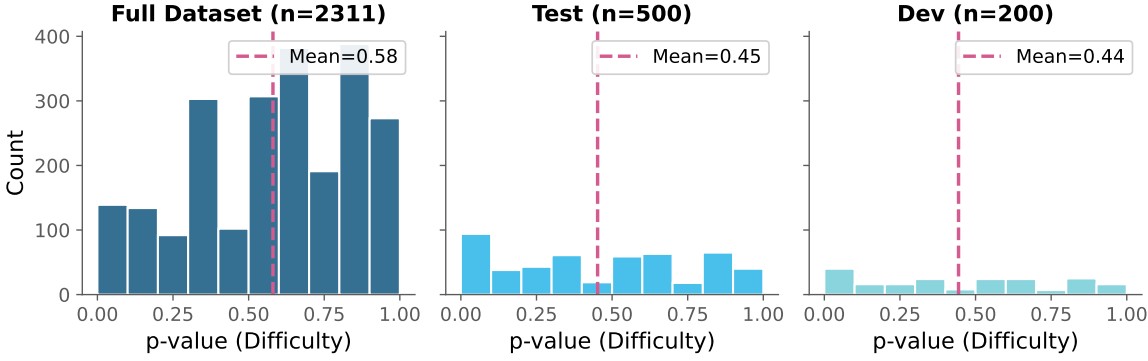

*Figure 11.* **Distribution of item difficulties** ($p$-values) in the Full Dataset vs. the Selected Test Subset. The selection strategy boosts the density of "Hard" and "Sentinel" items to preventing saturation.

# E. Construct Validity Analyses

## E.1. Lexical Overlap vs. Reasoning

To verify that MADQA cannot be solved through simple keyword matching (i.e., "Ctrl+F" strategies), we conducted a systematic n-gram analysis across the entire corpus.

For each question, we extract n-grams ($n \in \{1, 2, 3\}$) from the question text after removing stopwords and common question words (e.g., "what," "which," "how," "describe"). We then search the full corpus (18,619 pages) for pages containing these n-grams and measure: (1) *Pages matched.* How many corpus pages contain at least one question n-gram. (2) *Precision.* What fraction of matched pages are gold evidence pages. (3) *Recall.* What fraction of gold evidence pages are retrieved.

*Table 9.* **Lexical solvability analysis.** Unigrams and bigrams match too many pages (low precision), while trigrams miss half the questions entirely (low recall). No n-gram size enables effective Ctrl+F-based retrieval.

| N-gram | Median Pages | Precision | Recall |
|---|---|---|---|
| Unigram | 4,125 | 0.03% | 99% |
| Bigram | 24 | 2.6% | 86% |
| Trigram | 1 | 1.9% | 51% |

This vocabulary gap between questions and evidence confirms that solving MADQA requires semantic understanding to bridge the lexical mismatch, not just pattern matching. An effective system must reason about what information is needed and formulate appropriate search strategies—the core capability our benchmark is designed to evaluate.

## E.2. Semantic-Retrieval Upper Bound

The lexical-overlap analysis (§E.1) shows that no n-gram strategy can recover all evidence in a single pass. A natural follow-up question is whether modern *dense* retrievers eliminate the need for iteration. We answer this empirically.

**Setup.** We embed every corpus page with *Qwen3-Embedding* (4B and 8B variants) and, for each question, rank pages by cosine similarity against the question text. For a given $k$, the question is *single-pass solvable* if the gold evidence set $\mathcal{G}$ is contained in the top-$k$ retrieved pages.

*Table 10.* **Single-pass recall of dense retrieval** on the full 2,250 questions, broken down by hop type. Even with top-10 retrieval, $\sim$27% of questions cannot be solved without iteration; the gap grows to 45–49% for cross-page multi-hop.

| Setting | Overall | Single-hop | X-Page | X-Doc |
|---|---|---|---|---|
| Recall@5  (4B) | 65.0% | 73.6% | 39.0% | 47.1% |
| Recall@5  (8B) | 65.0% | 72.5% | 37.8% | 56.9% |
| Recall@10 (4B) | 73.2% | 79.8% | 51.2% | 62.7% |
| Recall@10 (8B) | 72.6% | 78.1% | 54.9% | 62.7% |

**Findings.** At Recall@5, 35% of all questions are unsolvable in a single pass, rising to $\sim$61% for cross-page multi-hop. At Recall@10, 27% remain unsolvable. These numbers represent an *upper bound* on single-pass agent performance: even an oracle that always commits to the top-$k$ dense retrieval cannot answer the missed questions without further iteration. Iterative agents can also exceed this ceiling in practice—additional steps allow them to refine their hypothesis, convert a refusal into a grounded answer, or recover after a missed first query (cf. §H.5). This confirms that the *Agentic* property (§1.2) is non-vacuous even when measured against modern semantic retrieval.

## E.3. Parametric Knowledge vs. Grounding

Before attributing model performance to document understanding, we must establish what fraction of answers can be "guessed" without any document evidence. This baseline captures two phenomena: (1) random chance on constrained answer types, and (2) training data contamination where models have memorized facts from public documents seen during pre-training.

We prompted six frontier models to answer all test questions *without* showing any document text or images (Prompt 1). The prompt explicitly instructed models to guess even when uncertain, selecting randomly for yes/no or binary choice questions. We then classified each correctly guessed answer into three categories using GPT-5-mini (Prompt 2):

---

**Guessing Prompt**

You are participating in an answer prediction experiment. You will be given a question about a document, but you will NOT see the document itself. **Your task:** PREDICT or GUESS what the answer might be, based solely on the question text.

**Critical Rules**

1. **You MUST always provide an answer** – never refuse or say you cannot answer.

2. **For yes/no questions:** Pick one randomly (Yes or No).

3. **For specific values (dates, names, numbers):** Make your best guess based on common patterns.

4. **For multiple formats:** Guess the most likely format.

5. **For list-type questions:** Provide 1–3 plausible items.

**Output Format**

- Return a **list of values**.

- If there is a single answer, return a one-element list.

- If there are multiple items/entities, return multiple elements.

- Use as few words as possible.

- **Do NOT explain your reasoning** – just provide the answer.

**Remember** This is a guessing experiment. There is no "wrong" answer - we want to see what predictions are possible from the question alone.

*Prompt 1.* **Guessing prompt** for parametric knowledge analysis. Models are instructed to predict answers without document evidence to measure training data contamination.

- *Yes/No.* Questions with binary yes/no answers (50% random baseline)

- *Other Binary Choice.* Questions explicitly listing two options, e.g., "Which is larger: A or B?" (50% random baseline)

- *Memorization.* All other questions requiring recall of specific facts, names, dates, or values

As shown in Table 11, approximately 3% of questions would be answered correctly by chance alone, stemming from 50% accuracy on yes/no and binary choice questions. However, the actual correct rate on these question types exceeds 50%, indicating that models also *know* many yes/no and binary answers from training data.

The "Memorized" column shows questions requiring specific factual recall (names, dates, values). However, total contamination is higher: some yes/no and binary questions are also answered from knowledge, not chance. We estimate total knowledge-based answering at approximately 8% (total guessability minus random baseline), with GPT-5 exhibiting the highest contamination at 11.8% (15.2% total − 3.4% random).

The 9–15% guessability (varying by model) represents an upper bound on "free" performance—answers obtainable without document understanding. Decomposing this: ∼3% from random chance on constrained answer types, and ∼8% from training data contamination. When models achieve 85%+ accuracy with document evidence, the additional 70+ percentage points represent genuine document comprehension beyond prior knowledge.

**Examples of Memorized Facts.** The following questions were answered correctly by GPT-5 purely from training data:

- "The 17,082 metric tons of SO2 emissions reported by TVA in 2020 were 99% below the peak level from which year?" → 1977

---

**Question Classification Prompt**

You are classifying questions for an analysis of answer guessability.

**Categories** Classify each question into **ONE** of these categories:

1. **"yes_no"** – Questions that have Yes or No as the answer.
   - *Examples:* "Does X offer Y?", "Is X greater than Y?", "Did X happen?"

2. **"binary_choice"** – Questions asking to choose between exactly 2 options mentioned in the question.
   - *Examples:* "Which is larger: A or B?", "Is it X or Y?", "Did they choose option A or option B?"
   - *Note:* The two options must be explicitly stated in the question text.

3. **"other"** – Everything else.
   - Questions about specific facts, names, dates, numbers, values.
   - Questions about people, organizations, locations.
   - Questions asking "who", "what", "when", "where", "how many", etc.
   - Any question that would require looking up specific information.

**Thinking Process** Think step by step:

1. Does the answer have to be Yes or No? → **"yes_no"**

2. Does the question explicitly list 2 choices to pick from? → **"binary_choice"**

3. Otherwise → **"other"**

**Input Task** Question: {`question`}

Gold answer: {`answer`}

**Classify this question:**

---

*Prompt 2.* **Question classification prompt.** Used to categorize correctly guessed answers into yes/no, binary choice, or memorization to distinguish random chance from training data contamination.

- "In what year was Chicken Annie's Original established?" → 1934

- "Who was the Special Inspector General for Afghanistan Reconstruction in 2014?" → John F. Sopko

- "What is the fee that the U.S. Department of Treasury charges for debt collection?" → 30%

- "What does MOTCA stand for (Afghan Ministries)?" → Ministry of Transport and Civil Aviation

These examples demonstrate that public documents (government reports, court filings, annual reports) in our benchmark were likely indexed on the web and incorporated into model training data.

**Why we retain the guessable questions.** A natural alternative is to filter out the ∼11% of guessable questions. We choose to keep them, for three reasons:

1. **Memorisation is model-dependent.** Pairwise Jaccard overlap of correctly-guessed question sets across the six models in Table 11 ranges from 0.18 to 0.50, and only $4/500$ test questions ($0.8\%$) are memorised by all six models simultaneously. Each new model release memorises a different subset, so removing "memorised" questions today would shift the benchmark with every future evaluation campaign.

2. **The 8% is an upper bound under an unrealistic prompt.** The numbers in Table 11 are obtained under a guessing setup that explicitly instructs the model to answer without sources. In the standard evaluation setup, agents are instructed to *search* and *cite* sources, which discourages reliance on parametric memory.

*Table 11.* **Answer guessability analysis across frontier models.** "Correct" shows questions answered correctly without evidence. Categories show the breakdown of correct guesses by question type. "Random" estimates the expected contribution from 50% chance on yes/no and binary questions; "Contam." shows pure memorization questions (facts, names, values) as percentage of test set.

| Model | Correct | Yes/No | Binary | Memorized | Random | Contam. |
|---|---|---|---|---|---|---|
| Claude Haiku 4.5 | 45 (9.1%) | 16 (35.6%) | 16 (35.6%) | 13 (28.9%) | 3.2% | 2.6% |
| Claude Sonnet 4.5 | 53 (10.7%) | 16 (30.2%) | 13 (24.5%) | 24 (45.3%) | 2.9% | 4.8% |
| Gemini 2.5 Flash | 48 (9.7%) | 13 (27.1%) | 14 (29.2%) | 21 (43.8%) | 2.7% | 4.2% |
| Gemini 2.5 Pro | 65 (13.1%) | 15 (23.1%) | 17 (26.2%) | 33 (50.8%) | 3.2% | 6.7% |
| GPT-5 | 75 (15.2%) | 19 (25.3%) | 15 (20.0%) | 41 (54.7%) | 3.4% | 8.3% |
| GPT-5 Mini | 47 (9.5%) | 14 (29.8%) | 13 (27.7%) | 20 (42.6%) | 2.7% | 4.0% |
| **Average** | 55.5 (11.2%) | — | — | — | 3.0% | 5.1% |

3. **Attribution metrics penalise ungrounded answers.** Page F1 and Doc F1 score 0 when a system answers from memory without citing the gold evidence page. Reporting accuracy jointly with attribution prevents purely-memorised answers from inflating the headline number.

Together, these arguments support retaining the items and reporting the memorization bound transparently, rather than removing a model-dependent subset that would compromise comparability across future evaluation rounds.

### E.4. Visual Perception

To validate Property 6 (Visual) and understand what types of visual comprehension our benchmark requires, we developed a taxonomy for classifying questions based on whether visual modality is beneficial for answering them, assuming the correct evidence pages have been retrieved. Our taxonomy comprises five primary categories, each representing a distinct type of visual requirement:

1. *Free Text.* Answer is present in free-flowing narrative text. Clean OCR extraction would be sufficient; no table, form, or layout comprehension required.

2. *Tabular.* Requires understanding tabular relationships—correlating rows with columns, comparing values across rows, or aggregating data from multiple cells. The key requirement is comprehending *table relationships*, not merely reading structured text.

3. *Structured.* Requires navigating structured forms (permits, licenses, certificates, applications) to locate specific fields. Distinguished from tables by the label-value pair structure rather than row-column correlation.

4. *Other Spatial.* Requires understanding spatial positioning on the page (e.g., "at the bottom of the page," "in the header"). Applied only when position matters beyond what table/form structure captures.

5. *Artifacts.* Visual artifacts not captured above, including: handwritten text, signatures, checkbox/tick mark status, stamps and seals, charts and diagrams, figures and images.

We employed *Gemini 3 Flash* to classify each question (Prompt 3), providing the model with the question text, gold-standard answer, and rendered images of all gold evidence pages. The prompt instructed the model to determine which visual modalities would be beneficial for a human or AI to answer the question correctly. Each question receives a primary category and may receive secondary categories when multiple visual modalities are genuinely required (e.g., a question requiring both table comprehension and reading a handwritten annotation)

Table 12 presents the distribution across our taxonomy. Notably, fewer than half (42.8%) of questions can be answered from unstructured free-text alone, while the majority require some form of visual or structural comprehension.

While most questions (87.7%) require only a single visual modality, 12.3% benefit from multiple. The most common combination is *Structured + Other Visual* (7.0%), typically occurring when forms contain handwritten entries or checkbox fields. Only 1.5% of questions require both *Tabular* and *Structured*, confirming that tables and forms represent distinct visual challenges.

These findings have several implications for system design:

*Table 12.* Visual modality requirements for answering MADQA questions. Categories are mutually exclusive for primary assignment; some questions have secondary categories when multiple modalities apply.

| Category | Count | % |
|---|---|---|
| Free Text | 959 | 42.8 |
| Structured | 644 | 28.8 |
| Tabular | 471 | 21.0 |
| Artifacts | 150 | 6.7 |
| *(of which: charts/diagrams)* | *41* | *1.8* |
| *(handwriting/signatures)* | *92* | *4.1* |
| *(images/photos)* | *6* | *0.3* |
| Other Spatial | 16 | 0.7 |
| **Requires Visual/Structure** | **1,281** | **57.2** |

- *Structure is pervasive.* Over 57% of questions require understanding document structure, yet this structure is often *implicit* in the visual layout rather than explicit in the text.

- *Visual encoders are not strictly necessary.* PDF-to-markdown conversion (e.g., via document understanding models) represents a viable alternative to end-to-end visual encoding, though with potential information loss. Our benchmark enables measuring these trade-offs.

- *Text is structured, not free-form.* While documents are highly text-intensive, the text is rarely unstructured. Models must either explicitly comprehend visual layout or successfully infer structural relationships from linearized text—a capability that varies significantly across architectures.

- *Retrieval vs. reading.* This analysis isolates the *reading* component by providing gold evidence pages. The impact of visual capabilities on *retrieval* accuracy represents a complementary research direction, as visual document representations may help identify relevant pages that text-based retrieval misses.

This taxonomy and analysis complement the text-vs-image ablation study (Figure 3), which measures *performance* differences, by providing a *structural* understanding of why visual modality matters for specific question types.

**Question Modality Classifier Prompt**

You are an expert at analyzing document VQA (Visual Question Answering) questions to determine whether visual modality is required or beneficial for answering them.

**Category Definitions** Classify the question into one or more categories based on these precise definitions:

1. **FREE_TEXT** – Answer is present in free-flowing text.

   - Answer is extracted directly from running/narrative text.
   - No reference to tables, charts, forms, or visual elements.
   - No need to understand document layout or spatial relationships.
   - Linear reading comprehension; clean OCR text would be sufficient.

2. **TABLE_STRUCTURE** – Requires understanding TABLE relationships (rows/columns).

   - Answer requires correlating rows and columns in a TABLE.
   - Need to understand which values belong to which headers.
   - Requires comparing multiple rows (finding max/min, aggregating data).
   - **NOT for forms:** Forms go to STRUCTURED_DATA_EXTRACTION.

3. **CHART_DIAGRAM** – Requires visual chart/diagram interpretation.

   - Data encoded visually: bar heights, line trends, pie slices, diagrams.
   - Requires counting visual elements or extracting values from visual scales.
   - Visual relationships not captured in text.

4. **TEXT_FORMATTING** – Requires seeing text formatting ONLY.

   - Identifying bold, italic, underlined, colored, or highlighted text.
   - Font size or style differences (headings vs body).
   - **NOT for:** handwriting, signatures, checkboxes → use OTHER_VISUAL.

5. **IMAGE_PHOTO** – Requires actual image/photo content.

   - Asks about photo descriptions, captions, or content visible in photos.
   - References specific images/figures containing visual information.

6. **SPATIAL_LAYOUT** – Requires spatial/layout understanding NOT covered by others.

   - **ONLY use if:** Location matters (top/bottom, left/right, margins, header/footer).
   - **Special cases only:** "at the bottom of page", "in the sidebar".
   - **NOT for tables/forms:** Use TABLE_STRUCTURE or STRUCTURED_DATA...

7. **STRUCTURED_DATA_EXTRACTION** – Extraction from structured FORMS.

   - Source is a structured FORM (permit, license, application, financial form).
   - Visual structure helps navigate the form and find the right field.
   - Answer is typically a specific value from ONE form field (label:value pairs).
   - **NOT for tables:** If comparing rows/columns, use TABLE_STRUCTURE.

8. **OTHER_VISUAL** – Visual artifacts not covered above.

   - Handwritten text, signatures, checkboxes/tick marks.
   - Stamps, seals, logos, mathematical notation, special symbols.

*Prompt 3.* **Question modality classifier prompt.** Classifies questions by visual requirements to quantify the importance of layout understanding, table comprehension, and visual artifacts.

**Input** Question: {question}

Answer: {answer} **The gold evidence page images are provided below.** Examine them to determine whether visual modality is needed to answer the question.

**Guidance & Priority** **Multi-label Guidance:**
- **Most questions should have ONLY ONE category** (the primary visual requirement).
- Only assign multiple categories if the question TRULY requires combining different modalities.
- **Table vs Form:** TABLE_STRUCTURE + STRUCTURED_DATA is extremely rare (requires both correlating rows AND extracting form fields).
- **Spatial:** Only use if position matters AND it is not covered by table/form structure.

**Selection Priority:**
1. If it's a table with rows/columns to correlate → **TABLE_STRUCTURE**.
2. If it's a form with label:value pairs → **STRUCTURED_DATA_EXTRACTION**.
3. If spatial position matters beyond structure → add **SPATIAL_LAYOUT** (rare).

**Instructions**
- Judge based on whether visual modality would actually help answer the question.
- Choose the ONE primary visual requirement in most cases.
- Do not rely on keyword presence - analyze the actual task.
- Consider whether clean OCR text would be sufficient.

**Output Schema** Respond with a JSON object:

```
{
  "primary_category": "the single most important category",
  "all_categories": ["list", "of", "ALL", "applicable", "categories"],
  "confidence": 0.0-1.0,
  "reasoning": "brief explanation covering all assigned categories"
}
```

*Prompt 3.* **Question modality classifier prompt.** *(Continued)*

# F. Formal Definition of Metrics

## F.1. Accuracy Metric

We evaluate answer accuracy using a hybrid metric that combines string-based matching with LLM-based semantic judgment, designed to be both strict on factual correctness and flexible regarding valid textual variations.

When answer yields no exact match in any of the alternative gold standard answers, we invoke an LLM judge to assess semantic correctness (Prompt 4). Following the G-Eval framework (Liu et al., 2023b), the judge performs chain-of-thought evaluation across five criteria:

1. *Refusal detection.* Does the answer refuse or claim inability to respond?

2. *Content matching.* Does the core meaning match any ground truth variant?

3. *Critical errors.* Missing unit qualifiers, wrong entities, or type mismatches?

4. *Format issues.* List structure presented as comma-separated string?

5. *Verbosity.* Excessive explanation beyond direct answer?

The judge outputs a three-way classification: *Correct* (score = 1.0), semantically equivalent to ground truth; *Partial* (score = 0.5), correct content with format/verbosity issues; *Incorrect* (score = 0.0), wrong content, missing answer, or critical errors. We use Gemini 2.5 Flash with function calling to ensure structured output.

**Handling Multiple Valid Answers.** Questions may have multiple valid ground truth representations. Let $\mathcal{G} = \{G^{(1)}, \ldots, G^{(k)}\}$ be valid alternatives. The final score is: $\text{Score}(P) = \max_k \left( \text{Metric}(P, G^{(k)}) \right)$ where Metric applies our two-stage evaluation pipeline.

The evaluation prompt was developed iteratively: starting from a vanilla LLM judge, we conducted multiple rounds of human review on 100 stratified samples, identifying systematic disagreements and refining the prompt to address edge cases—particularly around list formatting, acceptable verbosity levels, and unit qualifier handling. This calibration process improved human-LLM agreement from 82% to 90% on held-out samples before final specificity/sensitivity measurement.

**Bias Correction.** Following Lee et al. (2026), we apply Rogan-Gladen correction to adjust for LLM judge bias. Based on a 200-sample human evaluation, we measured sensitivity $q_1 = 0.980$ (probability the LLM judges correct when human judges correct) and specificity $q_0 = 1.000$ (probability the LLM judges incorrect when human judges incorrect). The bias-adjusted score is computed as:

$$\hat{\theta} = \frac{\hat{p} + q_0 - 1}{q_0 + q_1 - 1} \tag{9}$$

where $\hat{p}$ is the raw LLM judgment score. Confidence intervals account for both the test sample variance and calibration uncertainty.

**Why the judge's specificity is near-perfect.** A specificity of 1.00 on the 200-sample calibration set might look suspicious. This value is in fact a deliberate property of the design, not a sampling artifact:

1. The judge is *not invoked* when the prediction exactly matches a gold variant ($\sim$50% of predictions). On this subset specificity and sensitivity are both 1.00 by construction.

2. For the remaining half, Prompt 4 applies the evaluation steps *sequentially*. Steps 1–3 (refusal, content mismatch, critical error) each independently route a prediction to "incorrect". Steps 4–5 (format and verbosity) only fire *after* a prediction has survived the rejection gates, so they can never *rescue* a wrong answer.

3. For QA with concrete extractive answers, a wrong value almost always hits one of the rejection gates, producing the observed one-sided behavior. The 2% gap is in *sensitivity*, not specificity: correct answers occasionally get rejected for excessive verbosity or minor formatting issues.

**LLM Judge Prompt**

You are evaluating answer correctness for a Document QA benchmark.

**Input** Question: {question}

Predicted Answer: {predicted}
Gold Answer Variants: {gold_variants}

**Evaluation Criteria** Evaluate the predicted answer based on the following definitions:

**Correct** Predicted answer is semantically equivalent to at least one gold variant. Minor format differences are acceptable.

**Partial** Predicted answer contains correct core information but has a significant format issue (e.g., list presented as comma-separated string when items are short/atomic) OR includes irrelevant additions.

**Incorrect** Predicted answer is factually wrong, missing, contains different information, or fails to answer the question type (e.g., no Yes/No for binary questions). Missing unit qualifiers that change magnitude (thousands, millions) are incorrect.

**Evaluation Steps** Follow these steps in order:

1. **Check for refusal:** Does the answer refuse or claim inability to answer? If yes → incorrect.
2. **Compare content:** Does the predicted answer match the core meaning of any gold variant? If content is wrong or different → incorrect.
3. **Check critical errors:** (any of these → incorrect)
   - Missing scale qualifiers (e.g., "50" vs "$50 million").
   - Binary questions without explicit Yes/No.
   - Wrong entity/value (different person, company, number).
   - Partial list with wrong items mixed in.
4. **Check format:** (only if content is correct)
   - Gold expects list but Predicted is string → partial.
5. **Check verbosity:** (only if content is correct)
   - **Correct:** Extra qualifiers, relevant context, clarifying phrases.
   - **Partial:** Unrequested details, over-specific precision.
   - **Incorrect:** Multi-sentence responses, full paragraphs, conversational preambles.

**Instructions** Based on your step-by-step analysis, provide your final judgment. After your reasoning, you MUST call

`submit_judgment` with your final decision.

*Prompt 4.* **LLM judge prompt** for semantic answer correctness evaluation. The judge performs chain-of-thought assessment across five criteria: refusal detection, content matching, critical errors, format issues, and verbosity.

## F.2. Adversarial Stress Test of the LLM Judge

To probe the judge on the hardest cases, we annotated 100 predictions from Gemini 3 Pro chosen such that (i) none had an exact string match to the gold answer and (ii) list-type answers were oversampled to 50%—i.e. exactly the regime where format and partial-credit disagreement is most likely.

*Table 13.* **Adversarial stress test** of the judge on 100 hard predictions (Gemini 3 Pro; no exact matches; 50% list-type oversampling). Specificity remains at 100%; sensitivity drops to 91.8% as the judge applies its rejection rules conservatively.

|                    | LLM = Correct | LLM = Incorrect |
| ------------------ | ------------- | --------------- |
| Human = Correct    | 45            | 4               |
| Human = Incorrect  | 0             | 51              |

The four over-rejections (Human-correct/LLM-incorrect) follow a consistent pattern of strict-but-reasonable adjudication:

1. A sensor-component list correct except for "window portion" vs. "a window"—judge applied the wrong-entity rule.

2. Substance correctly identified but the model omitted the concentration—judge rejected for "missing detail".

3. All items listed correctly as a single string rather than separate list elements—judge scored *partial* (Step 4); the human assigned full credit because the list structure was arguably not required.

4. Different wording whose equivalence was only apparent with access to the source document.

Specificity is unaffected: across these 100 adversarial cases, no wrong answer was ever promoted to "correct".

## F.3. Per-Answer-Type Calibration

One might ask specifically how the judge holds up on multi-part list answers, unit qualifiers, and numerical values—the cases where partial-credit ambiguity is most likely. We ran an additional stratified human calibration on 200 predictions sampled from three frontier models (Gemini 3 Pro, Claude Sonnet 4.5, GPT-5), with explicit per-category quotas.

*Table 14.* **Per-answer-type calibration of the LLM judge** ($N$=200 stratified predictions, three frontier models). Multi-part lists are the hardest category, but specificity remains above 90% and the only false positive is a single 0.5-scored near-match.

| Answer Type                        | Sensitivity     | Specificity      |
| ---------------------------------- | --------------- | ---------------- |
| Single value (name/date/number)    | 100.0% (42/42)  | 100.0% (8/8)     |
| Numerical with units               | 97.4% (38/39)   | 100.0% (11/11)   |
| Multi-part list                    | 94.9% (37/39)   | 90.9% (10/11)    |
| Yes/No / Binary                    | 100.0% (42/42)  | 100.0% (8/8)     |

The three over-rejections on multi-part lists are the same family of strict-adjudication errors reported above (first names vs. full names, one-character typo in the gold answer, correct number with appended paraphrase). The single false positive is the only one across all 200 samples: an answer containing all gold items plus one extra item received a partial score of $0.5$, where the extra item could not be verified without access to the source document.

Across the original calibration set, the adversarial stress test, and this per-category study, we have human judgments on *500 predictions*. The latter two are deliberately biased toward hard cases, but they confirm the judge's one-sided error profile: false *rejections* are the dominant failure mode and false *acceptances* are rare ($1/500$ in the unbiased aggregate).

**Implementation detail.** We invoke the judge through a structured-output *tool-call* interface rather than constrained decoding, which allows the model to reason about each rejection criterion before emitting the final judgment. This empirically improves both specificity and sensitivity over constrained-decoding alternatives in our calibration runs.

## F.4. Retrieval and Attribution Metrics

To quantify the agent's ability to locate and attribute relevant information sources, we employ the F1 score evaluated at two distinct levels of granularity: Page-level and Document-level.

For a given question $i$, let $\mathcal{R}_i$ denote the set of unique evidence units (pages or documents) cited by the agent in its final response, and let $\mathcal{G}_i$ denote the minimal set of ground truth evidence units required to answer the question.

We first calculate the Precision ($\mathrm{P}_i$) and Recall ($\mathrm{R}_i$) for the instance:

$$\mathrm{P}_i = \frac{|\mathcal{R}_i \cap \mathcal{G}_i|}{|\mathcal{R}_i|} \quad , \quad \mathrm{R}_i = \frac{|\mathcal{R}_i \cap \mathcal{G}_i|}{|\mathcal{G}_i|} \tag{10}$$

We handle edge cases as follows:

- If $|\mathcal{R}_i| = 0$ (the agent cites nothing), then $\mathrm{P}_i = 0$ and $\mathrm{F1}_i = 0$.

- Since our benchmark guarantees solvability, $|\mathcal{G}_i| \geq 1$ for all valid questions.

The F1 score for the $i$-th question is the harmonic mean of precision and recall:

$$\mathrm{F1}_i = \begin{cases} 2 \cdot \frac{\mathrm{P}_i \cdot \mathrm{R}_i}{\mathrm{P}_i + \mathrm{R}_i} & \text{if } (\mathrm{P}_i + \mathrm{R}_i) > 0 \\ 0 & \text{otherwise} \end{cases} \tag{11}$$

The final benchmark score is reported as the arithmetic mean over all $N$ test samples:

$$\text{Score} = \frac{1}{N} \sum_{i=1}^{N} \mathrm{F1}_i \tag{12}$$

We apply this formulation at two levels to diagnose specific bottlenecks in the agentic workflow:

**Page F1.** The elements of the sets $\mathcal{R}_i$ and $\mathcal{G}_i$ are unique page identifiers defined as tuples $(d, p)$, where $d$ is the document ID and $p$ is the page index. This metric enforces strict grounding: an agent is penalized if it correctly identifies the document but cites the wrong page (e.g., citing the Table of Contents instead of the specific clause on page 42).

**Doc F1.** The elements of the sets are reduced to unique document identifiers $d$. This metric evaluates the retrieval system's ability to locate the correct file within the corpus, disregarding intra-document navigation errors. Comparing Doc F1 against Page F1 allows us to isolate "last-mile" navigation failures from fundamental retrieval failures.

### F.5. Efficiency and Calibration Metrics

To evaluate whether agentic systems effectively allocate computational resources (steps) in proportion to problem difficulty, we employ a non-parametric calibration metric based on the Cumulative Difference method (Kloumann et al., 2024). Let the evaluation set consist of $N$ samples. For each sample $i$, we observe a tuple $(s_i, y_i)$, where:

- $s_i \in \mathbb{N}$ represents the *effort*, quantified as the number of discrete steps (e.g., tool calls, search actions) taken by the agent.

- $y_i \in \{0, 1\}$ represents the *outcome*, defined as the binary correctness of the final answer (based on Accuracy thresholding).

Let $\bar{y} = \frac{1}{N} \sum_{i=1}^{N} y_i$ denote the global average accuracy of the agent on the benchmark.

We first order the test samples by ascending effort. Let $\pi$ be a permutation of indices $\{1, \ldots, N\}$ such that the step counts are non-decreasing:

$$s_{\pi(1)} \leq s_{\pi(2)} \leq \cdots \leq s_{\pi(N)} \tag{13}$$

The Cumulative Difference sequence, $D = (D_0, D_1, \ldots, D_N)$, measures the accumulating deviation of the agent's performance from its mean performance as effort increases. It is defined recursively:

$$D_0 = 0 \tag{14}$$

$$D_k = \sum_{j=1}^{k} (y_{\pi(j)} - \bar{y}) \quad \text{for } k = 1, \ldots, N \tag{15}$$

The trajectory of $D_k$ reveals conditional performance regimes. Positive slope indicates a sub-population where the local accuracy exceeds the global mean $\bar{y}$. This typically occurs in low-step regimes for well-calibrated agents (easy problems are solved efficiently). Negative slope indicates a sub-population where local accuracy is below $\bar{y}$. This typically occurs in high-step regimes, where the agent expends significant effort without achieving success.

To summarize the calibration quality into a scalar metric, we utilize the Kuiper statistic $K$. This statistic measures the total range of the cumulative deviations, capturing the severity of non-uniform performance across the effort spectrum.

$$K = \max_{0 \le k \le N} (D_k) - \min_{0 \le k \le N} (D_k) \tag{16}$$

A lower $K$ value indicates "effort-invariant" performance, where the probability of correctness $P(y = 1|s)$ is independent of the number of steps taken. This implies the agent is equally reliable whether it takes 1 step or 20. A high $K$ value indicates distinct regimes of systematic over-performance and under-performance. For agentic systems, this usually manifests as a sharp decline in accuracy as step counts increase, highlighting an inability to recover from initial errors.

### F.6. Diagnostics for Interpreting K

A low Kuiper alone cannot distinguish a well-calibrated agent that terminates early from a uniformly poor agent that simply does not vary its effort. We therefore recommend reading $K$ jointly with two diagnostics: an *adaptivity sanity check* on the agent's step distribution, and a *budget-response* test on its accuracy curve.

**Adaptivity sanity check (CV).** Let $\text{CV} = \text{std}(s)/\text{mean}(s)$ denote the coefficient of variation of the per-question step counts. When $\text{CV} \approx 0$ the agent is non-adaptive (it spends roughly the same effort on every question), so $K$ is not a meaningful calibration signal regardless of its value. All evaluated agentic systems show substantial effort variation—e.g. humans $\text{CV} = 0.58$, Gemini 3 Pro $0.62$, GPT-5 $0.71$—so the trivial "always use the same effort" failure mode does not explain the observed $K$ values.

**Budget-response A(N).** For each agent we additionally compute accuracy $A(N)$ when the step cap is set to $N \in \{1, \ldots, T\}$ and test $H_0 \colon A(1) = \cdots = A(T)$. Rejecting $H_0$ means that extra compute *does* convert into accuracy, so the way the agent allocates that compute—measured by $K$—is informative. If $H_0$ cannot be rejected, the agent has saturated its budget and a low $K$ merely reflects non-improvement, not good calibration.

*Table 15.* $K$ **read jointly with budget-response.** Frontier agents are still gaining significantly between steps 9 and 10 (distinguishing high-$K$ from over-searching). GLM-4.6V Flash plateaus already at step 6, so its low $K$ reflects near-saturation rather than calibration.

| System | Significant through | $K$ |
|---|---|---|
| Gemini 3 Pro | step $9 \to 10$ | 25.8 |
| Claude Sonnet 4.5 | step $9 \to 10$ | 35.1 |
| GPT-5 | step $9 \to 10$ | 52.6 |
| GLM-4.6V Flash | step $5 \to 6$ | 27.5 |

The GLM-4.6V Flash row makes the joint reading concrete: its low $K=27.5$ is paired with low final accuracy (46.0%) and a nearly flat $A(N)$ curve (steps 5–10 add only $\sim$2 pp). The low $K$ therefore reflects an *inability* to convert extra compute into accuracy rather than well-calibrated termination. By contrast, frontier agents keep gaining significantly through step 10, so their $K$ values can be interpreted as genuine calibration signals over a budget the agent actually exploits.

We recommend that future evaluations report $K$ alongside both the CV adaptivity check and the budget-response $A(N)$.

## F.7. Sensitivity of $K$ to Step Budget $T$

We re-evaluated three frontier BM25 agents at step caps $T \in \{3, 5, 10\}$. Accuracy and $K$ both grow with $T$, but at different rates per system: when extra steps fail to convert into correct answers, hard questions move further into the high-effort tail and inflate $K$. Under low $T$ the effort distribution is flattened, so $K$ becomes less meaningful and rankings can invert (§5).

*Table 16.* **Accuracy and $K$ as a function of step budget $T$ for three BM25 MLLM agents.**

|  | Gemini 3 Pro | | Claude Sonnet 4.5 | | GPT-5 | |
|---|---|---|---|---|---|---|
| $T$ | Acc. | $K$ | Acc. | $K$ | Acc. | $K$ |
| 3 | 59.3% | 9.9 | 57.3% | 6.4 | 51.6% | 5.4 |
| 5 | 71.8% | 11.6 | 68.4% | 11.4 | 62.4% | 6.9 |
| 10 | 82.2% | 25.8 | 80.6% | 35.1 | 77.7% | 52.6 |

## F.8. Sensitivity of $K$ to Retrieval Width $k$

We re-ran two BM25 MLLM agents under retrieval widths $k \in \{3, 5, 10\}$ with all other parameters fixed. ($k=5$ is near the knee of the recall curve; 91% of search calls return fewer than 5 results in practice.) Accuracy varies by at most 2.0 percentage points (Gemini 3 Pro) and 5.1 pp (GPT-5), and $K$ is largely invariant to $k$ (Gemini: 25–28; GPT-5: 59–67), confirming that the calibration assessment is robust to this hyperparameter.

*Table 17.* **Accuracy and $K$ as a function of retrieval width $k$ for two BM25 MLLM agents.**

|  | Gemini 3 Pro | | GPT-5 | |
|---|---|---|---|---|
| $k$ | Acc. | $K$ | Acc. | $K$ |
| 3 | 80.2% | 25.5 | 69.7% | 67.0 |
| 5 | 81.8% | 25.3 | 72.6% | 61.1 |
| 10 | 82.2% | 27.8 | 74.8% | 59.1 |

## F.9. Sensitivity of Kuiper Statistic to Effort Measure

We investigate whether the choice of effort metric materially affects the Kuiper calibration statistic. For each system, we compute Kuiper under four effort definitions where available: (1) steps/calls—the number of tool calls or LLM invocations, (2) tokens (total)—sum of input and output tokens, (3) tokens (generated)—output tokens only, and (4) execution time.

Table 18 reports pairwise Spearman rank correlations between effort measures. For models achieving $\geq 60\%$ accuracy, correlations between steps and token measures range from $\rho = 0.72$ to $\rho = 0.95$, indicating that sample orderings are largely preserved across definitions.

*Table 18.* Spearman rank correlation ($\rho$) between effort measures across systems. Higher values indicate more similar sample orderings.

| System | Acc. | Steps vs Total | Steps vs Gen. | Total vs Gen. |
|---|---|---|---|---|
| BM25 GPT-5.2 | 67.8% | 0.76 | 0.95 | 0.78 |
| BM25 Claude Sonnet | 80.6% | 0.89 | 0.92 | 0.90 |
| BM25 GPT-4.1 | 60.0% | 0.79 | 0.85 | 0.72 |
| RLM GPT-5.2 | 64.2% | 0.83 | 0.74 | 0.80 |
| RLM Gemini-3-Pro | 73.8% | 0.84 | 0.93 | 0.87 |
| RLM Claude Sonnet | 70.5% | 0.91 | 0.85 | 0.89 |

Table 19 shows the resulting Kuiper statistics under each effort definition. For the six systems above, Kuiper values vary by at most 20% across metrics, confirming that the calibration assessment is robust to effort definition choice.

*Table 19.* Kuiper statistic under different effort definitions. Missing entries (–) indicate unavailable measurements.

| System | Acc. | Steps | Tokens (Total) | Tokens (Gen.) | Time |
|---|---|---|---|---|---|
| BM25 GPT-5.2 | 67.8% | 64.1 | 53.3 | 59.9 | – |
| BM25 Claude Sonnet | 80.6% | 35.4 | 36.4 | 37.8 | – |
| BM25 GPT-4.1 | 60.0% | 42.0 | 35.9 | 36.2 | – |
| RLM GPT-5.2 | 64.2% | 28.4 | 31.2 | 21.1 | 22.3 |
| RLM Gemini-3-Pro | 73.8% | 22.3 | 20.9 | 25.1 | 21.2 |
| RLM Claude Sonnet | 70.5% | 42.7 | 38.1 | 34.3 | 30.8 |
| Human | 82.2 % | 14.6 | – | – | 12.5 |

# G. Baseline Implementation Details

## G.1. BM25 MLLM Agent.

We implement a search-augmented agent baseline that combines text-based retrieval with vision-language model (VLM) reasoning. The agent iteratively searches a document collection and analyzes retrieved page images to answer questions.

We first construct a full-text search index from OCR-extracted text. Each document page is indexed with its source filename, page number, and OCR content using the Whoosh search library. The index supports boolean queries (AND, OR, NOT), phrase matching, and wildcard patterns.

The agent operates in an iterative loop, equipped with a `search_documents` tool that accepts natural language queries and returns rendered images of the top-$k$ matching pages. The agent receives these images directly (not OCR text), enabling it to leverage the VLM's visual understanding capabilities for layout-sensitive documents such as tables, forms, and figures. Tool is defined using JSON Schema and passed to the model alongside the system prompt with the following description:

> Search document collection and return images of matching pages. Supports: terms and phrases (use quotes for exact match), boolean operators (AND, OR, NOT - AND is default), wildcards (* for multiple chars, ? for single char). Examples: 'engine specifications', '"Bell 407" AND accessories', 'Bell*', 'incorporation NOT date'.

The agent produces structured outputs containing: (1) a list of answer strings, and (2) citations specifying the exact source file and page number for each piece of evidence. On the final iteration, the model is forced to provide an answer via constrained decoding (OpenAI) or tool forcing (Anthropic).

---

**Algorithm 1** Search Agent for Document QA

---

**Require:** Question $q$, search index $\mathcal{I}$, VLM $\mathcal{M}$, max iterations $T$, top-$k$
**Ensure:** Answer $a$, citations $C$
1: *messages* $\leftarrow$ [SYSTEMPROMPT, $q$]
2: **for** $t = 1$ to $T$ **do**
3:     **if** $t = T$ **then**
4:         *response* $\leftarrow \mathcal{M}(messages, force\_answer = \text{True})$
5:     **else**
6:         *response* $\leftarrow \mathcal{M}(messages, tools = [\texttt{search}, \texttt{answer}])$
7:     **end if**
8:     **if** *response* is `answer` tool call **then**
9:         **return** *response.answer, response.citations*
10:     **end if**
11:     **if** *response* is `search` tool call **then**
12:         *query* $\leftarrow$ *response.query*
13:         *results* $\leftarrow$ SEARCH$(\mathcal{I}, query, k)$         ▷ Returns (file, page) tuples
14:         *images* $\leftarrow$ [RENDERPAGE$(f, p)$ for $(f, p) \in$ *results*]
15:         Append tool result and *images* to *messages*
16:     **end if**
17: **end for**
18: **return** fallback answer from final response

---

Default hyperparameters are $T = 10$ maximum iterations and $k = 5$ results per search. Document pages are rendered as PNG images at high resolution; for Anthropic's API, images exceeding the 5MB limit are progressively downscaled using Lanczos resampling. The system prompt is provided in Prompt 5.

## G.2. Managed RAG Services

We evaluate two proprietary RAG-as-a-Service offerings that abstract the document processing pipeline, allowing us to benchmark against industry-standard solutions.

---

**BM25 MLLM Agent Prompt**

You are a document QA assistant with access to a search tool. The search tool returns images of document pages.

**Search Instructions** The answer is definitely in the documents. If search returns no results, try different terms (synonyms, abbreviations, rephrasing).

**Analysis & Output** Once relevant pages are found, analyze images and provide:

1. **answer**: List of answer values.
   - Single answer → one-element list.
   - Multiple items/entities → several elements.
   - Use as few words as possible (exact document words preferred).
   - Do not write full sentences.

2. **citations**: List of sources. Each citation must have:
   - `file`: Exact PDF filename shown in image (e.g., "1007969.pdf").
   - `page`: The page number (integer).

---

*Prompt 5.* **BM25 MLLM Agent system prompt.** The agent receives this prompt alongside the tool definition for document search.

**Gemini File Search.** Google's File Search API provides end-to-end managed retrieval. We upload all 796 PDF documents to a *file search store*, where Google's infrastructure automatically handles chunking, embedding generation, and vector indexing. At query time, the Gemini model receives access to this store as a tool. Unlike agentic approaches with explicit search–read–reason loops, File Search operates as a *single-shot* retrieval mechanism: one API call triggers retrieval, context injection, and answer generation atomically. The system returns grounding metadata indicating which document chunks were retrieved, though the internal retrieval strategy (query reformulation, re-ranking, chunk selection) remains opaque. We prompt the model to output answers in a structured format with explicit page-level citations, which we extract for evaluation.

**OpenAI Assistants File Search.** OpenAI's Assistants API provides a similar managed RAG capability through its `file_search` tool. We upload documents to a *vector store*, which handles chunking and embedding using OpenAI's proprietary models. An Assistant configured with the `file_search` tool automatically retrieves relevant chunks when processing queries. The system operates via a threads/runs abstraction: each question creates a conversation thread, triggers a run, and returns results with inline citations. We prompt the model to include page-level citations in its structured output when identifiable from context.

For both services, we provide the complete document collection without any task-specific fine-tuning or prompt engineering beyond basic answer formatting instructions (Prompt 6). This establishes a fair comparison point representing what practitioners would obtain "out of the box" from these commercial offerings.

### G.3. HEAVEN

We use the original encoder configurations by Kim et al. (2025): DSE (`MrLight/dse-qwen2-2b-mrl-v1`), and ColQwen2.5 (`vidore/colqwen2.5-v0.2`). Document Layout Analysis uses DocLayout-YOLO for title region extraction. All hyperparameters follow paper defaults. The VLM prompt is shown in Prompt 7.

### G.4. Claude Agent with Semtools

We implement an agentic baseline using the Claude Agents SDK integrated with `semtools`, a suite of CLI tools designed for semantic document processing. The agent has access to three composable Unix-style utilities:

1. *parse*: Converts non-text formats (e.g., PDF) into Markdown using LlamaParse as the backend. Outputs file paths to `stdout`, enabling pipeline composition.

---

**Managed RAG Prompt**

You are a document QA assistant with access to a file search tool. The file search tool retrieves relevant content from a collection of PDF documents.

**Important** The answer to the question is definitely in the documents. Search carefully using different terms if needed.

**Output Format** When you have the answer, respond with a JSON object in this exact format:

```
{
  "answer": ["answer value 1", "answer value 2", ...],
  "citations": [
    {"file": "exact_filename.pdf", "page": 1},
    {"file": "another_file.pdf", "page": 3}
  ]
}
```

**Where:**
- `answer`: list of answer values (use as few words as possible, exact document wording preferred)
- `citations`: list of sources with exact PDF filename and page number

*Prompt 6.* **Managed RAG prompt** used with both Gemini File Search and OpenAI Assistants File Search services.

*Table 20.* HEAVEN hyperparameters (paper defaults).

| Parameter | Symbol | Value |
|---|---|---|
| Stage 1 candidates | $k_1$ | 200 |
| Final results | $k_2$ | 5 |
| VS-page reduction factor | $r$ | 15 |
| VS-page filter ratio | – | 0.5 |
| Query filter ratio | $\rho$ | 0.25 |
| Stage 1 weight | $\alpha$ | 0.1 |
| Stage 2 weight | $\beta$ | 0.3 |

2. *search*: Performs semantic keyword search over text files using static embeddings. Supports configurable context windows (`--n-lines`), top-$k$ retrieval (`--top-k`), and distance thresholding (`--max-distance`). The tool operates on `stdin` or explicit file lists, returning matching passages with surrounding context.

3. *workspace*: Manages persistent embedding caches for document collections. Once a workspace is activated (`export SEMTOOLS_WORKSPACE=name`).

The agent uses Claude's `claude_code` system prompt preset, granting it bash execution capabilities. Tool permissions allow `Bash`, `Read`, `Glob`, `Grep`, and `Search` operations. The agent produces structured JSON output containing the answer, page-level citations, and the search history.

This paradigm allows flexible, iterative exploration but relies on the agent's ability to formulate effective bash pipelines and interpret unstructured search results. The user prompt and CLAUDE.md configuration are shown in Prompt 8.

### G.5. Recursive Language Models

Recursive Language Models (RLMs) (Zhang et al., 2025a) represent a task-agnostic inference paradigm that enables language models to handle arbitrarily long contexts by *programmatically* examining, decomposing, and recursively calling themselves over the input. Unlike traditional retrieval-augmented generation approaches that rely on fixed chunking and embedding strategies, RLMs offload the entire document corpus as a variable within a REPL (Read-Eval-Print Loop) environment that the model can interact with through code execution.

The RLM framework replaces the canonical `llm.completion(prompt)` call with an `rlm.completion(prompt)`

---

**HEAVEN VLM Prompt**

You are a document QA assistant. Answer the question based ONLY on the provided document images.

**Input** Document sources: {`context`}

Question: {`question`}

**Instructions**

1. Carefully examine ALL provided document images.

2. Find the specific information that answers the question.

3. Respond with a JSON object in the exact format defined below.

**Output Schema**

```
{
  "answer": ["answer value 1", "answer value 2", ...],
  "citations": [
    {"file": "exact_filename.pdf", "page": 1},
    {"file": "another_file.pdf", "page": 3}
  ]
}
```

**Where:**
- `answer`: list of answer values (use as few words as possible, exact document wording preferred)
- `citations`: list of sources with exact PDF filename and page number (1-indexed)

**Important** The answer to the question is definitely in the documents. Examine all provided pages carefully.

---

*Prompt 7.* **HEAVEN VLM prompt.** After hybrid retrieval returns top-$k$ pages, this prompt instructs the VLM to extract answers with citations.

call. When invoked, the system: (1) Loads the document corpus into a code-accessible variable with explicit structural markers (e.g., [FILE: `doc.pdf`], [PAGE N]). (2) Provides the LLM with a sandboxed execution environment where it can write and execute code. (3) Enables the model to launch recursive sub-LM calls via `llm_query()` to process subsets of the context.

Our implementation uses the RLM[4] library. The document corpus is sourced from OCR-processed markdown representations of the PDF collection (created with Mistral OCR 3), where page boundaries are preserved using horizontal rule separators (`---`). We augment the corpus with explicit page markers ([PAGE N] ... [END PAGE N]) to enable accurate citation extraction.

The model receives a prompt containing the full corpus text and the question, with instructions to search the corpus and return both the answer and source locations (Prompt 9). The raw RLM response is then processed through a structured output extraction step (using the same underlying model) to normalize answers and citations into a consistent format.

---

[4] https://github.com/alexzhang13/rlm

**Claude Agent User Prompt**

Try to answer the question based on the PDFs at {PDF_PATH}. All of them have been converted to Markdown and can be found at {MARKDOWN_PATH}.

**Question:** {question}

Provide the answer along with citations (file names and page number) and the search queries you used.

**Claude Agent CLAUDE.md**

**Environment Setup** Use `uv` alongside the virtual environment when running Python scripts. When requiring environment variables, use:

```
uv run --env-file keys.env main.py
```

**Augmented CLI Tooling** If executing bash commands, you have three very helpful utilities installed:

- **parse**: Converts any non-grep-able format into markdown. Outputs a filepath for a converted markdown file for every input file to stdin.

- **search**: Performs a search using static embeddings (similar to grep). Works best with keyword-based queries. Only works with text-based files (requires `parse` first for PDFs).

- **workspace**: Management for accelerating search over large collections.

**1. Parse CLI**

```
parse [OPTIONS] <FILES>...
Options:
  -c, --parse-config <PATH>  Path to config file
  -b, --backend <BACKEND>    Backend type (default: llama-parse)
```

**2. Search CLI**

```
search [OPTIONS] <QUERY> [FILES]...
Options:
  -n, --n-lines <N>        Lines context [default: 3] (Suggest: 30-50)
  --top-k <K>              Top-k files/texts [default: 3]
  -m, --max-distance <D>  Return results below distance threshold
  -i, --ignore-case       Case-insensitive search
```

**3. Workspace CLI**

```
workspace <COMMAND>
Commands:
  use     Use or create a workspace
  status  Show active workspace and stats
  prune   Remove stale files
```

**Common Usage Patterns**

```
# Parse a PDF and search for specific content
parse document.pdf | xargs cat | search "error handling"

# Search with custom context and thresholds
search "machine learning" *.txt --n-lines 5 --max-distance 0.3

# Create/Use a workspace (caches embeddings)
workspace use my-workspace
export SEMTOOLS_WORKSPACE=my-workspace
search "some keywords" ./large_dir/*.txt --n-lines 5 --top-k 10
```

*Prompt 8.* **Claude Agent user prompt and CLAUDE.md configuration.** The agent uses the `claude_code` system prompt preset with access to Unix-style semantic search tools.

**Evaluation Rules**

Verbose answers score POORLY. Answer format rules:

- answer: A list of SHORT strings with ONLY essential info.
- NO full sentences. NO explanations.
- Always return a list, even for single answers (e.g., `[\$1.2M"]`).
- Use EXACT words from the document when possible.
- NEVER include phrases like "According to...", "The document states...".

**Examples: Correct vs Incorrect**

Q: Federal housing laws protect people from discrimination on the basis of what?

[WRONG] Federal housing laws protect people from discrimination on the basis of race, color...

[CORRECT] `[\race", \color", \religion", ...]`

Q: What was the total live weight of sheep?

[WRONG] The total live weight ... was 2,400 cwt.

[CORRECT] `[\2,400 cwt"]`

**Expected Output Format** You only need to return the "answer" and "citations".

```
{
  "id": "test/0",
  "question": "What is the total revenue?",
  "answer": ["$1.2M"],
  "citations": [
    {"document": "report.pdf", "page": 5}
  ],
  "search_history": ["query1", "query2"]
}
```

*Prompt 8.* **Claude Agent user prompt and CLAUDE.md configuration.** *(Continued)*

**RLM Prompt**

You are tasked with answering a query with associated context. You can access, transform, and analyze this context interactively in a REPL environment that can recursively query sub-LLMs. You will be queried iteratively until you provide a final answer.

**REPL Environment Initialization**

1. `context`: Variable containing extremely important information. Check its content to understand what you are working with.

2. `llm_query`: Function to query an LLM (handles 500K chars) inside the REPL.

3. `llm_query_batched`: Function to query multiple prompts concurrently.
   *Signature:* `llm_query_batched(prompts: List[str]) -> List[str]`

4. `print()`: Use statements to view REPL output and continue reasoning.

**Strategy** You will only see truncated outputs from the REPL. Use `llm_query` on variables to analyze semantics. Use variables as buffers to build your answer.

**Suggested Workflow:**

- Look at `context` and determine a chunking strategy.

- Break context into smart chunks.

- Query sub-LLMs per chunk and save answers to a buffer.

- Query an LLM with all buffers to produce the final answer.

*Note:* Sub-LLMs fit ∼500K characters. You can feed ∼10 documents per sub-LLM query.

**Execution Format** Wrap Python code in triple backticks with `repl` identifier:

```repl
chunk = context[:10000]
answer = llm_query(f"What is the magic number? Chunk: {chunk}")
print(answer)
```

**Final Answer Protocol** When the task is complete, you MUST use one of these functions (do not use code blocks for this):

- `FINAL(your final answer here)`

- `FINAL_VAR(variable_name)`

Think step by step. Plan and execute immediately. Explicitly answer the original query in your final output.

*Prompt 9.* **RLM prompt.** The model receives a REPL environment with the document corpus loaded as a variable and functions to spawn recursive sub-LLM queries.

# H. Extended Results

## H.1. Multi-Hop Question Complexity Analysis

Our benchmark contains 372 multi-hop questions requiring evidence integration across multiple pages: 186 same-document (50%) and 186 cross-document (50%). For each multi-hop question, we compute: (1) *Physical distance.* The page index difference between evidence locations (same-document only). (2) *Semantic distance.* Cosine distance between page embeddings using Snowflake Arctic Embed (`snowflake-arctic-embed-m-v2.0`). We measure accuracy using five models (Claude 4.5 Sonnet, Claude 4.5 Haiku, Gemini 3.0 Pro, GPT-5, GLM-4.6V).

**Same-Document Multi-Hop.** Table 21 shows accuracy by semantic distance. Semantic similarity is a stronger predictor of difficulty than physical page distance ($r$=-0.26 vs $r$=-0.06). Questions with semantically similar evidence achieve 72.4% accuracy, while dissimilar evidence drops to 34.8%—a 38 percentage point decline.

*Table 21.* Same-document multi-hop accuracy by semantic distance between evidence pages.

| Semantic Distance | Questions | Accuracy |
|---|---|---|
| 0.0–0.15 (similar) | 29 | 72.4% |
| 0.15–0.3 | 38 | 71.1% |
| 0.3–0.45 | 51 | 56.9% |
| 0.45–0.6 | 45 | 64.0% |
| 0.6+ (dissimilar) | 23 | 34.8% |
| **Overall** | **186** | **61.2%** |

**Cross-Document Multi-Hop.** Cross-document questions (n=186) show higher overall accuracy (75.7%) compared to same-document (61.2%). The relationship between semantic distance and accuracy is weaker ($r$=-0.09), with accuracy ranging from 72.4% to 80.0% across semantic distance bins. This suggests that explicit document boundaries may help models structure the retrieval task.

*Table 22.* Cross-document multi-hop accuracy by semantic distance between evidence pages.

| Semantic Distance | Questions | Accuracy |
|---|---|---|
| 0.0–0.15 | 18 | 80.0% |
| 0.15–0.3 | 17 | 80.0% |
| 0.3–0.45 | 26 | 77.7% |
| 0.45–0.6 | 43 | 77.2% |
| 0.6+ | 82 | 72.4% |
| **Overall** | **186** | **75.7%** |

Counter-intuitively, cross-document multi-hop is easier than same-document multi-hop (75.7% vs 61.2%). This may be because cross-document questions often involve explicit comparisons (e.g., "which event occurs earlier?"), providing clearer task structure, while same-document reasoning requires subtle integration of information across distant sections of a single coherent document.

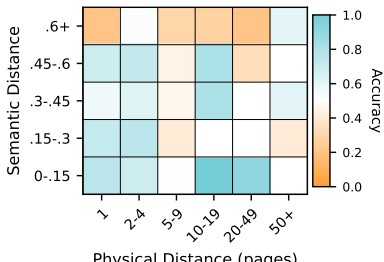

*Figure 12.* **Same-document multi-hop accuracy** by physical distance (page gap) and semantic distance (embedding cosine). Semantic distance is a stronger predictor of difficulty.

**Why the gap reflects task structure, not question-design bias.** A natural concern is whether the X-Doc / X-Page accuracy gap is an artifact of how multi-hop questions were elicited. Three pieces of evidence indicate the gap is a structural property of the task:

1. **The gap is consistent but not universal.** 25 of 35 BM25 MLLM Agent systems show X-Doc > X-Page (sign test $p = 0.008$), but the magnitude varies systematically with capability: the gap is largest for mid-tier agents ($+13$ to $+18$ pp), shrinks for the strongest agents ($+6$ pp for Gemini 3 Pro), and *reverses* for humans ($-7.6$ pp). If X-Doc questions were simply easier by construction, humans would benefit too. This capability-threshold pattern is hard to explain by elicitation bias.

2. **X-Doc evidence is harder to retrieve, not easier.** Semantic distance between evidence pages (§H.1) is a strong predictor of difficulty, and X-Doc evidence pages are *more* semantically distant than X-Page pages (mean cosine distance 0.51 vs 0.38). The X-Doc advantage therefore cannot be attributed to easier-to-find evidence; it persists after controlling for semantic distance.

3. **Answer complexity explains the gap.** X-Doc questions gravitate toward easily-comparable quantities: 27% involve explicit numerical or temporal comparisons, compared to only 5% of X-Page questions. They consequently yield shorter answers (mean 22.6 characters vs. 52.6 for X-Page), which are easier to extract once retrieval succeeds. X-Page questions more often require stitching scattered details into multi-item lists.

Other potentially confounding parameters (number of evidence pages per question, document-domain distribution) are similar across the two subsets, so we attribute the gap to the combination of retrieval difficulty and answer-extraction simplicity above.

## H.2. Qualitative Analysis of Human Baseline Errors

During the establishment of the human baseline (Section 4), we observed that human participants, despite having access to the same search tools and documents as the agentic systems, occasionally failed to provide correct answers. These instances were not results of data annotation errors, but rather genuine performance failures reflecting the cognitive load and complexity of the benchmark.

*Table 23.* **Examples of human errors**. Observed during baseline creation. These represent genuine misunderstandings rather than annotation noise, highlighting the reasoning challenges inherent in the benchmark.

| Question | Error Type | Characterization of Mistake |
|---|---|---|
| What types of pants are **not** allowed at MTM? | Negation Blindness | The participant located the correct dress code section but overlooked the negation in the query, listing permitted items rather than prohibited ones. |
| Was the Sodium level detected [...] within normal range? | Incomplete Evidence Retrieval | The participant found the specific sodium test result but failed to retrieve the separate page containing the reference "normal range" table required for verification. |
| Which Forensic Scientist **approved** the [...] report? | Role Conflation | The participant provided the name of the *Certifying Scientist* who electronically signed the work order, confusing this role with the *Forensic Scientist* mentioned elsewhere in the document chain. |
| What water-related **act** did Pennsylvania introduce...? | Lexical Polysemy | The participant was misled by the polysemy of the word "act," interpreting it as a physical deed (answering that they "opened a bridge") rather than a legislative statute. |
| **How long after** public transportation started [...] did deaths decrease? | Temporal Reasoning Failure | The question required calculating a duration (time delta). The participant identified the correct dates but answered with a specific calendar year instead of the elapsed time. |
| What is the **largest** penalty assessed in case 3AN-24-04508CI? | Ordinal Ranking Error | The participant located the correct legal document but reported the second-largest penalty listed, likely due to skimming or a failure to sort the values exhaustively. |

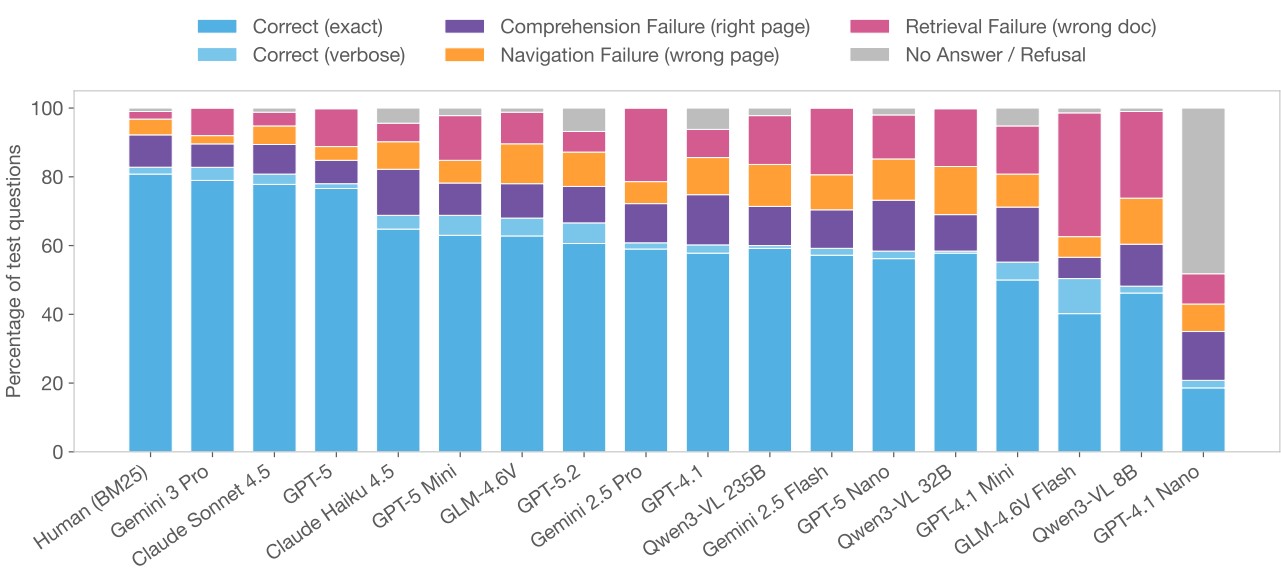

*Figure 13.* **Error decomposition across all BM25 MLLM agents**, ordered by accuracy. Each bar decomposes a system's test predictions into correct (exact and verbose) and four error types. Weaker models are dominated by refusals and retrieval failures, while stronger models shift toward comprehension errors—suggesting that retrieval is largely solved for top systems and answer extraction remains the bottleneck.

These errors underscore the difficulty of the task, specifically regarding attention to detail, cross-page navigation, and semantic precision. Table 23 categorizes common failure modes observed during the human baseline study.

### H.3. Error Decomposition

**Methodology.** We analyze the BM25 MLLM Agent configuration, which provides the broadest model coverage (17 systems spanning five model families, plus the human baseline with the same search tool). We decompose every prediction into a four-stage cascade using citation metadata and the paper's LLM judge (Section 3) for correctness. Correctness is determined by the Semantic Accuracy score (LLM judge): a score $\geq 0.5$ counts as correct (with $0.5$ indicating a verbose but semantically valid answer, and $1.0$ a precise match). For each *incorrect* prediction, the error type is assigned deterministically:

1. **Retrieval Failure** — the system never retrieved the gold document (Doc F1 = 0).
2. **Navigation Failure** — the gold document was found but the gold page was not (Doc F1 > 0, Page F1 = 0).
3. **Comprehension Failure** — the correct page was retrieved but the answer is wrong (Page F1 > 0).
4. **No Answer / Refusal** — the system returned an empty response.

This taxonomy requires no subjective LLM classification for the error *type* assignment—only for the correct/incorrect boundary.

**Aggregate Error Distribution.** Across 17 agent systems (8,499 predictions, 3,273 incorrect), the error breakdown is:

*Table 24.* Aggregate error distribution across all agent systems (N=3,273 incorrect predictions).

| Error Type | Count | % |
|---|---|---|
| Retrieval (wrong document) | 1,167 | 35.7 |
| Comprehension (right page, wrong answer) | 941 | 28.8 |
| Navigation (right doc, wrong page) | 753 | 23.0 |
| No Answer / Refusal | 412 | 12.6 |

**Per-System Failure Profiles.** Figure 13 shows the full decomposition across all systems. Key observations:

- *Claude Sonnet 4.5* has the lowest retrieval failure rate among agents (4.0%) but the highest comprehension failure rate among top models (8.6%)—it finds content effectively but misinterprets it.
- *Smaller and open-weight models* exhibit the highest retrieval failure rates: *GLM-4.6V Flash* (36.0%), *Qwen3-VL 8B* (25.2%), and *Gemini 2.5 Pro* (21.4%), indicating that search query formulation is a primary bottleneck for less capable agents.
- *GPT-4.1 Nano* is dominated by refusals (48.2% of predictions)—the model abandons search prematurely rather than failing at any specific stage.
- The *Human* baseline has the lowest retrieval failure (2.2%) and navigation failure (4.6%), but a relatively high comprehension rate (9.4%), confirming that when humans have the right evidence, occasional attention errors still occur (cf. Table 23).

**How Error Composition Shifts with Model Capability.** Figure 14 shows that as model accuracy increases, the composition of errors changes qualitatively. Weaker models are dominated by refusals ("no answer")—they give up. As capability increases, retrieval failure becomes the dominant bottleneck. The strongest models have error profiles dominated by comprehension failures, indicating that retrieval is largely solved and the remaining challenge is answer extraction.

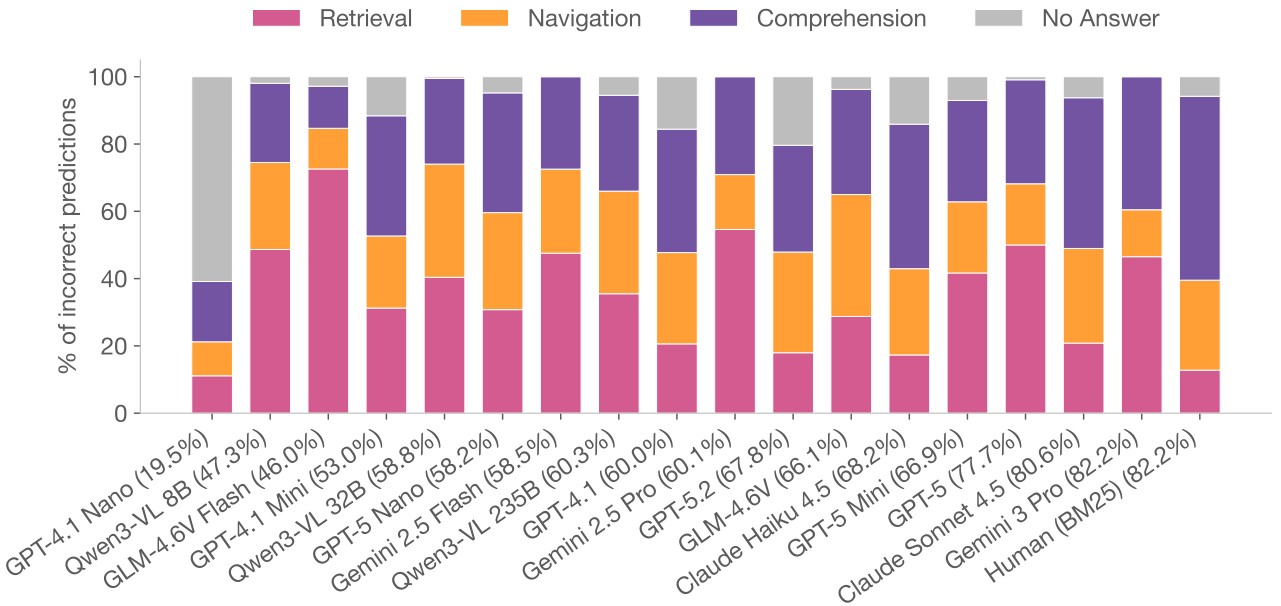

*Figure 14.* **Error composition shifts with model capability.** Systems ordered by increasing accuracy (left to right). Weaker models are dominated by refusals; stronger models shift toward retrieval and comprehension failures.

**Retrieval vs. Comprehension Failure Profiles.** Figure 15 maps each system's retrieval and comprehension failure rates, revealing distinct clusters. The strongest models (Gemini 3 Pro, GPT-5, Claude Sonnet 4.5) occupy the bottom-left quadrant—low failure rates on both axes. Mid-tier models split into two profiles: *retrieval-limited* systems (Qwen3-VL, GLM-4.6V Flash) that rarely reach the comprehension stage because they fail to find the right document, and *comprehension-limited* systems (GPT-4.1 family, Claude Haiku 4.5) that retrieve well but struggle with answer extraction. Notably, the human baseline has the lowest retrieval failure rate but non-trivial comprehension failure, consistent with the attention-based errors described in Table 23.

**Error Types by Evidence Complexity.** Figure 16 breaks down errors by hop type for the top four systems. Cross-page (same-document) questions are hardest, with error rates rising from ∼13% for single-evidence to ∼35% for cross-page. Retrieval failure accounts for a larger share of cross-page errors, suggesting that navigating within long documents is harder than finding separate documents.

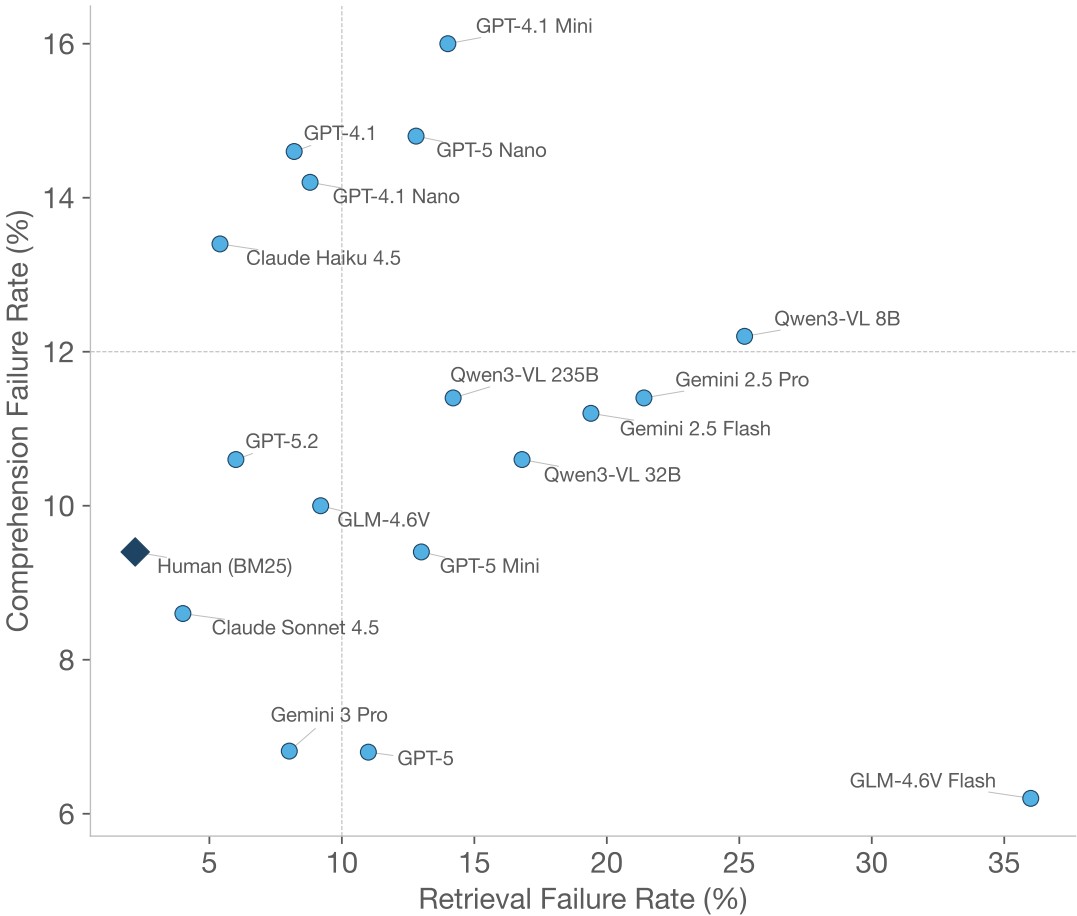

*Figure 15.* **Retrieval vs. comprehension failure profiles.** Each point is one system. Models in the bottom-left corner fail least on both dimensions. The dashed lines mark approximate cluster boundaries.

**Error Types by Document Domain.** Figure 17 aggregates errors by document domain for the top four systems. Event-related documents are easiest (92% accuracy), likely due to structured formats with clear named entities. Media/Publishing (69%) and Financial documents (74%) are hardest, with Financial documents showing the highest retrieval failure rate—consistent with the challenge of navigating dense numerical tables across many pages.

**Failure Cascade.** Figure 18 presents the failure cascade for four representative systems. Each error stage is shown as a cumulative funnel: total questions → document found → correct page found → correct answer. This visualization makes the progressive loss at each stage immediately visible.

### H.4. Cross-System Item Agreement

**Methodology.** For each of the 500 test questions, we build a binary correctness vector across all BM25 MLLM Agent systems and the human baseline. We measure pairwise agreement using Cohen's $\kappa$, which accounts for chance agreement due to marginal accuracy rates.

**Pairwise Agreement.** Figure 19 shows the full pairwise $\kappa$ matrix. Model–model pairs within similar capability tiers achieve moderate agreement ($\kappa \approx 0.4$–$0.6$), confirming shared systematic strengths and weaknesses. Cross-family agreement is lower (e.g., GPT vs. Gemini $\kappa \approx 0.3$–$0.4$), indicating that model families have distinct failure signatures.

The most striking pattern is the *Human row*: agreement between the human baseline and every agent is low ($\kappa = 0.06$–$0.24$), despite similar overall accuracy. For example, Human and Gemini 3 Pro both achieve $\sim 82\%$ accuracy yet share $\kappa = 0.24$. Of

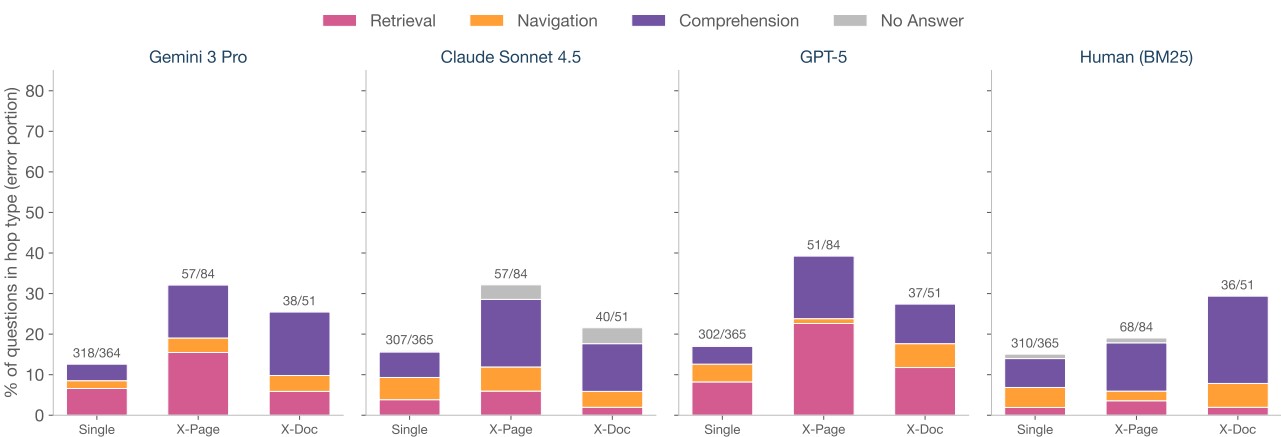

*Figure 16.* **Error types by evidence complexity** for top-4 systems. Cross-page (same document) questions are consistently hardest.

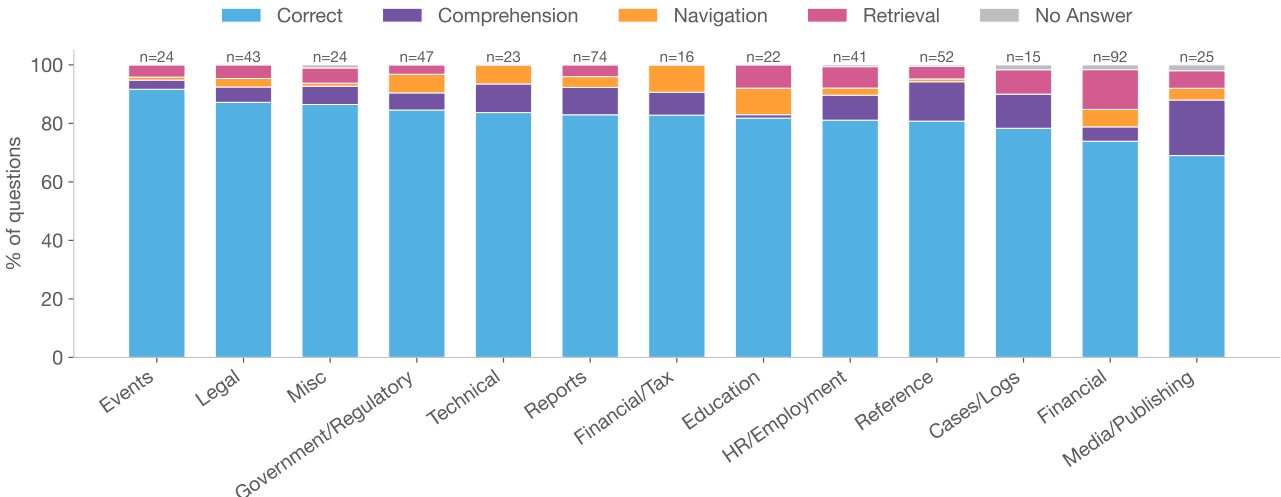

*Figure 17.* **Error decomposition by document domain** (top-4 systems aggregated). Media/Publishing and Financial domains are hardest; Events documents are easiest.

their 107 disagreement items, human-specific failures are dominated by comprehension errors (64%), while model-specific failures split between retrieval (43%) and comprehension (43%). This confirms that humans and agents solve fundamentally different subsets of the benchmark.

**Item Difficulty Spectrum.** Across the 500 test items, 23.8% are universally easy (>90% of models correct), 8.4% are universally hard (<10% correct), and the remaining 66.8% are discriminating (Figure 20). Critically, only 5 items (1.0%) are *anomalous*—easy for weak models but hard for strong ones—confirming that the benchmark contains no "trick questions" and exhibits sound psychometric properties: difficulty increases monotonically with model capability. Among the 86 items where humans fail, 24 have model difficulty ≥70%, revealing human-specific cognitive limitations (attention fatigue, negation blindness) on questions that most models solve reliably.

### H.5. Search Trajectory Analysis

**First-Query Advantage.** Figure 21 plots cumulative document and page retrieval rates by search iteration. Humans find the gold document on their first query ∼80% of the time, whereas the best agent (Gemini 3 Pro) starts at ∼70%. The gap narrows by iteration 4–5, but the human first-query advantage persists in page-level retrieval.

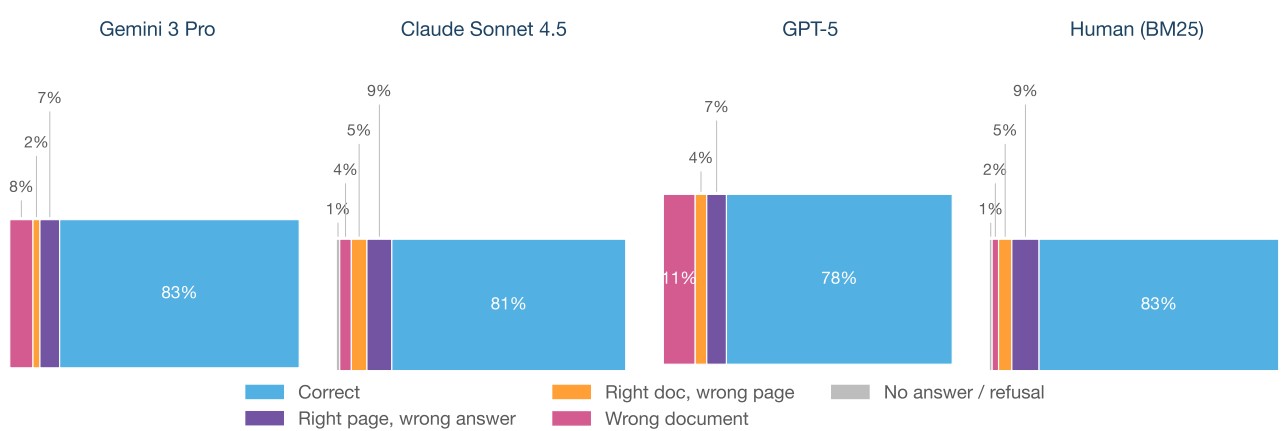

*Figure 18.* **Error cascade: where do systems fail?** Each horizontal bar decomposes a system's predictions by error stage. Percentages indicate the share of predictions in each category.

**Recovery After Initial Failure.** When the first search query fails to retrieve the gold document, recovery success varies dramatically (Figure 22). Claude Sonnet 4.5 and Gemini 3 Pro recover in >90% of cases, matching the human rate (∼97%). Weaker models (Gemini 2.5 Pro: 56%, GPT-4.1 Nano: 12%) rarely recover, explaining much of the accuracy gap.

### H.6. Query Reformulation Analysis

**Methodology.** To quantify how aggressively systems reformulate their search queries, we embed all queries using Snowflake Arctic Embed (`snowflake-arctic-embed-m-v2.0`) and measure cosine distance between consecutive queries for each question trajectory. We define *mean drift per step* as $\bar{d} = \frac{1}{k-1} \sum_{i=1}^{k-1} (1 - \cos(q_i, q_{i+1}))$ for a trajectory with $k$ queries.

**Reformulation Magnitude by System.** Figure 23 shows the distribution of mean drift per step across systems, ordered by median. Top-performing systems reformulate more aggressively: Claude Sonnet 4.5 has a median drift of 0.38, while GPT-4.1 Nano barely changes its queries (median drift 0.10). The correlation between reformulation magnitude and accuracy is strong (Figure 24, right panel).

**Single-Query vs. Multi-Query Accuracy.** Every system performs better on questions answered in a single query. The gap quantifies multi-step difficulty: for example, Human achieves 86.4% on single-query questions vs. 79.5% on multi-query, while GPT-4.1 Nano drops from 46.1% to 13.2%—a 33 percentage point gap that reveals its inability to effectively reformulate.

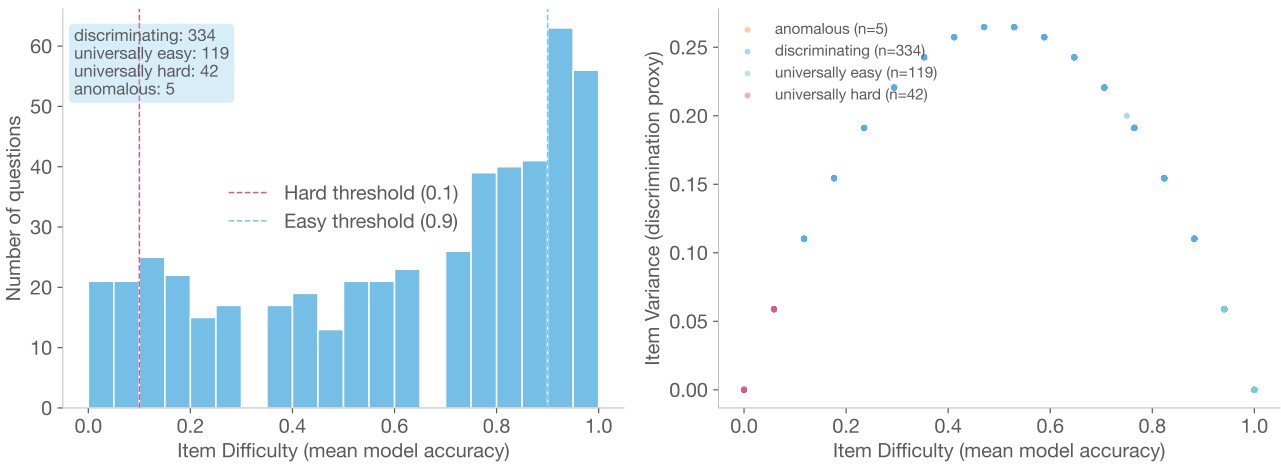

*Figure 19.* **Pairwise system agreement (Cohen's $\kappa$).** Darker cells indicate higher agreement. The Human baseline shows low agreement with all agent systems despite comparable accuracy, revealing complementary error patterns.

*Figure 20.* **Left:** Distribution of item difficulty (mean model accuracy). **Right:** Difficulty vs. variance; high-variance items are the most discriminating. Only 5 anomalous items exist, confirming monotonic difficulty scaling.

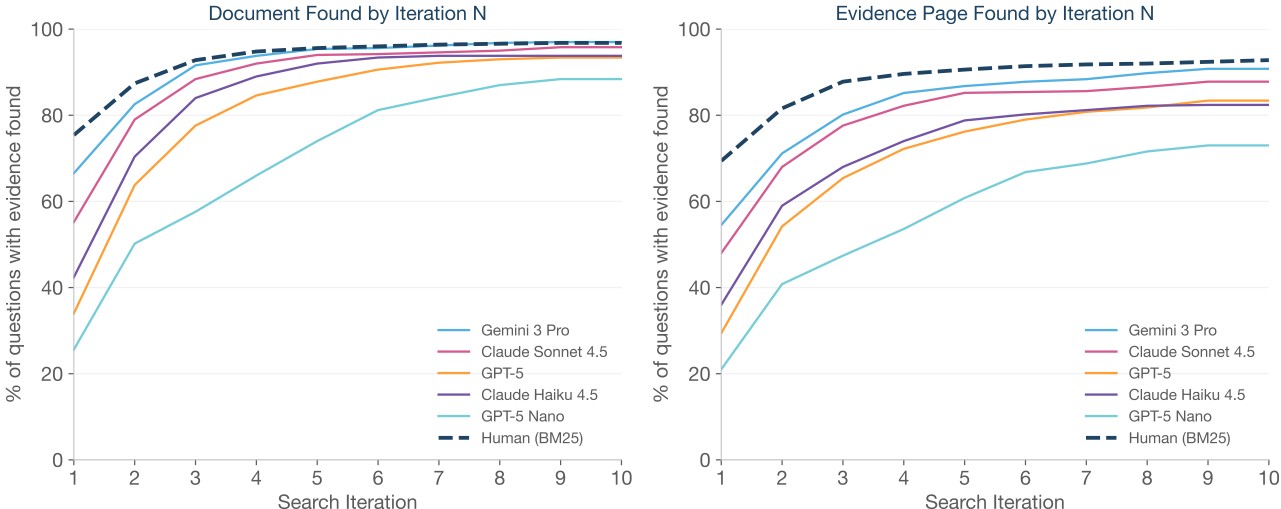

*Figure 21.* **Evidence found by iteration** $N$**.** Left: document retrieval. Right: page retrieval. Humans find evidence earlier, but top agents converge by iteration 5.

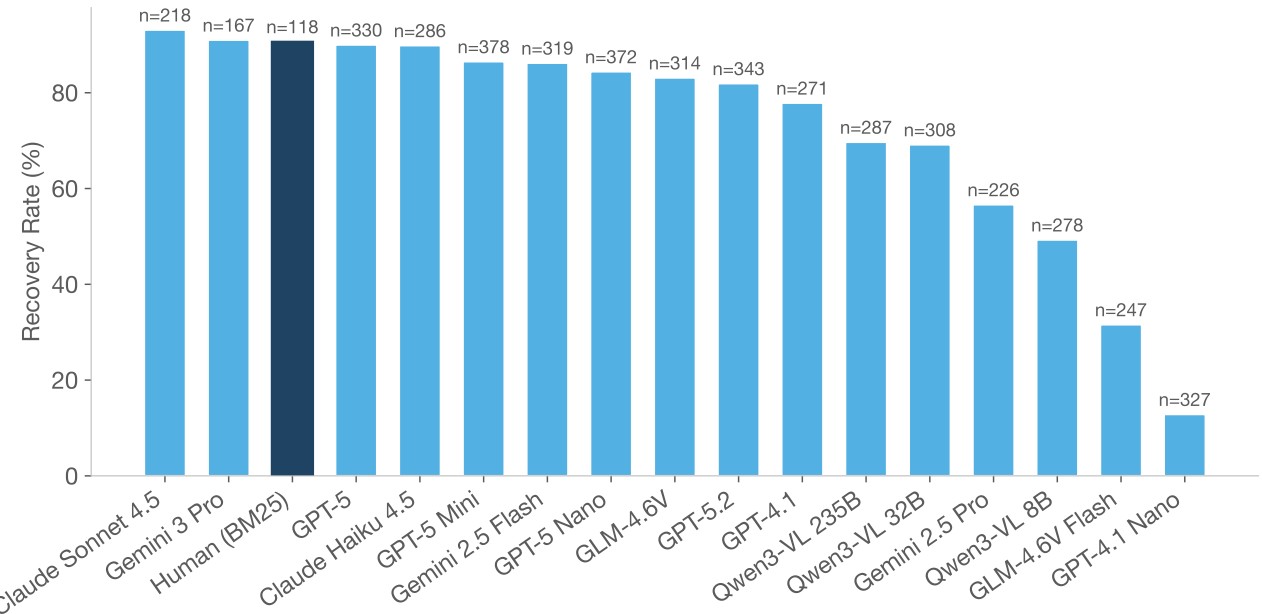

*Figure 22.* **Recovery rate after initial query miss.** Percentage of questions where the system eventually finds the gold document after failing on the first query. Labels show the number of questions requiring recovery ($n$). Notably, the human baseline needs recovery for only 118 questions—far fewer than most agents (typically 250–380)—indicating that humans craft more precise initial queries. Among the cases that do require recovery, top agents match human recovery rates ($\sim$93%); weaker models rarely recover.

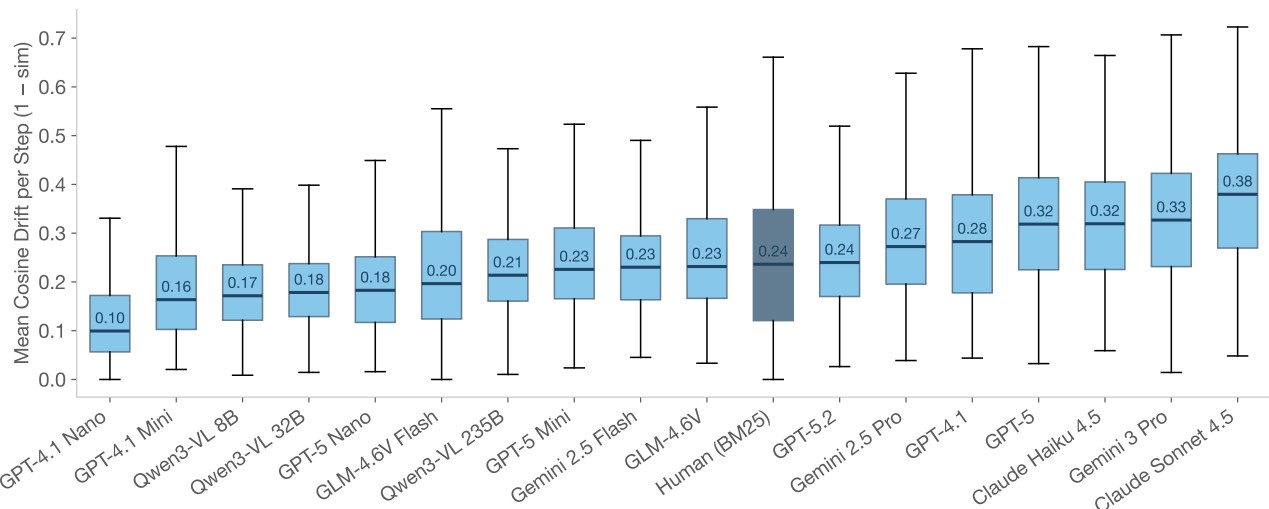

*Figure 23.* **Query reformulation magnitude by system.** Higher drift indicates more aggressive query changes. Box shows IQR with median labeled. Systems sorted by median drift.

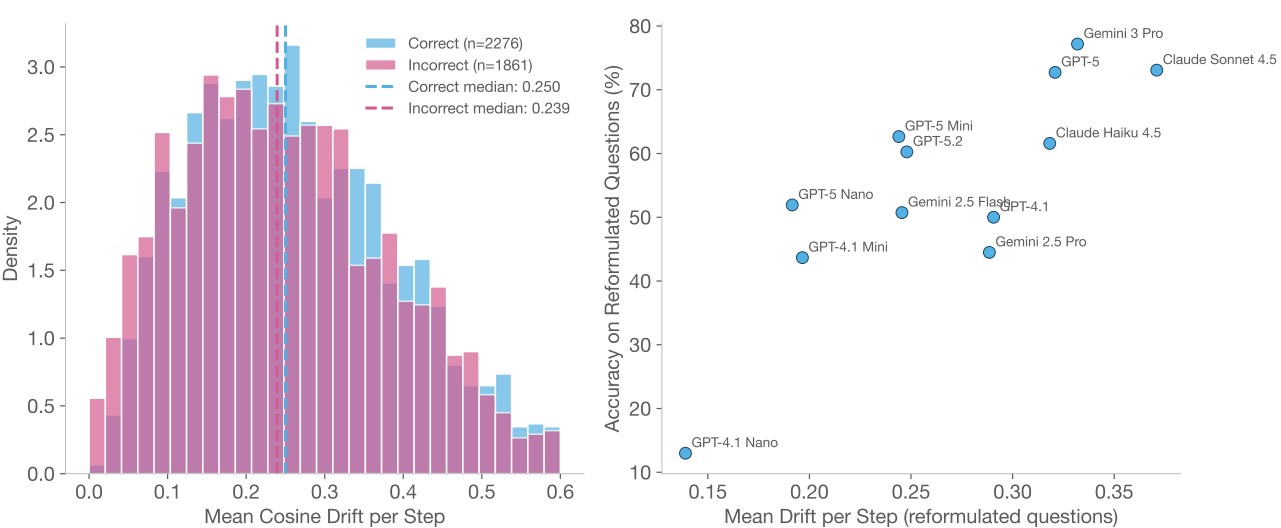

*Figure 24.* **Left:** Drift distribution for correct vs. incorrect predictions. **Right:** Per-system mean drift vs. accuracy on reformulated questions, showing a positive correlation between reformulation effort and success.

# I. Solution Space

This design space provides a structural lens for interpreting the empirical findings presented in the main text. The comparative analysis highlights critical trade-offs across these specific dimensions:

While flexible *Programmatic* flows (e.g., Recursive Language Models) offer theoretical advantages, our efficiency analysis demonstrates that they currently suffer from catastrophic computational overhead compared to constrained *Iterative* architectures (e.g., BM25 MLLM Agent). This suggests that without robust guardrails, unconstrained reasoning leads to the diminishing returns observed in the efficiency calibration curves.

The necessity of the *Visual* primitive is validated by our Construct Validity analysis, which finds that 57.2% of questions require structural or visual comprehension (e.g., tables, forms). This confirms that systems relying solely on *Textual* representations (Markdown/JSON) are architecturally limited compared to those utilizing full page images or visual crops.

With retrieval failures accounting for 39.1% of all errors, the selection of *Indexing* and *Scope* is paramount. The performance hierarchy suggests that simple Sparse indexing often fails to capture semantic nuance, necessitating more advanced *Late Interaction* or hybrid strategies (like HEAVEN) to handle the high recall requirements of the benchmark.

Finally, the pervasive issue of "stochastic search" identified in our behavioral analysis highlights the *Memory* sub-dimension as a critical area for future optimization. Implementing *Episodic Stores* could allow agents to learn corpus-specific terminology across queries, reducing the wasteful exploratory loops observed in current frontier models.

*Table 25.* Morphological Analysis of the Agentic Document Collection Answering Solution Space

| Dimension | Sub-Dimension | Primitives / Options |
|---|---|---|
| **1. Control Flow** | Architecture | Single-step RAG, Iterative (ReAct), Hierarchical (Manager-Worker), Map-Reduce |
| | Reasoning | Zero-shot, Chain-of-Thought, Tree-of-Thoughts, Reflexion |
| | Memory | Stateless, Sliding Window, Summarized History, Episodic Store |
| **2. Representation** | Textual | Plain Text, Markdown, JSON, HTML, Semantic Chunks |
| | Visual | Full Page Image, Bounding Box Crops, Latent Visual Embeddings |
| | Graph | Knowledge Graph (Entities), Document Structure Graph (DOM) |
| **3. Retrieval** | Indexing | Sparse (BM25), Dense (Bi-encoder), Late Interaction (ColBERT/Multi-vector) |
| | Reranking | None, Cross-Encoder, LLM-based Reranking |
| | Scope | Chunk-level, Page-level, Document-level, Parent-Child |
| **4. Generation** | Synthesis | Extractive (Span selection), Abstractive (New synthesis) |
| | Grounding | Citation (Sentence-level), Citation (Chunk-level), Visual Bounding Boxes |

# J. Landscape of Agentic Approaches to Documents

Over the past few years, document understanding approaches have progressed from systems combining OCR with layout-aware transformers (Xu et al., 2020; Powalski et al., 2021; Garncarek et al., 2021) to specialized OCR-free models (Kim et al., 2022; Lee et al., 2023), and most recently, MLLMs (Chen et al., 2024; Bai et al., 2023; Wu et al., 2024; OpenAI et al., 2024; Liu et al., 2023a). Despite this architectural evolution, the evaluation remains largely focused on answering questions based on a single input document.

The benchmarks in this area evolved from simpler single-page documents (Mathew et al., 2021; 2022) to slightly longer (Tito et al., 2023; Landeghem et al., 2023), and massive PDFs (Chia et al., 2025; Ma et al., 2024b). Although emerging datasets introduce document collections as input, they often rely on semi-automated annotation pipelines (Cho et al., 2024; Dong et al., 2025) or frame the problem as single-step retrieval (Faysse et al., 2025b), failing to capture the iterative planning required for complex reasoning.

To address this complexity, recent research has adopted agentic frameworks (Yao et al., 2023; Shinn et al., 2023), which enable models to interleave reasoning, retrieval, and self-correction. However, the benchmarks used to evaluate these capabilities (Rosset et al., 2025; Su et al., 2025), predominantly operate over HTML web pages or plain text.

## J.1. Related Works Assessment

Table 1 summarizes prior benchmarks along three axes: (1) document diversity in terms of subject domains and layouts; (2) whether questions and answers are fully human-annotated or partially automated; and (3) the task framing (single-document VQA vs. document RAG vs. agentic research). We briefly justify our assignments here.

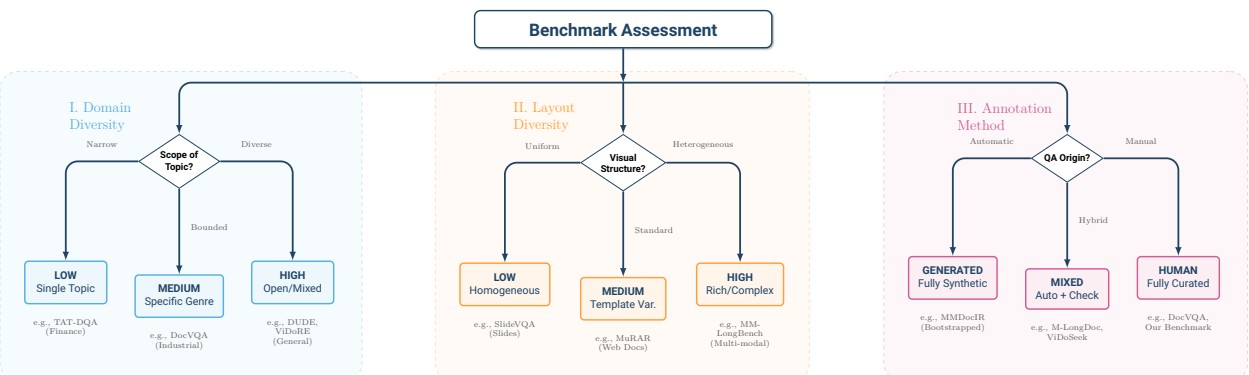

*Figure 25.* **Taxonomy of document-based benchmarks.** We structure our assessment of related works along three axes: (I) Domain Diversity, ranging from single-topic to open-domain; (II) Layout Diversity, evaluating visual structural complexity; and (III) Annotation Method, distinguishing between synthetic, hybrid, and fully human-curated pipelines.

We use ● low, ● medium, and ● high as coarse, relative categories: ● low indicates a single or very narrow domain or layout type; ● medium covers a small number of fairly similar domains/layouts; ● high corresponds to broad open-domain coverage or many heterogeneous layouts (See Figure 25).

**DocVQA (Mathew et al., 2021).** DocVQA collects scanned business documents such as invoices, forms, and letters. The subject matter is mostly industrial/administrative, but spans multiple document types, so we mark domain diversity as ● medium. Layouts vary across templates but remain in the family of structured forms and reports, hence ● medium layout diversity. Questions and answers are human-written and grounded in each single page, so we mark the dataset as fully human-annotated (✓).

**InfographicVQA (Mathew et al., 2022).** This benchmark uses infographics as the sole document type. Infographics cover a fairly wide range of topics but are all designed as visual posters, leading to ● medium domain diversity and ● high layout diversity. Questions and answers are manually annotated on each infographic, so we use ✓.

**TAT-DQA (Zhu et al., 2022).** TAT-DQA focuses on question answering over financial tables from corporate reports. Both the domain (finance) and the layouts (tabular pages with similar structures) are narrow, which we denote as ● low domain

and ● low layout diversity. QA pairs are human-annotated, so we mark ✓.

**DUDE (Landeghem et al., 2023).** DUDE includes multi-page PDFs from several domains (e.g., legal, scientific, technical), with rich layouts involving paragraphs, tables, and figures. We therefore assign ● high diversity for both domains and layouts. Questions and answers are manually curated, so the dataset is fully human-annotated (✓).

**MP-DocVQA (Tito et al., 2023).** MP-DocVQA extends DocVQA to multi-page documents but still focuses primarily on industrial PDFs (forms, reports). We rate both domain and layout diversity as ● medium. Many QA pairs originate from DocVQA and are extended heuristically to multiple pages, so we mark the annotation as mixed (✓ / ✗).

**SlideVQA (Tanaka et al., 2023).** SlideVQA consists of slide decks (PowerPoint-style documents) from a variety of topics, yielding ● medium domain diversity but relatively homogeneous slide layouts (● low). Questions are collected and verified by humans, thus the dataset is fully human-annotated (✓).

**M-LongDoc (Chia et al., 2025).** M-LongDoc comprises recent, very long multimodal documents with hundreds of pages from three domains: academic research papers, financial reports, and product manuals. These three domains justify a ● medium rating for domain diversity. The documents exhibit multiple structures (multi-column text, tables, figures, diagrams), but still within a limited set of genres, so we mark layout diversity as ● medium. Questions are generated by several powerful LMMs (Claude 3.5, GPT-4o, Gemini 1.5 Pro) and then filtered by both automated checks and human annotators, i.e., a semi-automatic pipeline, so we mark the annotation as mixed (✓ / ✗).

**MMLongBench-Doc (Ma et al., 2024b).** MMLongBench-Doc is constructed over 130+ long PDF documents from seven diverse domains with rich multimodal layouts (text, tables, charts, images). Questions are expert-annotated and carefully designed to require evidence from multiple pages and multiple modalities. We therefore assign ● high domain and ● high layout diversity, and ✓ for fully human-annotated QA.

**MuRAR (Zhu et al., 2025).** MuRAR is a multimodal RAG framework evaluated on a knowledge base built from Adobe Experience League documentation pages. Because all content comes from a single enterprise product ecosystem, we assign ● low domain diversity. Pages mix text, screenshots, tables, and videos inside a single website template, which we label as ● medium layout diversity. Questions and answers are largely generated and refined by LLMs and then evaluated by humans, so we mark mixed annotation (✓ / ✗).

**M$^2$RAG (Liu et al., 2025).** M$^2$RAG builds on ELI5-style open-domain questions and retrieves multimodal evidence (web pages plus images) across many topical categories (science, society, health, politics, etc.). This leads us to mark both domain and layout diversity as ● high. Questions are human-written, but document retrieval, multimodal evidence construction, and some labels rely on automatic pipelines and LLMs, so we use ✓ / ✗.

**ViDoRE (Faysse et al., 2025b).** ViDoRE is the visual document retrieval benchmark introduced in the ColPali work and aggregates several page-level tasks (DocVQA, InfographicVQA, TAT-DQA, TabFQuAD, and synthetic DocQA datasets) across multiple domains and languages. Documents include forms, infographics, tables, reports, and synthetic PDFs, so we mark ● high for both domains and layouts. Since ViDoRE inherits human QA from existing datasets and also introduces synthetic queries and QA, we regard the benchmark as having mixed human and automated annotation (✓ / ✗).

**ViDoRE v3 (Loison et al., 2026).** Queries are generated via contextually-blind LLM prompting (conditioned on document descriptions, not the documents themselves), then filtered and verified by multiple human annotators who also provide reference answers and bounding-box annotations. While this mitigates extractive bias, the generating model still influences the distribution of question types and linguistic patterns. We therefore retain the mixed annotation mark (✓ / ✗), acknowledging that the human component is considerably stronger than in most benchmarks with this designation.

**DocBench (Zou et al., 2024).** DocBench collects 229 real documents from five domains (scientific articles, annual reports, legal documents, government reports, and newspaper front pages) with heterogeneous layouts (multi-column text, figures, tables, charts). We thus assign ● high domain and ● high layout diversity. Questions and reference answers are partly generated by GPT-4 and subsequently refined or checked by human annotators, so we mark ✓ / ✗.

**M3DocRAG / M3DocVQA (Cho et al., 2024).** M3DocRAG introduces M3DocVQA, an open-domain DocVQA benchmark created by taking MultimodalQA question–answer pairs and converting their supporting Wikipedia pages into PDFs. Because all documents originate from Wikipedia, topics are broad but still constrained to encyclopedic style, which we mark as ● medium domain diversity and ● low layout diversity (Wikipedia and its rendered PDFs share fairly uniform templates). The QA origin is human (from MultimodalQA), but the new open-domain, multi-document setting and PDF rendering are built

automatically, so we use ✓ / ✗.

**MMDocIR (Dong et al., 2025).** MMDocIR is a multimodal retrieval and DocRAG benchmark constructed by combining many existing document-understanding datasets (including DocVQA, InfographicVQA, TAT-DQA, MMLongBench-Doc, DocBench, etc.) and adding 1.6k+ manually annotated queries plus a much larger automatically bootstrapped set. It inherits broad domain and layout heterogeneity, so we assign ● high / ● high. Given the mixture of expert labels and bootstrapped or inherited labels, we mark ✓ / ✗.

**FinRAGBench-V (Zhao et al., 2025).** FinRAGBench-V is a financial visual RAG benchmark: its corpus consists of financial reports, statements, regulatory filings, and related documents, all within the financial domain but with many kinds of charts, tables, and text-heavy pages. We therefore mark domain diversity as ● low (single domain) but layout diversity as ● high. The accompanying QA dataset is explicitly described as high-quality and human-annotated, so we assign ✓.

**BRIGHT (Su et al., 2025).** BRIGHT is a reasoning-intensive *retrieval* benchmark: it comprises 1,384 real-world queries from 12 datasets spanning biology, earth science, economics, psychology, robotics, StackOverflow, AoPS, TheoremQA, and more, with documents that include blogs, news articles, documentation, and reports. We thus assign ● high domain diversity and ● medium layout diversity (web and text-centric pages). Queries and relevance judgments are built from human-authored sources (no synthetic questions), so we mark BRIGHT as fully human-annotated (✓).

**Researchy Questions (Rosset et al., 2025).** Researchy Questions starts from real user queries in commercial search logs and filters them to non-factoid, decompositional "researchy" questions using GPT-3 / ChatGPT classifiers and GPT-4 for the final filtering stage. Each question is linked to clicked URLs in ClueWeb22, giving a large, open-domain web corpus with many site types and layouts; we assign ● high domain and ● high layout diversity. Because questions and clicks are human, but the filtering and decompositional labels rely heavily on LLMs, we mark mixed annotation (✓ / ✗). The dataset does not ship reference answers, only queries and document signals.

**ViDoSeek (Wang et al., 2025).** ViDoSeek is a benchmark for visually rich document retrieval–reason–answer, consisting of PDF documents (often slides, but also standards, technical reports, and whitepapers) and annotated queries with single- or multi-hop types, reference answers, and reference pages. Documents span many technical and governmental topics and combine 2D layout, charts, tables, and free text, so we assign ● high domain and ● high layout diversity. Queries and answers are produced via an LLM-assisted pipeline with human filtering and quality control, hence we mark ✓ / ✗.

**Our benchmark.** Our benchmark is built from a curated collection of complex, multi-page PDFs drawn from heterogeneous real-world sources (e.g., technical reports, standards, slide decks, and research documents), which we categorize as ● high domain and ● high layout diversity. Importantly, all questions are written by humans specifically for this benchmark, and the documents themselves are selected by humans to support challenging multi-hop, agentic reasoning. We therefore mark the dataset as fully human-annotated (✓).

