# OpenReview forum: "Strategic Navigation or Stochastic Search? How Agents and Humans Reason Over Document Collections"
_ICML.cc/2026/Conference — ICML 2026 spotlight_

### Official Review · Reviewer_xNxE · 2026-03-01

**Soundness:** 3
**Presentation:** 4
**Significance:** 3
**Originality:** 3
**Overall Recommendation:** 6
**Confidence:** 3

**Summary:**

This manuscript presents MADQA (Multimodal Agentic Document QA), a new benchmark for evaluating document question answering systems that require multi-hop reasoning, visual understanding, and agentic behavior across heterogeneous PDF collections.

**Compliance With Llm Reviewing Policy:**

Affirmed.

**Final Justification:**

I have read the responses from the authors and the other reviewers, and I will maintain my score.

**Key Questions For Authors:**

See Weaknesses.

**Strengths And Weaknesses:**

strengths:
- The paper addresses a fundamental question in agentic document understanding: whether modern multimodal agents perform strategic reasoning or rely on stochastic search. This is an important and timely problem for both academic research and real-world document automation systems.
- The paper provides a thorough construct validity analysis, including lexical overlap experiments and parametric-knowledge tests, demonstrating that the benchmark requires genuine document grounding and multi-step reasoning.
- The overall work is very solid and well presented. In my opinion, this is a very strong paper.

weaknesses:
- The proposed Kuiper-based calibration metric relies on step counts as a proxy for effort, but the relationship between step count and true computational effort or uncertainty is not rigorously justified.

---

> ### Author Rebuttal · Authors · 2026-03-30
>
> **We thank Reviewer xNxE for the strong endorsement** and for identifying the step-count limitation.
>
> ---
>
> ### Weakness 1 (W1)
> We share the concern of using steps as an effort measure. To assess their robustness, we compute Kuiper under four different effort definitions in Appendix F.4: step counts, total tokens, generated tokens, and wall-clock time.
>
> Pairwise Spearman correlations between these measures range from ρ = 0.72 to 0.95, and the resulting Kuiper values vary by less than 20% across definitions. The calibration assessment is somewhat sensitive to the choice of effort measure, but the relative rankings of systems remain largely consistent across definitions.
>
> That said, we acknowledge this is an empirical correlation, not a formal justification. A "step" bundles different amounts of computation depending on tool design and model interface.
>
> We chose step counts as a simple, safe, and broadly applicable default.
>
> Every alternative has practical drawbacks:
> - Token counts raise questions about how to weight input, reasoning, and output tokens, and whether to account for decoding optimizations.
> - Dollar cost depends on third-party pricing, which may not be available for open-source models.
> - Wall-clock time is sensitive to parallelization, machine load, and requires averaging over multiple runs.
>
> Steps are easy for any evaluator to report and are comparable across diverse architectures without additional assumptions.
>
> We added a dedicated discussion of this limitation in the revision, recommending that Kuiper be reported alongside complementary cost measures when comparing across heterogeneous architectures. We are open to adopting alternative measures if the community converges on a standard.

---

> > ### Author Rebuttal · Reviewer_xNxE · 2026-04-04
> >
> > I have read the responses from the authors and the other reviewers, and I will maintain my score.

---

### Official Review · Reviewer_kCRz · 2026-03-13

**Soundness:** 4
**Presentation:** 3
**Significance:** 2
**Originality:** 3
**Overall Recommendation:** 4
**Confidence:** 3

**Summary:**

This paper introduces MADQA, a benchmark of 2,250 questions over 800 pdf documents. The corpus of pdfs includes varied layouts, including forms, tables, dense text, and figures. The benchmark aims to test whether AI agents reason strategically or search by trial and error. The authors find that humans calibrate their search effort far better than frontier models, and that retrieval failure (rather than reasoning) is the primary bottleneck for agents.

**Compliance With Llm Reviewing Policy:**

Affirmed.

**Key Questions For Authors:**

See above.

1. Given that around 11% of questions were answerable without any documents, could the authors clarify why these were retained given the potential for training data contamination?
2. Could the authors provide more detail on how questions were distributed across annotators, and whether inter-annotator agreement was measured on the baseline task itself?

**Limitations:**

Yes.

**Strengths And Weaknesses:**

Strengths: One clear strength is that all questions are written by professional annotators rather than generated by AI. The test split also uses Classical Test Theory to select questions that best separate strong from weak models. They use the Kuiper statistic to measure whether agents that expend more steps are actually solving harder problems, and to my understanding, the use of this metric is a novel contribution. The human baseline is also well-designed as annotators used the same search tool as the AI agents, allowing a fair side-by-side comparison of behavior. Also, the corpus is impressively broad and varied (63 document types across 13 domains with varying layouts).

Weaknesses: While the dataset that the paper contributes is valuable, some claims in the paper may reach further than the evidence supports.
1. It is claimed that questions satisfy the "agentic" property. While it is verified that lexical retrieval fails to retrieve all evidence in one shot, this only demonstrates a weaker version of the claimed property. Verifying the claimed property would require showing that semantic retrieval systems would be unable to retrieve the evidence.
2. Since over 80% of the questions are single-hop (noted in the paper), the framing of this benchmark being multi-hop at the start feels a bit overstated.
3. The findings also suggest that agents fail mostly because their search terms don't match the language in the document. The multiple causes weakens the paper's central claim: that agentic systems fail due to a lack of strategic reasoning.

The authors could strengthen the paper by pulling back on some of these claims. For (1) and (2), it would be more accurate to present the benchmark properties as partially or not fully verified, rather than fully satisfied. For (3), the paper would benefit from more clearly acknowledging that there are multiple causes of failure, so readers are not left with the impression that poor strategic reasoning is the dominant one, or alternatively, the paper would benefit from more justification to support this claim.

---

> ### Author Rebuttal · Authors · 2026-03-30
>
> **We appreciate Reviewer kCRz's concrete suggestions.** As recommended, we will:
>
> - Clarify that the agentic and multi-hop properties are merely targeted (with a target much below 100% for multi-hop).
> - Clearly state that multiple causes contribute to failure, not only strategic reasoning.
>
> ---
>
> ## Weakness 1 (W1)
> Please see Reviewer 1mvH (W1).
>
> ## Weakness 2 (W2)
> We'll soften the multi-hop framing. In fact, multi-hop is one of six properties, "targeted by design" at ~20%. The rationale for the single-hop majority:
>
> 1. A multi-hop-dominated benchmark would bias toward some patterns and miss the retrieval challenges of single-hop questions (the majority of real-world workflows). A single-hop majority is also needed to detect over-searching.
>
> 2. Single-hop questions can also benefit from iterative search. Our analysis (Reviewer 1mvH, W1) shows that many aren't retrievable by dense retrieval at Recall@10. Even when evidence is found, additional iterations can increase confidence and turn a refusal into an answer. Our reformulation analysis (Reviewer wGS7, W2) shows that success depends on how aggressively agents reformulate after a failed or uncertain search (regardless of hop count).
>
> ## Weakness 3 (W3)
> We agree. Our error decomposition (Reviewer wGS7, W2) shows that failures stem from multiple causes: retrieval, navigation, comprehension, and refusal. Vocabulary mismatch contributes to several of these.
>
> One diagnostic that reveals systematic differences between agents is reformulation behavior. Sonnet aggressively reformulates after a miss (drift 0.38, >90% recovery), while GPT-4.1 Nano barely adjusts (0.10, 12% recovery). Humans need recovery 2-3x less often than strong models and have the highest recovery success rate and efficiency (what we called being more *strategic*).
>
> This vocabulary gap is inherent to real-world search: before the first query, the agent has no corpus vocabulary and must retrieve, learn terminology, and reformulate. Reformulation is one of several skills that contribute to success, and we will frame it accordingly.
>
> ## Question 1 (Q1)
> **Different models memorize different facts.** The ~11% figure decomposes into ~3% random chance on yes/no and binary-choice questions and ~8% genuine memorization: models recalling facts from public documents seen during pretraining (this is memorization of *facts* from public documents, e.g., knowing that a specific company is from Norway).
>
> Pairwise Jaccard overlap across six models is low (0.18-0.50); only 4/500 questions (0.8%) are memorized by all six. Removing them would shift which models benefit. Each new model memorizes a different subset.
>
> **The 8% is an upper bound measured under a different setup** where we encouraged models to answer without sources. In the actual evaluation setup, models are instructed to search for information and cite sources (discouraging the use of parametric knowledge).
>
> **Attribution metrics penalize ungrounded answers.** Our evaluation includes Page F1, which penalizes systems that do not ground their answers in retrieved evidence. A model that answers from memory without citing sources will receive a score of 0 for attribution.
>
> Consequently, we believe retaining these questions and transparently reporting the memorization bound is more principled than removing a model-dependent subset that would change with every new model release.
>
> ## Question 2 (Q2)
>
> We distinguish two processes:
>
> **Data annotation**: annotators received documents and created (Q, A) pairs. Since each produced different questions, IAA is not computable. *Question distribution:* documents were assigned to annotators by category; each annotator worked on a dedicated set of document categories. Quality was ensured via pilot (20 candidates), expert supervision, post-hoc review (<1% replaced, <5% extended; Sec. 2.2).
>
> **Human baseline** (Sec. 4.1, Appendix C.2): annotators received (document collection, Q) and produced A. IAA is meaningful here. *Question distribution:* 500 test questions distributed equally across 5 annotators (100 each, no overlap, different pool from annotation team). We assume this is what the reviewer is asking about.
>
> **Inter-annotator agreement on the human baseline.** A co-author re-annotated 50 stratified questions using the identical BM25 interface, without seeing original answers. We report agreement under three independent measures:
>
> | Metric | Agreement |
> |---|---|
> | ANLS* >= 0.5 | 43/50 (86.0%) |
> | LLM judge | 42/50 (84.0%) |
> | Manual review | 44/50 (88.0%) |
>
> Of the 6 disagreements per manual review, only 2 are genuine factual disagreements; the remaining 4 are differences in answer granularity (same core information at different levels of detail).
>
> **Note:** This measures agreement between two humans attempting the same search task, not annotation quality. Humans achieved ~82% accuracy on this benchmark, and the 84-88% agreement rate is consistent with this (other humans make errors in a few different places).

---

> > ### Author Rebuttal · Reviewer_kCRz · 2026-04-05
> >
> > Thank you for the rebuttal, my questions have been addressed.

---

> > > ### Author Response · Authors · 2026-04-05
> > >
> > > We're glad the clarifications were helpful. If you feel they merit an update to your score, it would mean a lot to us.
> > >
> > > Regardless, we are very grateful for your insights and your time.

---

### Official Review · Reviewer_wGS7 · 2026-03-14

**Soundness:** 3
**Presentation:** 3
**Significance:** 3
**Originality:** 3
**Overall Recommendation:** 5
**Confidence:** 3

**Summary:**

This paper introduces a benchmark of 2,250 human-authored questions over 800 heterogeneous PDFs and a Kuiper-statistic-based metric for measuring effort-accuracy calibration in agentic document QA. Experiments show that even when the best agents match human accuracy, they exhibit substantially worse effort calibration.

**Compliance With Llm Reviewing Policy:**

Affirmed.

**Final Justification:**

I thank the authors for the thorough and well-structured rebuttal. I raise my score from 4 to 5.

**Key Questions For Authors:**

1. For low-$K$ but low-accuracy systems (e.g., GLM-4.6V Flash BM25 Agent, 46.0% acc, $K$=27.5), how should one distinguish genuine calibration from uniformly poor performance?
2. How sensitive are accuracy and $K$ to the agent iteration budget ($T$=10, $k$=5)?
3. Why is cross-document multi-hop easier than same-document (75.7% vs. 61.2% in Appendix H.1)? Could this reflect question design bias?

**Limitations:**

yes

**Strengths And Weaknesses:**

**Strengths:**
1. **Rigorous benchmark construction.** Fully human-authored annotations, Classical Test Theory-based splits, and a sentinel pool for long-term headroom. The construct validity analyses (lexical overlap, guessability, visual necessity) are convincing.
2. **Useful evaluation framework.** The Kuiper calibration metric and the Page $F_1$ / Doc $F_1$ decomposition go beyond standard accuracy reporting and offer actionable diagnostics.
3. **Well-controlled experiments.** Broad baselines (managed RAG, agents, RLMs) with humans tested under matched conditions. The cost-accuracy tradeoff analysis is practically informative.

**Weaknesses:**
1. **Kuiper metric interpretation is ambiguous.** Low $K$ can reflect good calibration or effort-invariant behavior — the paper would benefit from a more formal way to distinguish these cases.
2. **The calibration gap analysis stays descriptive.** The error taxonomy (Appendix H) and the calibration analysis are not explicitly connected, leaving the mechanistic "why" underexplored.
3. **English-only, US-centric corpus** limits generalizability, as the authors acknowledge.

---

> ### Author Rebuttal · Authors · 2026-03-30
>
> We thank Reviewer wGS7 and agree that the Kuiper metric deserves more attention.
>
> ---
>
> ## Weakness 1 and Question 1 (W1, Q1)
>
> We agree that $K$ alone does not distinguish good calibration from effort-invariant behavior and should be read jointly with accuracy (Section 3.3).
>
> As keeping both is diagnostically valuable, we propose two checks:
>
> 1. **Adaptivity sanity check.**  CV = std(steps)/mean(steps). If  $CV ≈ 0$, the system is non-adaptive, and $K$ is not meaningful.
>
> 2. **Budget-response A(N).** Varying effort does not imply that extra effort is useful. Thus, we compute accuracy when capped at $N$ steps and test $H0: A(1) = ... = A(T)$. If $H0$ is rejected, extra compute has value and $K$ is informative about how efficiently the agent allocates effort.
>
> All evaluated agentic systems show high $CV$ (Gemini 3 Pro, GLM-4.6V Flash, Claude Sonnet 4.5, GPT-5 all > 0.75).
>
> In terms of A(N), the pattern differs. Frontier models gain significantly through step 10. **GLM-4.6V Flash gains significantly only through step 6**, plateauing early. Its low $K$ reflects near-saturation, not good calibration.
>
> | Model | Sig. through | K |
> |---|---:|---:|
> | Gemini 3 Pro | 9→10 | 25.8 |
> | Claude Sonnet 4.5 | 9→10 | 35.1 |
> | GPT-5 | 9→10 | 52.6 |
> | GLM-4.6V Flash | 5→6 | 27.5 |
>
> We'll make it more prominent that $K$ should be read jointly with acc., and include A(N) as a standard diagnostic.
>
> ## Weakness 2 (W2)
>
> To connect error analysis with the calibration findings, we provide three complementary results. High-$K$ systems fail at retrieval, reformulate too conservatively, and do not recover.
>
> **(1) Error decomposition.** We study a cascade of failures: retrieval (document never found, 35.7%), navigation (right document, wrong page, 23.0%), comprehension (right page, wrong answer, 28.8%), and refusal (12.6%). Weaker models are dominated by refusals and retrieval failures, while stronger by comprehension errors.
>
> **(2) Query reformulation and (3) recovery predict success.** We embed search queries and measure the cosine drift between consecutive queries. Sonnet reformulates aggressively after a miss (mean drift 0.38) and recovers >90% of the time; GPT Nano barely adjusts its queries (drift 0.10) and recovers only 12%. Overall, poorly calibrated systems (high $K$) persist through minor query variations, while well-calibrated systems succeed quickly or pivot substantially. Humans recover
> 97% of the time, and need for recovery is rarer.
>
> ## Question 2 (Q2)
> $K$ and accuracy are sensitive to $T$. We will urge to interpret them in the context of the evaluation budget ($T$, $k$).
>
> **Sensitivity to step budget T.** Acc. and $K$ grow with $T$ for all systems, but the rate depends on the model. This happens because if extra steps do not convert them into correct answers, some hard questions move further into the high-effort tail. Under low $T$, the effort is flattened, and $K$ is less meaningful.
>
> | T | Gemini 3 Pro (Acc / K) | Claude Sonnet 4.5 (Acc / K) | GPT-5 (Acc / K) |
> |---|---|---|---|
> | 3 | 59.3% / 9.9 | 57.3% / 6.4 | 51.6% / 5.4 |
> | 5 | 71.8% / 11.6 | 68.4% / 11.4 | 62.4% / 6.9 |
> | 10 | 82.2% / 25.8 | 80.6% / 35.1 | 77.7% / 52.6 |
>
> **Sensitivity to retrieval width k.** We reran two models with k={3, 5, 10}, and other parameters identical (k=5 is near the knee of the recall curve; 91% of searches return fewer results). Acc. varies by only 2.0pp (Gemini) and 5.1pp (GPT-5). $K$ is largely invariant to $k$ (Gemini: 25-28; GPT-5: 59-67).
>
> | | Gemini 3 Pro (Acc / K) | GPT-5 (Acc / K) |
> |---|---|---|
> | k=3 | 80.2% / 25.5 | 69.7% / 67.0 |
> | k=5 | 81.8% / 25.3 | 72.6% / 61.1 |
> | k=10 | 82.2% / 27.8 | 74.8% / 59.1 |
>
> ## Question 3 (Q3)
> Our analysis shows that the gap reflects a **structural property of the task**, not a design bias. Observed properties are inherent to how X-Doc questions arise and are difficult to avoid without artificially constraining the question types:
>
> 1. **The gap is consistent but not universal.** 25/35 systems show X-Doc > X-Page, but the gap is largest for mid-tier agents (+13-18pp), shrinks for the strongest agents (+6pp for Gemini 3 Pro), and reverses for humans (-7.6pp). If the questions were simply easier, humans would benefit too. This suggests a capability threshold.
>
> 2. **X-Doc evidence is harder, not easier to find.** As shown in Appendix H.1, semantic distance between evidence pages is a strong predictor of difficulty, but X-Doc evidence pages are actually *more* distant from each (mean 0.51 vs 0.38). Gap persists when we control for semantic distance.
>
> 3. **The gap is explained by answer complexity.** X-Doc questions naturally gravitate toward easily comparable values, like totals or dates. 27% of X-Doc questions involve comparisons (vs 5% for X-Page) and they yield short answers (mean 22.6 chars). X-Page more often requires connecting scattered details, producing multi-item lists (52.6 chars).
>
> Other parameters (e.g, number of evidence pages, domains) are similar across subsets.

---

> > ### Author Rebuttal · Reviewer_wGS7 · 2026-04-05
> >
> > I thank the authors for the thorough and well-structured rebuttal. I raise my score from 4 to 5.

---

### Official Review · Reviewer_1mvH · 2026-03-14

**Soundness:** 3
**Presentation:** 3
**Significance:** 3
**Originality:** 2
**Overall Recommendation:** 5
**Confidence:** 3

**Summary:**

The authors introduce MADQA, a benchmark of 2,250 human-authored questions over 800 heterogeneous PDFs, designed to evaluate multimodal agentic document QA. They formalize six task properties distinguishing agentic reasoning from standard document QA, apply Classical Test Theory to construct discriminative evaluation splits, introduce the Kuiper statistic to measure effort-accuracy calibration, and compare humans against agentic and non-agentic systems.

**Compliance With Llm Reviewing Policy:**

Affirmed.

**Key Questions For Authors:**

1. The LLM judge calibration reports perfect specificity on 200 samples. How does judge agreement hold on the subset of questions involving multi-part list answers, unit qualifiers, or numerical values, where partial credit and format ambiguity are most likely to cause systematic disagreement with human judgment?

**Limitations:**

yes

**Strengths And Weaknesses:**

### Strengths:

1. The annotation pipeline is rigorous with fully human-authored questions, 1,200+ hours of professional annotation, a two-step quality filter using GPT-5 to flag unanswerable instances, and a construct validity analysis that measures guessability, lexical overlap, and visual necessity.

2. The Classical Test Theory split construction is a principled alternative to arbitrary train/test splits, and the explicit reservation of sentinel items unsolvable by current models is a practical contribution to benchmark longevity.

3. The Kuiper statistic as an effort calibration metric is well-motivated and the human comparison is unusually thorough, with full trajectory logging enabling direct comparison of search strategies rather than just accuracy.

4. The finding that semantic distance between evidence pages predicts difficulty better than physical page distance is an interesting and non-obvious empirical result.

### Weakness:

1. The agentic property is defined as requiring iterative retrieval where no single query suffices, but the verification of this property relies on n-gram matching rather than semantic retrieval. A semantic retrieval baseline that fails to surface all evidence would be a stronger confirmation that the property holds at inference time for modern dense retrieval systems.

2. The LLM judge for answer correctness achieves near-perfect specificity (1.00) on a 200-sample calibration set, which is suspiciously high and likely reflects the clean, extractive nature of the answer format rather than genuine judge reliability on hard cases. The calibration set size is small relative to the diversity of the benchmark, and the bias correction applied is minimal by construction, making it difficult to assess whether the judge handles edge cases in list formatting and partial matches consistently.

---

> ### Author Rebuttal · Authors · 2026-03-30
>
> **We thank Reviewer 1mvH for the detailed evaluation and for highlighting the annotation rigor**, CTT splits, and the semantic distance finding.
>
> ---
>
> ## Weakness 1 (W1)
> **Property status.** The Agentic property is "targeted by design" (Section 1.2). Our protocol encourages them, but not every question requires multiple retrieval steps.
>
> This is intentional: real-world workflows contain a mix of retrieval problems, and benchmarks biased exclusively toward multi-hop would overrepresent specific question types at the expense of others.
>
> Moreover, having a significant fraction of questions that *can* be solved in a single pass is essential: without a large fraction of single-hop, we could not detect over-searching.
>
> **New experiment: semantic retrieval.** We embed all pages using Qwen3-Embedding, rank by cosine similarity, and check if required evidence appears in the top 5 or 10 results.
>
> |  | Overall | Single-hop | X-Page | X-Doc |
> |--|--|--|--|--|
> | Recall@5 (4B) | 65.0% | 73.6% | 39.0% | 47.1% |
> | Recall@5 (8B) | 65.0% | 72.5% | 37.8% | 56.9% |
> | Recall@10 (4B) | 73.2% | 79.8% | 51.2% | 62.7% |
> | Recall@10 (8B) | 72.6% | 78.1% | 54.9% | 62.7% |
>
> At Recall@5, 35% are unsolvable in a single pass, rising to ~61% for cross-page multi-hop. At Recall@10, 27% remain unsolvable.
>
> Retrieving evidence is an **upper bound** on agent performance. **Agents can benefit from additional iterations**, e.g., increasing the confidence and turning answer rejection into providing an answer.
>
> ## Weakness 2 (W2)
> In our calibration sample, the judge never promoted a wrong answer to a correct one.
>
> The judge is not invoked when an exact string match exists (~half of predictions, achieving 1.0 specificity and sensitivity). For the remaining half, the prompt (Appendix F1) applies a sequential evaluation:
> - Steps 1-3 each independently route to "incorrect" (critical errors).
> - Steps 4-5 (format and verbosity) fire afterwards and can never rescue a wrong answer.
>
> In practise, for QA with concrete answers, a wrong value almost always hits one of the rejection gates. This leads to a suspiciously-looking, ideal specificity.
>
> The observed 2% sensitivity gap (not specificity gap) comes from the opposite direction: correct answers rejected for excessive verbosity, formatting issues, etc.
>
> **Stress-test on adversarially selected hard cases.** We annotated a sample of 100 predictions from Gemini 3 Pro where: (i) none had an exact string match to the gold answer, and (ii) list-type answers were oversampled to 50%.
>
> |  | LLM=Correct | LLM=Incorrect |
> |---|---:|---:|
> | **Human=Correct** | 45 | 4 |
> | **Human=Incorrect** | 0 | 51 |
>
> Specificity remained at 100%. Sensitivity was 91.8% (4/49 correct predictions rejected). The judge was too strict:
>
> 1. A list of sensor components was correct, but the model wrote "window portion" instead of "a window." The judge applied a "wrong entity/value" rule.
>
> 2. The model correctly identified the detected substance but omitted the concentration. The judge rejected it for "missing detail".
>
> 3. The model listed all items correctly, but as a single string rather than separate list items. The judge scored this as partially correct (Step 4), the human gave a full point because the "list" structure was arguably not desired.
>
> 4. The model answered with a different wording, but this equivalence was only apparent with access to the source document.
>
> ## Question 1 (Q1)
> Our response to W2 explains why the judge's prompt design leads to one-sided errors. To provide the per-category breakdown requested here, we conducted a **stratified human calibration study** on 200 predictions from three frontier models (Gemini 3 Pro, Claude Sonnet 4.5, GPT-5).
>
> | Answer Type | Sensitivity | Specificity |
> |---|---|---|
> | Single value (name/date/number) | 100.0% (42/42) | 100.0% (8/8) |
> | Numerical with units | 97.4% (38/39) | 100.0% (11/11) |
> | Multi-part list | 94.9% (37/39) | 90.9% (10/11) |
> | Yes/No / Binary | 100.0% (42/42) | 100.0% (8/8) |
>
> Multi-part lists are indeed the hardest category: sensitivity on this sample drops to 94.9% (the judge rejects correct lists with minor name variations or formatting differences), and specificity to 90.9%. The single list false positive received a partial score (0.50) for an answer containing all gold items plus one extra item, which could not be verified without document access.
>
> All 3 over-rejections follow the same pattern described in W2: the judge applies its rejection rules too strictly on correct answers (first names only vs. full names, one-character typo in the gold answer, correct number with appended paraphrase). The **single over-acceptance** is the only false positive across all 200 samples.
>
> Across the original calibration, stress-test, and Q1 study, **we now have 500 human-judged predictions**. The latter two are deliberately biased toward hard cases, but they confirm the judge's error profile. We are open to expanding the unbiased calibration set further.

---

> > ### Author Rebuttal · Reviewer_1mvH · 2026-04-03
> >
> > Thank you for the rebuttal, my questions are mostly resolved. I will keep my positive score.

---

### Decision · Program_Chairs · 2026-04-30

**Decision:**

Accept (spotlight)

**Comment:**

This paper introduces MADQA, a rigorously constructed benchmark for evaluating multimodal agentic document QA, which the reviewers universally praised for its high-quality human annotations, principled Classical Test Theory-based splits, and the novel use of the Kuiper statistic to measure effort-accuracy calibration. During the review period, reviewers raised concerns regarding the framing of the benchmark's "agentic" and multi-hop properties (given the high proportion of single-hop questions), the interpretability of the Kuiper metric using step counts as a proxy for effort, and the suspiciously high specificity of the LLM judge. In a comprehensive and well-structured rebuttal, the authors successfully addressed these critiques by softening their claims with new semantic retrieval baselines, proposing additional diagnostic checks for the Kuiper metric, and demonstrating that the LLM judge's perfect specificity stems from an intentionally strict, one-sided rejection-gate design. As all reviewers explicitly acknowledged that their concerns were fully resolved—resulting in a final consensus of positive scores ranging from Weak Accept to Strong Accept—this paper represents a highly solid, well-evaluated contribution to the field and is therefore recommended for acceptance.